# 🎭 PERSONAX: MULTIMODAL DATASETS WITH LLM-INFERRED BEHAVIOR TRAITS

**Loka Li**[1*], **Wong Yu Kang**[1*], **Minghao Fu**[1,3], **Guangyi Chen**[1,2], **Zhenhao Chen**[1],
**Gongxu Luo**[1], **Yuewen Sun**[1,2], **Salman Khan**[1,4], **Peter Spirtes**[2], **Kun Zhang**[1,2]
[1] Mohamed bin Zayed University of Artificial Intelligence, [2] Carnegie Mellon University
[3] University of California San Diego, [4] Australian National University

## ABSTRACT

Understanding human behavior traits is central to applications in human-computer interaction, computational social science, and personalized AI systems. Such understanding often requires integrating multiple modalities to capture nuanced patterns and relationships. However, existing resources rarely provide datasets that combine behavioral descriptors with complementary modalities such as facial attributes and biographical information. To address this gap, we present PersonaX, a curated collection of multimodal datasets designed to enable comprehensive analysis of public traits across modalities. PersonaX consists of (1) CelebPersona, featuring 9444 public figures from diverse occupations, and (2) AthlePersona, covering 4181 professional athletes across 7 major sports leagues. Each dataset includes behavioral trait assessments inferred by three high-performing large language models, alongside facial imagery and structured biographical features.

We analyze PersonaX at two complementary levels. First, we abstract high-level trait scores from text descriptions and apply five statistical independence tests to examine their relationships with other modalities. Second, we introduce a novel causal representation learning (CRL) framework tailored to multimodal and multi-measurement data, providing theoretical identifiability guarantees. Experiments on both synthetic and real-world data demonstrate the effectiveness of our approach. By unifying structured and unstructured analysis, PersonaX establishes a foundation for studying LLM-inferred behavioral traits in conjunction with visual and biographical attributes, advancing multimodal trait analysis and causal reasoning. The code is available at `https://github.com/lokali/PersonaX`.

🤗 **CelebPersona:** huggingface.co/datasets/Persona-X/celebpersona

🤗 **AthlePersona:** huggingface.co/datasets/Persona-X/athlepersona

## 1 INTRODUCTION

Human behavior traits (or behavioral summaries) refer to outwardly observable patterns of conduct inferred from public information such as spoken or written language, facial expressions, and biographical records (Rothe, 2017; Johnson, 1997; DeNeve & Cooper, 1998; Briggs & Cheek, 1986). These traits differ from psychological personality, which concerns internal dispositions typically measured through self-reports or expert evaluation (Cattell et al., 1970; Eysenck & Eysenck, 1975; Myers et al., 1998; Goldberg, 1993). Unlike clinical diagnoses, behavior traits can be inferred ethically and at scale from non-intrusive signals, offering reproducible, population-level insights that complement personality research without medicalizing individuals. Advances in large language models (LLMs) (Achiam et al., 2023; Floridi & Chiriatti, 2020; Jiang et al., 2024; Liu et al., 2024) have further expanded this feasibility. Several studies demonstrate that LLM-based assessments of behavior traits aligned with the Big Five framework can be reliable under carefully designed prompting strategies (Serapio-García et al., 2023; Jiang et al., 2023; Tseng et al., 2024; Zou et al., 2024). These approaches enable large-scale, automated analysis while mitigating some biases inherent in self-reports.

**Related Work.** Research on human attributes spans two complementary directions: internal personality and external behavior traits. Psychologically internal personality has traditionally been measured

---

[*]Equal contributions.

Figure 1: **Data processing pipelines** of `AthlePersona` (Left) and `CelebPersona` (Right) datasets. (1) `AthlePersona` was constructed by collecting player rosters and publicly available data (including facial images and basic features) from the official websites of major sports leagues. These data were then processed with LLMs for inferring behavior traits. (2) `CelebPersona` was derived from the CelebA dataset (Liu et al., 2015). Celebrity face identities were linked to their corresponding Wikidata entities, enabling the retrieval of additional biographical details and physical characteristics, which were similarly processed with LLMs for inferring behavior traits.

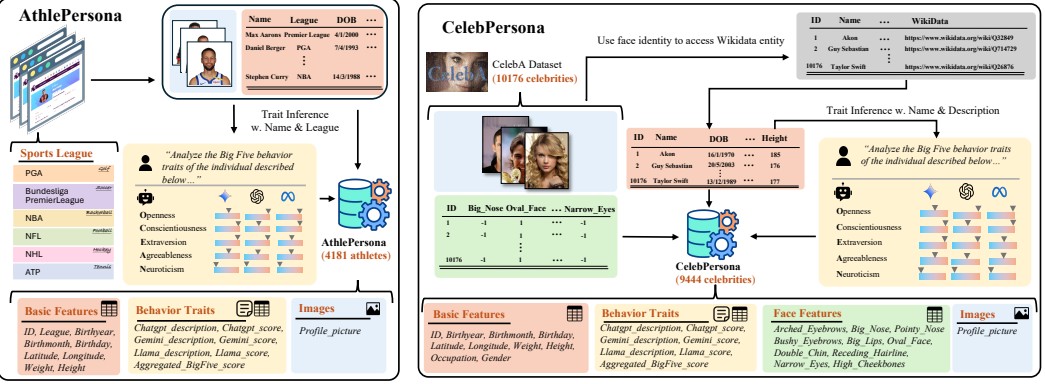

with self-report instruments such as the 16PF (Cattell et al., 1970), EPQ (Eysenck & Eysenck, 1975), MBTI (Myers et al., 1998), and the Big Five framework (Goldberg, 1993). In contrast, behavior traits emphasize outwardly observable patterns, inferred from signals such as text, facial expressions, physiology, or digital traces. Several datasets target this perspective, including SALSA for group interactions (Alameda-Pineda et al., 2015), nonsocial-context datasets for daily activities (Dotti et al., 2018), driving and physiological data for trait prediction (Evin et al., 2022), and lifelog corpora capturing multimodal daily behavior (Chung et al., 2022). Digital records such as Facebook Likes have also been shown to predict sensitive traits, including personality dimensions (Kosinski et al., 2013). Other multimodal resources, such as YouTube-Vlogs (Biel & Gatica-Perez, 2012), FI-V2 (Escalante et al., 2020), MuPTA (Ryumina et al., 2023), and MDPE (Cai et al., 2024), combine video, audio, or physiological signals for prediction tasks like impression analysis or deception detection, but they typically lack explicit textual trait descriptions or frameworks for cross-modal interpretation. A comparison table of different datasets is in Tab. A1. Beyond datasets, empirical studies show that observable features in one modality can signal traits in another. For instance, facial structure has been linked to health and aggression cues (Kramer & Ward, 2010; Carré & McCormick, 2008), body images to personality judgments (Naumann et al., 2009), and facial behavior to Big Five traits (Cai & Liu, 2022). Together, these works underscore the promise of behavior trait analysis, but existing resources are limited for systematic cross-modal and causal study. See App. A2 for more details.

To address these gaps, we introduce `PersonaX`, a curated collection of multimodal datasets that contain LLM-inferred behavior traits. The assessments are derived from public information, including direct quotes from interviews, observed behaviors, career trajectories, and biographical details. For consistency, we follow the Big Five framework (Goldberg, 1993), providing trait scores across its five dimensions. `PersonaX` includes (i) `CelebPersona`, comprising 9444 public figures from the CelebA dataset (Liu et al., 2015), and (ii) `AthlePersona`, covering 4181 professional athletes across seven major sports leagues. Each record integrates (1) textual trait descriptions and Big Five scores inferred by three high-performing LLMs, (2) facial images, and (3) structured biographical metadata. To safeguard privacy, we release only transformed embeddings rather than raw images or text. The proposed dataset provides a unique foundation for cross-modal and causal analysis.

Our contributions are mainly twofold. (i) We release `PersonaX`, a set of multimodal datasets that combine LLM-inferred behavior traits, facial embeddings, and biographical metadata for large populations of public figures. (ii) We introduce a two-level analysis framework: at the structured level, applying diverse independence tests to uncover behavior-trait dependencies; and at the unstructured level, proposing a causal representation learning approach with identifiability guarantees tailored to multimodal, multi-measurement settings. Experiments on both synthetic and real-world data demonstrate the practical effectiveness of this framework. By unifying structured and unstructured perspectives, `PersonaX` enables systematic study of LLM-inferred traits alongside visual and

Table 1: **Evaluations on LLM selection** with `CelebPersona` and `AthlePersona` subsets. Metrics consist of generation time (GT), missing rate (MR), indecisive rate (IR), privacy preservation (PP), output formatting (OF), context consistency (CC), factual accuracy (FA), and an **overall score (OS)**. Please refer to App. A4.1 for more details about models and the definition of metrics.

| Model (LLMs) | CelebPersona | | | | | | | | AthlePersona | | | | | | | |
|---|---|---|---|---|---|---|---|---|---|---|---|---|---|---|---|---|
| | GT↓ | MR↓ | IR↓ | PP↑ | OF↑ | CC↑ | FA↑ | OS↑ | GT↓ | MR↓ | IR↓ | PP↑ | OF↑ | CC↑ | FA↑ | OS↑ |
| ⑤ ChatGPT-4o | 4.19 | 0.03 | 0.17 | 0.99 | 1.00 | 1.00 | 1.00 | 0.96 | 3.92 | 0.27 | 0.17 | 1.00 | 1.00 | 0.99 | 1.00 | 0.93 |
| ✦ Gemini2.5-Pro | 23.48 | 0.06 | 0.19 | 0.99 | 1.00 | 1.00 | 1.00 | 0.96 | 21.31 | 0.29 | 0.22 | 0.99 | 1.00 | 1.00 | 1.00 | 0.91 |
| 🐋 Qwen2.5-Max | 9.10 | 0.24 | 0.29 | 1.00 | 0.99 | 0.99 | 1.00 | 0.91 | 8.93 | 0.32 | 0.36 | 1.00 | 0.99 | 0.99 | 1.00 | 0.88 |
| ⌀ Grok-3-Beta | 5.92 | 0.34 | 0.17 | 1.00 | 1.00 | 0.99 | 1.00 | 0.91 | 4.96 | 0.66 | 0.10 | 1.00 | 1.00 | 1.00 | 1.00 | 0.87 |
| ∞ Llama-4 | 3.73 | 0.25 | 0.29 | 1.00 | 1.00 | 0.97 | 1.00 | 0.90 | 3.99 | 0.30 | 0.43 | 1.00 | 1.00 | 0.95 | 1.00 | 0.87 |
| ✦ Gemini2.0-FT | 8.83 | 0.28 | 0.38 | 0.99 | 1.00 | 1.00 | 1.00 | 0.89 | 8.26 | 0.48 | 0.27 | 0.97 | 1.00 | 0.99 | 1.00 | 0.87 |
| 🐳 DeepSeek-R1 | 39.13 | 0.40 | 0.11 | 0.98 | 0.90 | 1.00 | 1.00 | 0.89 | 26.61 | 0.64 | 0.10 | 1.00 | 0.81 | 1.00 | 0.98 | 0.84 |
| 🐋 QwQ-32B | 9.22 | 0.24 | 0.46 | 1.00 | 0.99 | 1.00 | 1.00 | 0.88 | 8.87 | 0.30 | 0.48 | 1.00 | 0.98 | 1.00 | 1.00 | 0.87 |
| 🐳 DeepSeek-V3 | 14.18 | 0.47 | 0.24 | 0.99 | 1.00 | 1.00 | 1.00 | 0.88 | 7.75 | 0.64 | 0.20 | 1.00 | 1.00 | 1.00 | 1.00 | 0.86 |
| ✦ Gemini2.0-F | 2.53 | 0.43 | 0.32 | 1.00 | 1.00 | 0.99 | 1.00 | 0.87 | 2.29 | 0.69 | 0.18 | 1.00 | 1.00 | 1.00 | 1.00 | 0.86 |

biographical attributes, opening new pathways for deeper multimodal interpretation and causal reasoning. Our long-term vision is to leverage such resources to uncover invariant causal patterns across populations, thereby advancing diversity, equality, and mutual respect for all human beings.

# 2    PERSONAⅩ DATASET

In this section, we introduce `PersonaX`, a collection of two complementary multimodal datasets: `AthlePersona` and `CelebPersona`. Together, they provide large-scale resources for studying LLM-inferred behavior traits in conjunction with visual and biographical attributes. We first describe the construction of each dataset below, then detail the selection of LLMs and prompts for trait generation in § 2.1, and finally discuss consent, privacy, and bias considerations in § 2.2.

**AthlePersona.** Built from scratch, this dataset documents 4181 male professional athletes across seven major sports leagues worldwide, including the NBA, NFL, NHL, ATP, PGA, Premier League, and Bundesliga[1]. From official league sources, we collected biographical information (e.g., name, birth date, nationality), physical attributes (e.g., height, weight), and facial images. Nationalities were geocoded into continuous spatial coordinates (latitude and longitude) to support geographic analyses.

**CelebPersona.** This dataset builds on the established CelebA dataset (Liu et al., 2015), which contains rich facial attribute annotations. We linked each celebrity's name to its corresponding WikiData entity, enabling retrieval of additional biographical details and physical characteristics. From the original 40 CelebA attributes, we manually retained 10 (e.g., *Big Nose, High Cheekbones*) that reflect more stable, inherent appearance properties, while discarding attributes subject to short-term variation (e.g., *Heavy Makeup*). `CelebPersona` totally contains 9444 public figures.

**Multimodality.** Each record integrates three components: (1) textual behavior-trait descriptions and Big Five scores inferred by LLMs, (2) facial images or embeddings with attribute annotations, and (3) structured biographical metadata. The full feature lists of `AthlePersona` and `CelebPersona` are shown in Tab.A2 and Tab.A3, respectively. Dataset distributions for each feature are presented in Fig.A1 and Fig. A2. Terms-of-use compliance across all sports leagues for `AthlePersona` is summarized in Tab.A4. The complete prompt for inferring behavior traits is provided in Prompt 1.

## 2.1    LLM SELECTION AND PROMPT DESIGN

We systematically evaluated ten state-of-the-art LLMs[2] across both `AthlePersona` and `CelebPersona`. A summary of those model performances is shown in Tab. 1. Full model details,

---

[1]We also examined four additional leagues (MLB, La Liga, Serie A, and Ligue 1), but could not include them in the current release due to pending written consent requirements for academic research. A complete summary of terms-of-use compliance including the original statements is provided in Tab. A4. See App.A3.4 for details.

[2]Initially, we considered the Top 10 models from the Arena leaderboard (Chiang et al., 2024) on April 10, 2025, supplemented with Qwen2.5-Max and QwQ-32B (Bai et al., 2023) for diversity. GPT-4.5-Preview (Achiam et al., 2023) was excluded due to high API costs, and Gemini-2.0-Pro-Exp-02-05 (Team et al., 2023) was merged into the later Gemini-2.5-Pro release. Thus, ten models were ultimately retained for evaluation.

Figure 2: **Evaluation on LLM consistency for prompt design. Top**: Radar plots show the standard deviation (std) of Big Five trait scores across repeated runs under different prompt formats, for each model (by column) and dataset (by row). **Middle**: Box plots summarize the average of std across Big Five behavior traits, highlighting *intra-prompt* variability. **Bottom**: Manhattan distances between two prompt pairs quantify *inter-prompt* variability. Refer to § 2.1 for more setup and result analysis.

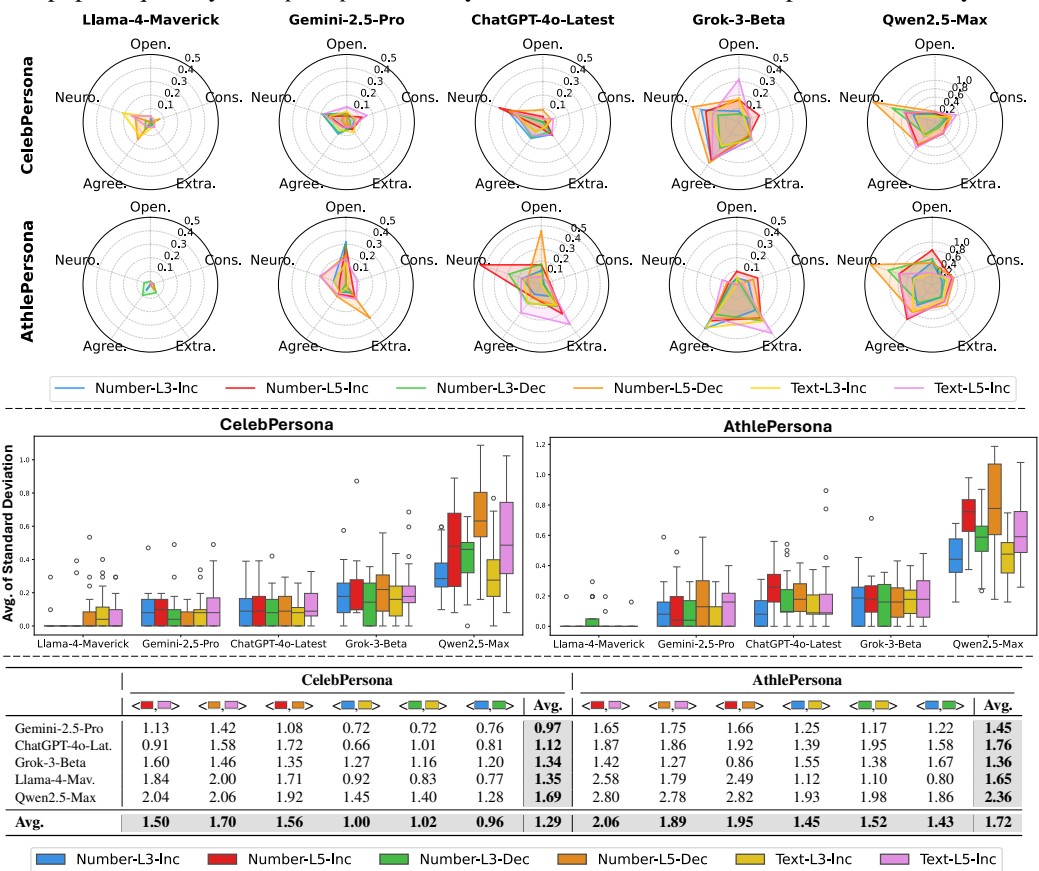

| | CelebPersona | | | | | | | AthlePersona | | | | | | |
|---|---|---|---|---|---|---|---|---|---|---|---|---|---|---|
| | ◁■,□▷ | ◁■,□▷ | ◁■,□▷ | ◁■,□▷ | ◁■,□▷ | ◁■,□▷ | Avg. | ◁■,□▷ | ◁■,□▷ | ◁■,□▷ | ◁■,□▷ | ◁■,□▷ | ◁■,□▷ | Avg. |
| Gemini-2.5-Pro | 1.13 | 1.42 | 1.08 | 0.72 | 0.72 | 0.76 | **0.97** | 1.65 | 1.75 | 1.66 | 1.25 | 1.17 | 1.22 | **1.45** |
| ChatGPT-4o-Lat. | 0.91 | 1.58 | 1.72 | 0.66 | 1.01 | 0.81 | **1.12** | 1.87 | 1.86 | 1.92 | 1.39 | 1.95 | 1.58 | **1.76** |
| Grok-3-Beta | 1.60 | 1.46 | 1.35 | 1.27 | 1.16 | 1.20 | **1.34** | 1.42 | 1.27 | 0.86 | 1.55 | 1.38 | 1.67 | **1.36** |
| Llama-4-Mav. | 1.84 | 2.00 | 1.71 | 0.92 | 0.83 | 0.77 | **1.35** | 2.58 | 1.79 | 2.49 | 1.12 | 1.10 | 0.80 | **1.65** |
| Qwen2.5-Max | 2.04 | 2.06 | 1.92 | 1.45 | 1.40 | 1.28 | **1.69** | 2.80 | 2.78 | 2.82 | 1.93 | 1.98 | 1.86 | **2.36** |
| Avg. | 1.50 | 1.70 | 1.56 | 1.00 | 1.02 | 0.96 | **1.29** | 2.06 | 1.89 | 1.95 | 1.45 | 1.52 | 1.43 | **1.72** |

Legend: Number-L3-Inc ■ Number-L5-Inc ■ Number-L3-Dec ■ Number-L5-Dec ■ Text-L3-Inc ■ Text-L5-Inc ■

including Arena score and API pricing, are given in Tab. A5. Models were assessed on eight criteria, including generation time, missing and indecisive rates, privacy preservation, output formatting, factual accuracy, and context consistency. See App. A4.1 for full metric definitions and model details.

Prompts were carefully designed to balance interpretability and consistency. We experimented with numeric vs. textual outputs (e.g., {1,2,3} vs. {disagree, neutral, agree}), 3-level vs. 5-level scoring scales (e.g., {1,2,3} vs. {1,2,3,4,5}), and different ordering directions (e.g., {1,2,3} vs. {3,2,1}), running controlled trials across all candidate models. Results showed that 3-level scales with numeric outputs minimized variability, whereas 5-level scales increased inconsistency. A detailed comparison of scoring scales and complete prompts is provided in Prompt 3. See App. A4.2 for more details. Based on these evaluations, we selected three consistently best-performing LLMs (i.e., ChatGPT-4o-Latest, Gemini-2.5-Pro, and Llama-4-Maverick) to generate both textual descriptions and Big Five scores for building our datasets. Detailed results and analysis of the experiments are in App. A4.2.

## 2.2 ETHICAL CONSIDERATIONS: CONSENT, PRIVACY, BIAS, AND USAGE

We emphasize four aspects to address ethical and technical concerns: (i) *Consent and legality*: both datasets are derived entirely from legally accessible, consent-based resources, including official sports league websites (non-commercial use), CelebA (non-commercial use), and WikiData (free license). (ii) *Privacy protection*: no raw images or trait texts are released. Each facial image is replaced with a 1024-dimensional embedding, and each textual description with a 3584-dimensional embedding, both further obfuscated through an invertible transformation. Categorical variables are converted into indices. (iii) *Bias*: AthlePersona currently includes only male athletes, while CelebPersona focuses on wealthy, high-visibility individuals. Although findings should be

Figure 3: **Independence test (IT) results and distributions of trait scores.** **(a)** and **(b)** present heatmaps of significant IT results between Big Five behavior traits and other structured features for `CelebPersona` and `AthlePersona`, respectively. Each cell reports "$x/y$," where $x$ is the number of methods that reject the null hypothesis ($p < 0.05$) and $y$ is the total number of applied methods. Lighter shades indicate stronger evidence of dependence. **(c)** shows the overall distribution of Big Five behavior scores across both datasets. Refer to Tab. A3 and Tab. A4 for complete $p$-values.

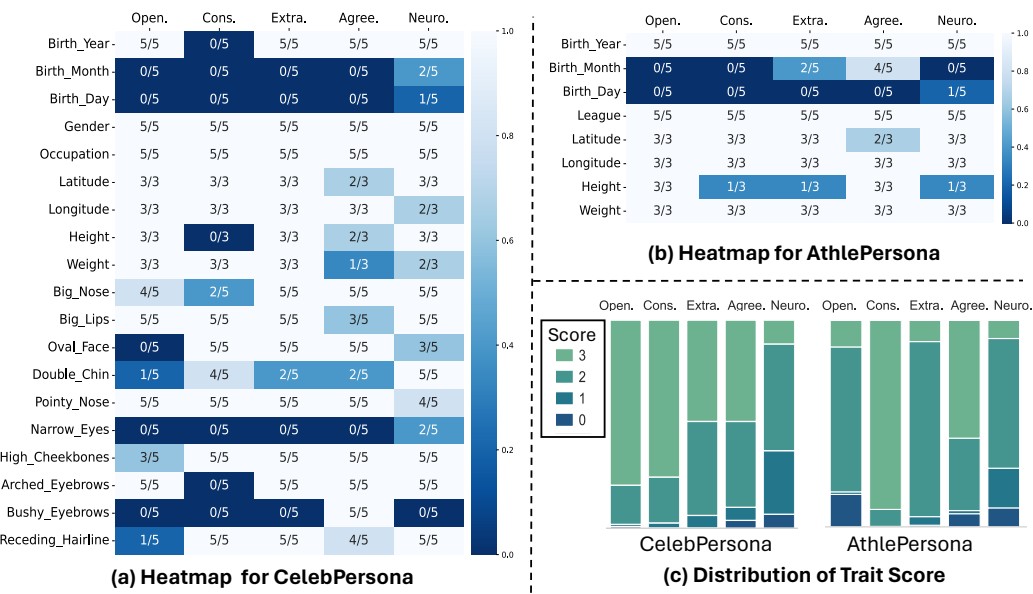

interpreted as population-specific rather than universal, these focused cohorts provide consistency, and the diversity across domains (e.g., different sports leagues) creates valuable opportunities to study invariant causal patterns. (iv) *Usage restrictions*: a mandatory usage guideline limits the dataset to non-commercial use and prohibits applications in high-stakes contexts (e.g., insurance or lending).

## 3 ANALYSIS LEVEL I: INFERRING STATISTICAL DEPENDENCE FROM STRUCTURED DATA

We analyze both datasets at two levels, starting with structured tabular data. For each individual, trait scores are derived by prompting three LLMs to generate text descriptions, which are then mapped to Big Five trait scores. To ensure robustness, we remove "0" scores (denoting insufficient information) and aggregate the remaining values using a median-based voting rule that minimizes sensitivity to outliers. For `CelebPersona`, facial attributes across multiple images are also aggregated into a single stable value per person through majority voting. More implementation details on the voting and aggregation procedures for trait scores and facial attributes are provided in App. A5.1.

To examine dependencies between trait scores and other structured features, we apply five different independence test methods: three non-parametric approaches (KCI (Zhang et al., 2012), RCIT (Strobl et al., 2019), HSIC (Gretton et al., 2005)) and two tests designed for discrete variables (Chi-square (Tallarida et al., 1987) and G-square (Tsamardinos et al., 2006)). The detailed descriptions about these methods are summarized in Tab. A6. A dependency is deemed significant if $p < 0.05$. Fig.3 shows how many of the five methods found significant dependence, and plots the score distributions. Complete $p$-value results for each independent test method are reported in Tab. A3 and Tab. A4.

The heatmaps in Fig.3 reveal clear and interpretable dependency patterns across celebrities and athletes. In `CelebPersona`, demographic attributes such as gender and occupation exhibit strong dependence with nearly all trait scores, whereas in `AthlePersona`, stronger dependencies arise with birth year and league affiliation. Physical attributes display divergent effects: celebrities show significant associations between facial features (e.g., pointy nose, arched eyebrows) and trait scores, while athletes exhibit more consistent yet moderate associations with height and weight. Geographic variables (latitude/longitude) demonstrate comparable moderate dependence in both

datasets, suggesting stable spatial influences. Taken together, these findings highlight systematic differences in information transfer mechanisms across persona types: celebrity representations are more strongly shaped by appearance cues, whereas athlete representations are more heavily influenced by organizational affiliation. This provides novel insights into the structure of human behavioral traits across varying social contexts. More detailed analyses are provided in App.A5.2 and App. A5.3.

# 4 ANALYSIS LEVEL II: LEARNING CAUSAL RELATIONS FROM UNSTRUCTURED DATA

Instead of using well-built tabular data, here we aim to directly learn the latent variables and their underlying causal mechanisms from unstructured data, such as text and images. This task has been widely studied as causal representation learning (CRL) (Xu et al., 2024; Zheng et al., 2022; Yao et al., 2023; Daunhawer et al., 2023; Sturma et al., 2023; Sun et al., 2025). Our persona datasets inherently contain both multi-measurement and multi-modal information [3], capturing rich observations across diverse formats. Inspired by that, we therefore design a corresponding multi-modality multi-measurement CRL method. Fig.4 shows the causal model of our unique problem setting. Our framework unifies and extends prior work (Yao et al., 2023; Sun et al., 2025), supported by a new identifiability theory specifically tailored to the multi-modality, multi-measurement setting. The

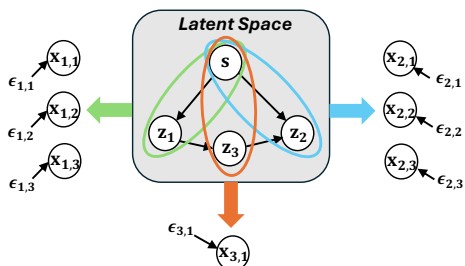

Figure 4: Multi-modality multi-measurement causal model. Latent space is in grey. $\mathbf{s}$ is shared latent variables across different modalities, $\mathbf{z}$ is modality-specific latent variables. $\mathbf{x}_{m,i}$ denotes the $m$-th modality $i$-th observed measurement. $\epsilon$ is the independent noise term.

overall structure of this section is as follows: we first formulate the causal model in § 4.1, then establish the identifiability theory results in § 4.2, followed by details on network training in § 4.3, synthetic experiments in § 4.4, and real-world analysis on the curated `PersonaX` dataset in § 4.5.

## 4.1 CAUSAL MODEL FORMULATION

**Data-generating processes.** Let $\mathbf{x} := [\mathbf{x}_1, \ldots, \mathbf{x}_M]$ be a set of observations/measurements from $M$ modalities, where $\mathbf{x}_m \in \mathbb{R}^{d_m}$ represents the observation from modality $m$ with dimensionality $d_m$. Let $\mathbf{z} = [\mathbf{z}_1, \ldots, \mathbf{z}_M]$ be the set of causally related latent variables underlying $m$-th modalities. Specifically, the data generation process (see Fig. 4) can be formulated as

$$z_{m,i} := g_{z_{m,i}}(\mathrm{Pa}(z_{m,i}), \mathbf{s}, \epsilon_{m,i}), \quad \text{(latent causal relations)} \tag{1}$$

$$\mathbf{x}_m := g_{\mathbf{x}_m}(\mathbf{z}_m, \boldsymbol{\eta}_m), \quad \text{(generating functions)} \tag{2}$$

where we denote the parents of a variable with $\mathrm{Pa}(\cdot)$. Since we allow for general causal relations within each modality and across multiple modalities, $\mathrm{Pa}(\cdot)$ potentially returns latent variables across multiple modalities. Additionally, we allow the shared latent variable $\mathbf{s}$ generally governing the modality-specific latent variables $\mathbf{z}_m$. The differentiable function $g_{\mathbf{z}}$ encodes the latent causal graph connecting latent components and its Jacobian matrix $\boldsymbol{J}_{g_{\mathbf{z}}}$ can be permuted into a strictly triangular matrix. We use $\epsilon_{m,i}$ to denote the exogenous variable for $z_{m,i}$ and exogenous variables are mutually independent. We use $\boldsymbol{\eta}_m$ to denote modality-specific information independent of other components.

**Definition of Identifiability.** As mentioned previously, our aim was to learn the latent variables underlying each modality and their causal relations. Formally, for two specifications $\boldsymbol{\theta} := \{g_{\mathbf{x}_m}, g_{\mathbf{z}_m}, p(\mathbf{s}), p(\boldsymbol{\epsilon}_m), p(\boldsymbol{\eta}_m)\}_{m=1}^M$ and $\hat{\boldsymbol{\theta}} := \{\hat{g}_{\mathbf{x}_m}, \hat{g}_{\mathbf{z}_m}, \hat{p}(\mathbf{s}), \hat{p}(\boldsymbol{\epsilon}_m), \hat{p}(\boldsymbol{\eta}_m)\}_{m=1}^M$ of the data-generating process Eq. 1 and Eq. 2 that fit the marginal distribution $p(\mathbf{x})$, we would like to show that: given the same $\mathbf{x}$ value, each latent component $\hat{z}_{m,i}$ is equivalent to its counterpart $z_{m,i}$ up to an invertible map $h_{m,i}$, i.e., $\hat{z}_{m,i} = h_{m,i}(z_{m,i})$. This property is known as identifiability.

---

[3]Multi-modality (a.k.a., multi-view) refers to different types of data formats, such as facial images and textual descriptions. Multi-measurement usually denotes the different instantiations of the same modality, for example, celebrity photos captured at different locations or an individual's trait description generated by different LLMs.

Figure 5: **Synthetic experiments**. (a) The synthetic dataset consists of two modalities, colored MNIST and fashion MNIST (LeCun, 1998; Xiao et al., 2017). (b) The underlying true causal graph is shown here. For colored MNIST we generated three different measurements. (c) Experimental results show that our method outperforms the other baselines in terms of both $R^2$ and MCC.

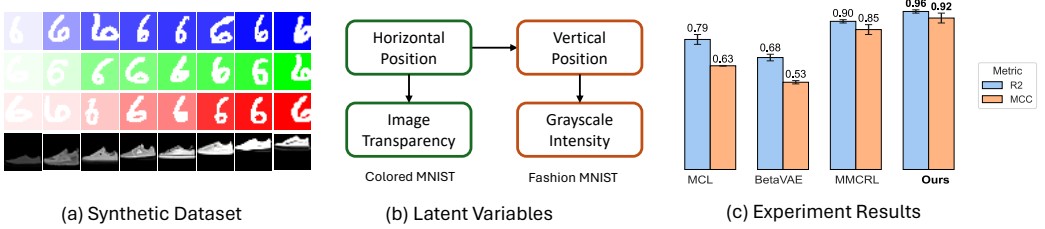

(a) Synthetic Dataset          (b) Latent Variables          (c) Experiment Results

## 4.2 IDENTIFIABILITY THEORY

A central challenge in causal representation learning is ensuring that the learned representations correspond to the true latent variables up to well-defined transformations. We establish theoretical guarantees for the identifiability of our model under specific conditions. The proofs are in App. A6.

**Theorem 1.** *(Identifiability of Subspace) Under the causal model described above, if the estimated observations matches the true joint distribution of any $\{\mathbf{x}_{m,A}, \mathbf{x}_{m,B}, \mathbf{x}_{m,C}\}$ (they are exchangable) which are three measurements draw from one modality, and:*

  i *(Well-Posed Probability): The joint, marginal, and conditional distributions of $(\mathbf{x}_{m,B}, \mathbf{z}_m)$ are all bounded and continuous.*

  ii *(Modality Variability): The operators $L_{\mathbf{x}_{m,C}|\mathbf{z}_m}$ and $L_{\mathbf{x}_{m,A}|\mathbf{x}_{m,C}}$ are injective.*

  iii *(Measurement Changes): For any $\mathbf{z}_t^{(1)}, \mathbf{z}_t^{(2)} \in \mathcal{Z}_t$ where $\mathbf{z}_t^{(1)} \neq \mathbf{z}_t^{(2)}$, we have $p(\mathbf{x}_{m,B}|\mathbf{z}_t^{(1)}) \neq p(\mathbf{x}_{m,B}|\mathbf{z}_t^{(2)}, \mathbf{s})$.*

  iv *(Differentiability): There exists a functional $M$ such that $M\left[p_{\mathbf{x}_{m,B}|\mathbf{z}_m,\mathbf{s}}(\cdot \mid \mathbf{z}_m, \mathbf{s})\right] = h(\mathbf{z}_m, \mathbf{s})$ for all $\mathbf{z}_m \in \mathcal{Z}_m$ and $\mathbf{s} \in \mathcal{S}$, where $h$ is differentiable.*

*Then we have $[\hat{\mathbf{z}}_m, \hat{\mathbf{s}}] = h(\mathbf{z}_m, \mathbf{s})$, where $h$ is an invertible and differentiable function.*

**Discussions.** Assumption *i* is a moderate condition that ensures the probability distributions are well-defined and computable. Assumption *ii* informally requires that distinct input distributions correspond to distinct output distributions. In a similar spirit, Assumption *iii* guarantees that different values of $\mathbf{z}_m$ induce different conditional distributions $p(\mathbf{x}_{m,B} \mid \mathbf{z}_m)$, e.g., heteroskedastic noise. Notably, this condition is significantly weaker than monotonicity. Finally, Assumption *iv* imposes the differentiability of the mapping from $[\mathbf{z}_m, \mathbf{s}]$ to $p_{\mathbf{x}_{m,B}|\mathbf{z}_m,\mathbf{s}}$, which can be explicitly enforced through the use of differentiable models, i.e., variational autoencoders (VAEs).

**Theorem 2.** *(Identifiability of Shared Subspace) Suppose assumptions are hold true for all the modality and the whole latent space, and we further assume*

  i *(Entropy Regularization): $\hat{g}_{\mathbf{x}_m}^{-1}$ represent a set of shared latent variable encoders that minimizes $\sum_{k \in [M]} H\left(\hat{g}_{\mathbf{x}_k}^{-1}(\mathbf{x}_k)\right)$.*

*Then we have the $\hat{\mathbf{s}} = h_s(\mathbf{s})$, where $h_s$ is an invertible function.*

**Discussions.** After identifying the entire latent space and each latent subspace underlying modality observations: $\{[\mathbf{z}_1, \mathbf{s}] \ldots, [\mathbf{z}_M, \mathbf{s}]\}$, the shared component $\mathbf{s}$ can be isolated by leveraging the preliminary result that each $\mathbf{s}$ is block-wise identifiable. This enables the application of existing techniques for isolating shared latent spaces, e.g., as developed in (Yao et al., 2023; Von Kügelgen et al., 2021).

**Theorem 3.** *(Component-wise Identifiability) Suppose the assumptions (a lot abuse) in Theorem 1, Theorem 2 are satisfied, suppose we have*

   i *(Sufficient Variability):* Denote $|\mathcal{M}_{\mathbf{z}_m}|$ as the number of edges in Markov network $\mathcal{M}_{\mathbf{z}_m}$. Let

$$
\begin{aligned}
w(m) = & \Big( \frac{\partial^3 \log p(\mathbf{z}_m|\mathbf{s})}{\partial z_{m,1}^2 \partial s_{d_s}}, \cdots, \frac{\partial^3 \log p(\mathbf{z}_m|\mathbf{s})}{\partial z_{m,d_m}^2 \partial s_{d_s}} \Big) \oplus \\
& \Big( \frac{\partial^2 \log p(\mathbf{z}_m|\mathbf{s})}{\partial z_{m,1} \partial s_{d_s}}, \cdots, \frac{\partial^2 \log p(\mathbf{z}_m|\mathbf{s})}{\partial z_{m,d_m} \partial s_{d_s}} \Big) \oplus \Big( \frac{\partial^3 \log p(\mathbf{z}_m|\mathbf{s})}{\partial c_{t,i} \partial c_{t,j} \partial s_{d_s}} \Big)_{(i,j) \in \mathcal{E}(\mathcal{M}_{\mathbf{z}_m})},
\end{aligned} \tag{3}
$$

   *where $\oplus$ denotes concatenation operation and $(i,j) \in \mathcal{E}(\mathcal{M}_{\mathbf{z}_m})$ denotes all pairwise indice such that $z_{m,i}, z_{m,j}$ are adjacent in $\mathcal{M}_{\mathbf{z}_m}$. For $m \in [1, \cdots, n]$, there exist $4n + |\mathcal{M}_{\mathbf{z}_m}|$ different values of $\mathbf{s}_{d_s}$, such that the $4n + |\mathcal{M}_{\mathbf{z}_m}|$ values of vector functions $w(m)$ are linearly independent.*

   ii *(Sparsity Regularization): Let $\mathbf{G} \in \{0,1\}^{d_z \times d_z}$ denote the true adjacency matrix of the latent causal graph, and $\hat{\mathbf{G}} \in \{0,1\}^{d_z \times d_z}$ be the estimated adjacency matrix. We assume that the estimated graph is at most as dense as the true graph:*

$$
\|\hat{\mathbf{G}}\|_0 \leq \|\mathbf{G}\|_0,
$$

   *where $\|\cdot\|_0$ denotes the elementwise $\ell_0$ norm, i.e., the number of nonzero entries.*

*Then we have $\hat{\mathbf{z}}_{m,i} = h_i(\mathbf{z}_{m,\pi(j)})$, where $h_i$ is an invertible and differentiable function.*

**Discussions.** The core idea is to exploit the rich multi-modal information present in behavior trait datasets to disentangle shared latent variables from modality-specific ones. Shared latent factors, e.g., genetic traits, act as confounders and induce sufficient variability across modality-specific components. This motivates the adoption of nonlinear ICA (Hyvarinen et al., 2019). To achieve identifiability, we impose structural constraints derived from the ground-truth Markov network $\mathcal{M}_{\mathbf{z}_m}$ onto the estimated network $\mathcal{M}_{\hat{\mathbf{z}}_m}$, leveraging the connection between conditional independence and vanishing cross-partial derivatives (Lin, 1997): if $z_{m,i} \perp z_{m,j} \mid \{\mathbf{s}, \mathbf{z} \setminus \{z_{m,i}, z_{m,j}\}\}$, then $\frac{\partial^2 \log p(\mathbf{z}_m)}{\partial z_{m,i} \partial z_{m,j}} = 0$. These established conditions shed light on the design of training network below.

## 4.3 NETWORK TRAINING

**Embedding extraction.** For images, we adopt `ImageBind` (Girdhar et al., 2023) to obtain 1024-dimensional embeddings, leveraging its strength in multimodal representation learning. For text, we use `gte-Qwen2-7B-instruct` (Bai et al., 2023), a foundation model optimized for long-sentence embeddings, yielding 3584-dimensional vectors. Importantly, the released dataset does not contain raw images or text. Instead, all images and texts are converted into embeddings and further transformed through an additional invertible transformation, ensuring privacy while preserving utility.

**Encoders and decoders.** Each modality has its own encoder, which estimates modality-specific latents $\hat{z}_m$, exogenous variables $\hat{\eta}_m$, and shared latents $s$. To maintain conditional independence across measurements, decoders reconstruct each observation separately. This reconstruction is optimized via mean squared error: $\mathcal{L}_{\text{Recon}} = \sum_m \sum_k \|x_{m,k} - \hat{x}_{m,k}\|_2^2$.

**Independence constraints.** To enforce theoretical assumptions, we require independence among latents and exogenous variables. This is implemented by aligning their joint distribution $\hat{\gamma}$ with an isotropic Gaussian prior using KL divergence: $\mathcal{L}_{\text{Ind}} = \text{KL}(p(\gamma) \| \mathcal{N}(0, I))$.

**Sparsity regularization.** Causal relations are captured through a learnable adjacency matrix $\hat{A}$, implemented via normalizing flows (Papamakarios et al., 2021). A sparsity penalty encourages minimal yet sufficient causal graphs: $\mathcal{L}_{\text{Sp}} = \|\hat{A}\|_1$.

**Final objective.** The overall training loss combines all three components:

$$
\mathcal{L} = \alpha_{\text{Recon}} \mathcal{L}_{\text{Recon}} + \alpha_{\text{Ind}} \mathcal{L}_{\text{Ind}} + \alpha_{\text{Sp}} \mathcal{L}_{\text{Sp}}.
$$

This framework enforces reconstruction fidelity, independence constraints, and causal sparsity simultaneously, enabling the recovery of identifiable and interpretable latent variables across modalities. More details about the loss functions and network design are shown in App. A7.

### 4.4 SYNTHETIC EXPERIMENTS ON VARIANT MNIST

We first evaluate our method on synthetic data derived from MNIST (LeCun, 1998), benchmarking against state-of-the-art baselines including BetaVAE (Higgins et al., 2017), Multimodal Contrastive Learning (MCL; (Daunhawer et al., 2023)), and Multimodal Causal Representation Learning (MM-CRL; (Sun et al., 2025)). We construct two modalities using Colored MNIST (Arjovsky et al., 2019) and Fashion MNIST (Xiao et al., 2017), where cross-modal causal dependencies are explicitly designed: the horizontal position of digits affects image transparency, which in turn causally influences the vertical placement of fashion items and consequently their grayscale intensity. Fig.5(a) illustrates some generated images, while Fig.5(b) shows the underlying causal graph. This setup provides a structured yet non-deterministic mapping across modalities. Refer to App. A8 for more details.

As shown in Fig. 5(c), our method achieves an $R^2$ of 0.96 and an MCC of 0.92, clearly outperforming MMCRL which reaches $R^2$ of 0.90 and MCC of 0.85, and also consistently surpassing both BetaVAE and MCL. These results highlight the advantage of explicitly modeling cross-modal causal dependencies with multiple measurements, a setting where existing approaches remain constrained.

### 4.5 REAL-WORLD TRAIT ANALYSIS ON PERSONAX

After validating our method on synthetic data with strong results, we next apply it to the PersonaX datasets, training networks to learn latent representations and then applying causal discovery to obtain a meaningful causal graph. Fig. 6 shows the causal graph obtained from AthlePersona, while results for CelebPersona are provided in App. A9. The discovered latents naturally group into three categories: shared latents ($S_k$), image-based latents ($Z_{1,k}$), and trait-based latents ($Z_{2,k}$). From AthlePersona, we identify two shared factors ($S_1, S_2$), five image-based latents ($Z_{1,1}$–$Z_{1,5}$), and five trait-based latents ($Z_{2,1}$–$Z_{2,5}$). Importantly, the estimated latents may correlate with, but are not necessarily identical to, the well-defined Big Five traits. After obtaining the causal graph, we assign each variable a concrete interpretation, guided by the independence test results reported in Tab. A7, to facilitate clearer analysis. The undirected edge between $S_1$ and $S_2$ suggests a bidirectional relation between *mindset* and *culture*. *mindset* ($S_1$) influences *self-awareness* ($Z_{2,4}$). Moreover, cross-modal links reveal that *confidence* ($Z_{2,1}$) affects *facial expressions* ($Z_{1,4}$), and *emotional stability* ($Z_{2,3}$) impacts *grooming* ($Z_{1,2}$). A sequential pathway emerges among image-based latents: $Z_{1,1}$ (skin tone) $\to Z_{1,3}$ (attractiveness) $\to Z_{1,4}$ (facial expressions), highlighting appearance factors in athletes.

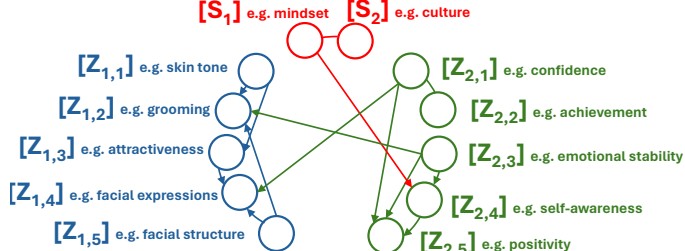

Figure 6: The causal graph with latent variables learned from AthlePersona dataset. Red, blue, and green nodes correspond to shared latents, image latents, and behavior trait latents.

Together, these findings validate our framework: synthetic experiments confirm identifiability and quantitative performance, while real-world analysis demonstrates its ability to uncover meaningful and interpretable cross-modal causal structures in human trait data. See App. A9 for more analysis.

## 5 DISCUSSIONS AND CONCLUSION

**Discussions.** (*i*) *Cohort-specific scope*: AthlePersona currently covers only male athletes, while CelebPersona only includes wealthy and high-visibility celebrities. These cohorts are not universally representative, but they provide controlled populations for analysis across different domains such as sports and entertainment. Looking ahead, we will expand PersonaX by continuously collecting and incorporating data from additional sources, enabling broader coverage and greater inclusivity over time. Specifically, we will introduce a version-controlled update framework with transparent

changelogs to ensure stability, inclusivity, and responsible evolution across dataset versions. (*ii*) *Lack of temporal stability*: Behavioral traits are subjective and dynamic, yet our traits are inferred by LLMs from static public data without longitudinal tracking. This complicates validation but points to future work on temporally rich datasets. We will further borrow the merits of recent advances in lightweight temporal modeling and semantic understanding Ye et al. (2025); Zhang et al. (2025), leveraging their advancements to rigorously evaluate and propel progress in this direction. See App. A1 for more discussions. (*iii*) *Multimodal CRL benchmarking*: Beyond supporting multimodal causal representation learning, $\mathbb{PersonaX}$ has the potential to serve as a dedicated benchmark for multimodal CRL, complementing existing resources such as CausalVerse Chen et al. (2025). As future work, we plan to develop a standardized benchmark suite built upon $\mathbb{PersonaX}$, featuring unified evaluation metrics for identifiability, cross-modal invariance, and causal graph recovery, thereby enabling fair comparison and reproducible evaluation of multimodal CRL methods.

**Conclusion.** We presented $\mathbb{PersonaX}$, two multimodal datasets linking LLM-inferred behavioral traits with facial and biographical information. Our two-level analysis pipeline combines structured dependence tests with unstructured causal representation learning, addressing both theoretical and empirical aspects: theoretically, we propose a novel identifiability theory tailored for multimodal, multi-measurement CRL; empirically, we demonstrate population-specific patterns and interpretable latent structures. These resources provide a foundation for studying invariant causal mechanisms of human behavioral traits while promoting diversity, equality, and mutual respect for all human beings.

## ACKNOWLEDGEMENTS

We would also like to acknowledge the support from NSF Award No. 2229881, AI Institute for Societal Decision Making (AI-SDM), the National Institutes of Health (NIH) under Contract R01HL159805, and grants from Quris AI, Florin Court Capital, MBZUAI-WIS Joint Program, and the Al Deira Causal Education project.

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

*Appendix for*

## "**PersonaX**: Multimodal Datasets with LLM-Inferred Behavior Traits"
Table of Contents:

## A1 Ethics Statements and Broader Impacts

Understanding human behavioral traits has broad implications for psychology, human–computer interaction, and AI personalization. By releasing two multimodal, publicly accessible datasets (`CelebPersona` and `AthlePersona`) together with a two-level causal analysis framework, this work provides the community with a resource to systematically investigate behavioral traits in relation to facial and biographical features, and to advance methodological research in multimodal CRL.

The ethical considerations and limitations must be acknowledged. We recognize that inferring behavioral traits from public data carries risks of reinforcing stereotypes or enabling misuse in sensitive domains (e.g., hiring, lending, surveillance). To mitigate these risks, (i) all data is sourced from consent-based, legally accessible platforms; (ii) no raw images or texts are released—only transformed embeddings with additional obfuscation; and (iii) we enforce strict non-commercial usage restrictions, accompanied by a detailed guideline file (`USAGE_GUIDELINES.md`). These safeguards are intended to reduce the likelihood of misuse while maintaining research utility.

The current release is demographically limited: `AthlePersona` covers only male athletes, while `CelebPersona` focuses on high-visibility celebrities. These choices were intentional: male professional leagues attract broader and more consistent public attention with accessible records, and the celebrity dataset builds upon the established CelebA benchmark. In both cases, we deliberately selected cohorts with sufficient public visibility to ensure the availability of high-quality data for reliable LLM inference for behavior traits. As such, results should not be interpreted as universally representative, but rather as population-specific analyses across complementary domains (sports and entertainment). We explicitly view broader demographic coverage (e.g., female athletes, less visible public figures, longitudinal data) as a crucial direction for subsequent dataset extensions.

Our overarching goal is to foster understanding of population-level patterns, not deterministic inference about individuals. We actively discourage applications in high-stakes decision-making, and instead encourage the community to use PersonaX for methodological development (e.g., causal discovery under selection bias, multimodal integration) and for examining fairness and robustness across social contexts. We also plan to update the dataset iteratively in response to community feedback, with inclusivity and transparency as guiding principles.

Overall, this work aims to advance the scientific study of behavioral traits while foregrounding fairness, privacy, and ethical responsibility. By articulating clear limitations, safeguards, and future commitments, we hope to enable constructive research and minimize risks of harmful deployment.

## A2 Details about Related Work

### A2.1 Human Behavior Trait Analysis

Human behavior traits have long been central to understanding how individuals reflect on their strengths, limitations, and interpersonal tendencies (Rothe, 2017; Johnson, 1997; DeNeve & Cooper, 1998; Briggs & Cheek, 1986). In contrast to psychological personality, which emphasizes internal dispositions typically assessed through self-reports or expert evaluation, behavior traits focus on outwardly observable patterns inferred from language, facial appearance, physiological states, and digital traces. Traditional self-report instruments, such as Cattell's 16PF (Cattell et al., 1970), the Eysenck Personality Questionnaire (Eysenck & Eysenck, 1975), and the Myers–Briggs Type Indicator (Myers et al., 1998), provided early frameworks for personality assessment. The Big Five (Goldberg, 1993) has since emerged as the prevailing paradigm, supported by strong empirical evidence and predictive power (Cobb-Clark & Schurer, 2012; Oshio et al., 2018; Komarraju et al., 2011; Roccas et al., 2002; Gerber et al., 2011). However, these methods remain vulnerable to self-report biases and limited scalability.

Computational approaches have sought to infer traits from observable signals rather than introspection. Examples include linguistic cues (Pennebaker et al., 1999), handwriting (Asra & Shubhangi, 2015),

| Dataset | Focus | Score | Text Desc. | Modalities | Task / Focus |
|---|---|---|---|---|---|
| myPersonality (Kosinski et al., 2015) | Inner | ✓ | ✗ | Text, Tabular | Personality Prediction |
| OCEAN (Park et al., 2015) | Inner | ✓ | ✗ | Text, Tabular | Psychometric Analysis |
| MuPTA (Ryumina et al., 2023) | Inner | ✓ | ✗ | Video, Audio, Tabular | Personality Prediction |
| MDPE (Cai et al., 2024) | Inner | ✓ | ✗ | Video, Audio, Tabular | Deception Detection |
| Amigos (Miranda-Correa et al., 2018) | Inner | ✓ | ✗ | Video, Audio, Sensor, Tabular | Affect, Personality and Mood Prediction |
| SALSA (Alameda-Pineda et al., 2015) | Inner | ✓ | ✗ | Video, Audio, Sensor, Tabular | Group Behavior and Personality |
| BPAC (Dotti et al., 2018) | Inner | ✓ | ✗ | Video, Tabular | Behavior Understanding and Personality Recognition |
| Driving (Evin et al., 2022) | Inner | ✓ | ✗ | Sensor, Tabular | Personality Prediction from Driving |
| Lifelog (Chung et al., 2022) | Outer | ✓ | ✗ | Sensor, Tabular | Real-world Behavior Study |
| YouTube-Vlogs (Biel & Gatica-Perez, 2012) | Outer | ✓ | ✗ | Video, Audio, Tabular | Personality Prediction |
| FI-V2 (Escalante et al., 2020) | Outer | ✓ | ✗ | Video, Audio, Text | First-Impression Recognition |
| CelebA (Liu et al., 2015) | - | ✗ | ✗ | Image | Facial Attribute Analysis |
| **Persona**X **(Ours)** | Outer | ✓ | ✓ | Image, Text, Tabular | Behavior Trait Interpretation & Causal Analysis |

Table A1: Comparison of multimodal datasets for personality or behavior-trait research. *Focus* distinguishes between inner personality (psychological, self-reported) and outer behavior traits (observable signals). *Score* indicates whether personality or trait scores are included. *Text Desc.* shows whether textual trait descriptions are available. *Modalities* lists the input data types. *Task / Focus* describes the primary research application. Unlike prior datasets, PersonaX uniquely combines multimodal signals with both scores and textual descriptions, supporting systematic cross-modal and causal analyses.

speech (Mohammadi et al., 2010), facial expressions (Güçlütürk et al., 2018), and online profiles (Youyou et al., 2015; Celli et al., 2014). A landmark study by (Kosinski et al., 2013) showed that Facebook Likes could predict a wide range of sensitive traits, including personality and demographics, with accuracies comparable to psychometric tests. Beyond digital traces, physiological and behavioral signals have been linked to traits in contexts such as driving (Evin et al., 2022), smart-home daily activities (Dotti et al., 2018), and long-term lifelogging (Chung et al., 2022). These works demonstrate the potential of behavioral data for trait inference, often in settings where traditional questionnaires are impractical.

Several multimodal datasets have been developed to study traits in richer contexts. SALSA (Alameda-Pineda et al., 2015) captures group behavior at social events through multimodal recordings with personality annotations. YouTube-Vlogs (Biel & Gatica-Perez, 2012), FI-V2 (Escalante et al., 2020), MuPTA (Ryumina et al., 2023), MDPE (Cai et al., 2024), and Amigos (Miranda-Correa et al., 2018) integrate video, audio, and physiological data for tasks such as impression analysis or affect recognition. While valuable, these datasets are generally small-scale and lack explicit textual trait descriptions or unified frameworks for cross-modal analysis. Other resources like CelebA (Liu et al., 2015), FFHQ (Karras et al., 2019), and FairFace (Kärkkäinen & Joo, 2021) enable large-scale analysis of facial attributes but do not provide trait or personality annotations.

Complementing dataset development, empirical studies have examined how outward features in one modality can signal traits in another. The "kernel of truth" hypothesis (Zebrowitz, 2018) suggests that physical cues may reflect behavioral tendencies, supported by findings linking facial morphology to health (Kramer & Ward, 2010), aggression (Carré & McCormick, 2008), and personality judgments from body images (Naumann et al., 2009). More recently, machine learning methods have shown that Big Five traits can be predicted from facial behavior (Cai & Liu, 2022; Youyou et al., 2015; Kachur et al., 2020). Beyond vision, correlations have also been observed with activity levels (Wilson & Dishman, 2015), sensor data (Dotti et al., 2018), and physiological signals (Gao et al., 2019).

Despite these advances, existing resources remain fragmented: many rely on self-reports, others are constrained to controlled laboratory settings, and most lack integration of textual, visual, and biographical information in a unified framework. Our contribution addresses this gap by introducing PersonaX, which provides two multimodal datasets, CelebPersona and AthlePersona, linking facial, physical, and occupational features with LLM-inferred Big Five behavior traits. This enables systematic exploration of cross-modal relationships and supports both predictive and causal analyses at scale.

## A2.2 Causal Discovery and Causal Representation Learning

Causal discovery (Spirtes et al., 2001) from observational data has attracted considerable attention in recent decades. Constraint-based and score-based methods are two primary categories in causal discovery. Constraint-based methods, such as PC (Spirtes & Glymour, 1991) and FCI (Spirtes et al.,

1995), leverage conditional independence tests (CIT; (Zhang et al., 2012; Strobl et al., 2019; Gretton et al., 2005; Tallarida et al., 1987; Tsamardinos et al., 2006)) to estimate the graph skeleton and then determine the orientation. For score-based methods, the approach can vary based on the search strategy, which may involve greedy search, exact search, or continuous optimization. One typical score-based method with greedy search is Greedy Equivalent Search (GES) (Chickering, 2002). The exact score-based methods are often time-consuming, such as dynamic programming (DP) (Koivisto & Sood, 2004), A* (Yuan et al., 2011), and integer programming (Cussens, 2011; Li & Wu, 2022). NOTEARS (Zheng et al., 2018) is the first work to cast the Bayesian network structure learning task into a continuous constrained optimization problem (Li et al., 2022; 2024d) with the least squares objective. Subsequent work GOLEM (Ng et al., 2020) adopts a continuous unconstrained optimization formulation with a likelihood-based objective. A line of works have extended NOTEARS to handle nonlinear cases via deep neural networks, such as DAG-GNN (Yu et al., 2019) and DAG-NoCurl (Yu et al., 2021). Some methods are developed to improve the computational efficiency, e.g., $\psi$DAG (Ziu et al., 2024). In recently years, there are activa researches on causal discovery from various data constraints, including distributed data (Li et al., 2024c), heterogeneous data (Huang et al., 2020), deterministic relations (Li et al., 2024b), latent confounder and selection bias (Luo et al., 2025a;b), and etc.

Causal representation learning (CRL) aims to recover high-level causal variables from low-level observations, bridging machine learning and causal inference (Schölkopf et al., 2021; Li et al., 2025a; Fan et al., 2024). It generalizes classical causal discovery (Spirtes et al., 2001) by learning structured representations that respect causal semantics. CRL methods with identifiability guarantees typically rely on additional assumptions: (1) *Functional constraints* on the data-generating process (Xu et al., 2024; Zheng et al., 2022); (2) *Interventional or multi-environment data* that introduce distributional shifts to expose latent structure (Hyvarinen et al., 2019; Khemakhem et al., 2020); (3) *Multimodal or multiview settings*, where aligned observations across modalities help identify shared causal factors through sample-level invariance (Yao et al., 2023; Daunhawer et al., 2023; Sturma et al., 2023). Recent studies unify these approaches under general invariance principles, showing that many can be seen as special cases of a broader framework (Ahuja et al., 2023).

Parallel to these theoretical advances, some works focus on practical CRL applications without strict identifiability. Examples include variational methods for biomedical data (Mao et al., 2022), contrastive learning for multimodal causal analysis (Zheng et al., 2024), learning causal insights from OpenReview systems (Li et al.), and causal discovery on neuroimaging datasets (Rawls et al., 2021). In contrast, our work aims to combine both theory and application: we derive formal identifiability conditions under multi-modality multi-measurement settings and integrate these insights into a practical estimation framework for human trait analysis.

## A2.3 MULTIMODALITY AND REPRESENTATION LEARNING

In the context of broader machine learning, multimodal representation learning focuses on integrating information from multiple modalities, such as text, images, and audio, to learn unified representations for downstream tasks (Manzoor et al., 2023; Zhang et al., 2020). Among these methods, contrastive learning has emerged as a powerful approach, particularly for weakly supervised settings, due to its scalability and effectiveness (Daunhawer et al., 2023; Wang et al., 2022; Peng et al., 2022; Radford et al., 2021; Khosla et al., 2020; Oord et al., 2018). A prominent example is CLIP model (Radford et al., 2021), which aligns text and image embeddings through contrastive objectives (Sun et al., 2024; Lin et al., 2022; Girdhar et al., 2023).

Unlike these methods that primarily aim for discriminative or generative performance, our work is centered on uncovering the underlying causal structure shared across modalities, with the specific goal of generating insights into personalities through principled causal representations.

## A2.4 LLM REASONING AND INFERENCE

Large Language Models (LLMs) such as GPT-4 (OpenAI, 2023), PaLM (Chowdhery et al., 2022), Gemini (Team et al., 2023), DeepSeek (Liu et al., 2024), Qwen (Bai et al., 2023), and Claude (Anthropic, 2023) have demonstrated remarkable reasoning and inference capabilities across a wide range of tasks, including arithmetic (Cobbe et al., 2021), commonsense reasoning (Srivastava et al., 2022), text editing and generation (Tang et al., 2025), code generation (Chen et al., 2021), and scientific

**Prompt 1.** *(Complete Prompt for Inferring Behavior Traits in AthlePersona and CelebPersona)*

*Task Description*

*Analyze the Big Five behavior traits of the individual described below. Base the analysis on publicly available information, such as direct quotes from interviews, observed public behavior, documented career patterns, and biographical details. Avoid speculation or information from unreliable gossip sources. The analysis should reflect the public persona, not a definitive psychological diagnosis or clinical evaluation.*

*Individual Information*

- *Name:* {*name*}
- *Gender:* {*gender*}
- *Description:* {*league*} *player, from* {*country*} *(AthlePersona) —* {*occupation*}*, from* {*country*} *(CelebPersona)*

*Instructions*

1. *For each of the five Big Five behavior traits (Openness, Conscientiousness, Extraversion, Agreeableness, Neuroticism):*
   - ***Analysis:*** *Provide a concise (1–2 sentences) analysis. If there is sufficient public information, identify specific examples of behaviors, statements, or patterns and explain how they relate to the definition of the trait. If there is insufficient information, state that clearly.*
   - ***Score:*** *Assign a score from 0 to 3 based on the scale below.*
   - ***Justification:*** *Provide a brief (1 sentence) justification for the score, directly referencing the evidence mentioned in the analysis or the lack thereof.*
2. ***Summary:*** *After analyzing all five traits, provide a summary string containing the five scores separated by hyphens.*
3. ***Anonymity:*** *Do not explicitly mention the name of the individual in the output, use pronouns {He/His or She/Her} instead.*
4. ***Distinguishing Scores:*** *When analyzing each trait, carefully consider whether the information is insufficient (Score 0) or if it's present but indecisive (Score 2). If the available information is too sparse or vague to form any meaningful analysis, assign Score 0. If there is sufficient information but it leads to an indecisive conclusion, assign Score 2.*
5. ***Strict Formatting:*** *Adhere EXACTLY to the "Expected Output Format" template below, including line breaks. Do not add any introductory or concluding remarks outside this structure.*

*Scoring Scale*

- ***0 = Insufficient information*** *– Not enough reliable public information to assess the trait. The trait's presence or absence is unknown or unclear due to lack of data.*
- ***1 = Disagree*** *– Clear evidences contradict the trait*
- ***2 = Neutral*** *– Evidence is mixed or the trait is not prominent. There is enough information, but it does not strongly support or contradict the trait.*
- ***3 = Agree*** *– Clear evidences support the trait*

*Expected Output Format*

   ***Openness:***
      *- Analysis: [Analysis]*
      *- Score: [0–3]*
      *- Justification: [Justification]*
   ***Conscientiousness:***
      *- Analysis: [Analysis]*
      *- Score: [0–3]*
      *- Justification: [Justification]*
   ***Extraversion:***
      *- Analysis: [Analysis]*
      *- Score: [0–3]*
      *- Justification: [Justification]*
   ***Agreeableness:***
      *- Analysis: [Analysis]*
      *- Score: [0–3]*
      *- Justification: [Justification]*
   ***Neuroticism:***
      *- Analysis: [Analysis]*
      *- Score: [0–3]*
      *- Justification: [Justification]*
   ***Summary: [ScoreO-ScoreC-ScoreE-ScoreA-ScoreN]***

QA (Wang et al., 2023). These models perform zero-shot or few-shot reasoning using techniques such as chain-of-thought prompting (Wei et al., 2022), self-consistency (Wang et al., 2023), active prompting (Diao et al., 2023), confidence-based If-or-Else prompting (Li et al., 2024a), reflective reasoning (Du et al., 2025) and tool-augmented reasoning (Paranjape et al., 2023). Despite their black-box nature, LLMs have shown emergent abilities to perform structured reasoning without explicit supervision, making them powerful general-purpose inference engines.

Table A2: Full Table of Features and Descriptions for `AthlePersona`.

| AthlePersona Dataset | | | |
|---|---|---|---|
| **Feature** | **Type** | **Description** | **Missing Rate (%)** |
| Id | string | Unique identifier for each athlete | 0 |
| Height | float32 | Height in centimeters | 0 |
| Weight | float32 | Weight in kilograms | 0 |
| Birthyear | int32 | Year of birth | 0 |
| Birthmonth | int32 | Month of birth | 0 |
| Birthday | int32 | Day of birth | 0 |
| League | string | Name of the athlete's league | 0 |
| Latitude | float32 | Latitude of country's central location, transformed from the nationality | 0 |
| Longitude | float32 | Longitude of country's central location | 0 |
| Chatgpt_output | string | Full trait analysis by ChatGPT encoded in embeddings | 0 |
| Gemini_output | string | Full trait analysis by Gemini encoded in embeddings | 0 |
| Llama_output | string | Full trait analysis by LLaMA encoded in embeddings | 0 |
| Chatgpt_o to Chatgpt_n | int32 | Big Five scores (OCEAN) by ChatGPT | 0 |
| Gemini_o to Gemini_n | int32 | Big Five scores (OCEAN) by Gemini | 0 |
| Llama_o to Llama_n | int32 | Big Five scores (OCEAN) by LLaMA | 0 |
| Final_o to Final_n | int32 | Final aggregate scores for Big Five traits | 0 |
| Image_1 | image | First facial image embeddings of the athlete | 0 |

Recent research has explored the extent to which Large Language Models (LLMs) can infer, simulate, and even express human personality traits. For example, (Peters et al., 2024) demonstrate that models such as GPT-4 can estimate Big Five personality dimensions from user-generated text with moderate accuracy, even in zero-shot settings. Similarly, (Serapio-García et al., 2023) show that LLMs can produce consistent personality profiles when prompted, often aligning with outputs from standardized psychometric assessments. Beyond inference, other studies examine how LLMs naturally exhibit personality-like traits in their responses. (Jiang et al., 2023) introduce methods to control and elicit desired personality traits in language model outputs, while (Wang et al., 2025) analyze the emergent ability of LLMs to emulate distinct personality patterns during generation. Furthermore, recent works such as (Tiuleneva et al., 2024; Rao et al., 2023) utilize LLMs to annotate or assess personality traits from textual data, illustrating their growing role in computational personality research.

Our work addresses these limitations by providing multimodal datasets that unite visual, physical, demographic, and personality dimensions, with multiple model-generated assessments that enable systematic evaluation.

## A3 DETAILS ABOUT ATHLEPERSONA AND CELEBPERSONA DATASETS

### A3.1 FULL FEATURE LISTS

The full feature tables of `AthlePersona` and `CelebPersona` are displayed in Table A2 and A3. The complete final prompt for generating personality is shown in Prompt 1, where the blue text is the highlighted information for each individual. Summary of terms of use compliance for all different sports leagues are in Table A4.

The CelebPersona dataset contains structured information about public figures, combining demographic attributes, facial characteristics, and personality assessments. Key features include basic physical attributes (height, weight), birth details (day, month, year), and location information (latitude and longitude, transformed from their nationality). In addition, each celebrity is assigned a categorical occupation and gender label. Rich personality data is captured in the form of full-text analyses generated by ChatGPT (ChatGPT-4o-latest (2025-03-26)), Gemini (Gemini-2.5-Pro-Exp-03-25),

Table A3: Full Table of Features and Descriptions for `CelebPersona`.

| Feature | Type | Description | Missing Rate (%) |
|---|---|---|---|
| **CelebPersona Dataset** | | | |
| Id | string | Unique identifier for each celebrity | 0 |
| Height | float32 | Height in centimeters | 71.5 |
| Weight | float32 | Weight in kilograms | 87.0 |
| Birthday | int32 | Day of birth | 2.0 |
| Birthmonth | int32 | Month of birth | 2.0 |
| Birthyear | int32 | Year of birth | 0.6 |
| Latitude | float32 | Latitude of country's central location | 0.2 |
| Longitude | float32 | Longitude of of country's central location | 0.2 |
| Occupation_Num | int32 | Occupation category:
0 = Entertainment & Performing Arts
1 = Music
2 = Sports
3 = Media & Film Production
4 = Business & Finance
5 = Academia & Science
6 = Healthcare
7 = Legal & Government
8 = Arts & Culture
9 = Religion & Service
10 = Aviation & Space
11 = Other | 0 |
| Gender_Num | int32 | Gender:
1 = Male
2 = Female | 0.2 |
| Chatgpt_output | string | Full trait write-up by ChatGPT encoded in embeddings | 0 |
| Gemini_output | string | Full trait write-up by Gemini encoded in embeddings | 0 |
| Llama_output | string | Full trait write-up by LLaMA encoded in embeddings | 0 |
| Chatgpt_o to Chatgpt_n | int32 | Big Five scores (OCEAN) by ChatGPT:
0 = Unknown
1 = Disagree
2 = Neutral
3 = Agree | 0 |
| Gemini_o to Gemini_n | int32 | Big Five scores (OCEAN) by Gemini | 0 |
| Llama_o to Llama_n | int32 | Big Five scores (OCEAN) by LLaMA | 0 |
| Final_o to Final_n | int32 | Final aggregated scores for Big Five traits | 0 |
| Arched_Eyebrows | int32 | Binary facial feature:
-1 = Absent
0 = Unknown
1 = Present | 0 |
| Big_Nose | int32 | Binary facial feature | 0 |
| Pointy_Nose | int32 | Binary facial feature | 0 |
| Bushy_Eyebrows | int32 | Binary facial feature | 0 |
| Big_Lips | int32 | Binary facial feature | 0 |
| Oval_Face | int32 | Binary facial feature | 0 |
| Double_Chin | int32 | Binary facial feature | 0 |
| Receding_Hairline | int32 | Binary facial feature | 0 |
| Narrow_Eyes | int32 | Binary facial feature | 0 |
| High_Cheekbones | int32 | Binary facial feature | 0 |
| Image_1 to Image_35 | image | Up to 35 facial images embeddings per identity | - |

and LLaMA (Llama-4-Maverick-03-26-Experimental), along with their respective Big Five scores (Openness, Conscientiousness, Extraversion, Agreeableness, Neuroticism, OCEAN). We choose these three models because they outperform the other models in various dimensions as shown in our preliminary experiments in Section 2.1. In order to represent each celebrity with one set of OCEAN

Table A4: Summary of terms-of-use compliance for different sports leagues. The column *Sports League* lists the league under consideration, and *Official Website Reference* points to the section of the terms-of-use policy from the league's official website. *Original Statement* excerpts the relevant clause directly from the website. *Requires Consent?* indicates whether explicit written consent is required even for non-commercial academic research use in our case.

| Sports Leagues Terms of Use Compliance | | | |
|---|---|---|---|
| **Sports League** | **Official Website Reference** | **Original Statement** | **Requires Consent?** |
| NBA | Terms of Use → 9. NBA STATISTICS (nba) | "By using such NBA Statistics, you agree that: (i) any use, display, or publication of the NBA Statistics shall include a prominent attribution to NBA.com in connection with such use, display, or publication; (ii) the NBA Statistics may only be used, displayed, or published for legitimate news reporting or private, non-commercial purposes;..." | No |
| NFL | Terms and Conditions → 1. INTRODUCTION; GENERAL; OWNERSHIP; PROHIBITIONS (nfl) | "You may use the Services solely for your own individual non-commercial and informational purposes only. Any other use, including for any commercial purposes, is strictly prohibited without our express prior written consent." | No |
| MLB | Terms of Use Agreement (mlb) | "... you must not reproduce, prepare derivative works based upon, distribute, perform or display the MLB Digital Properties without first obtaining the written permission of MLB or otherwise as expressly set forth in the terms and conditions of the MLB Digital Properties. The MLB Digital Properties must not be used in any unauthorized manner." | Yes |
| NHL | Terms of Service → 7. Intellectual Property (nhl) | "You may access, use, and display the Services, but only for non-commercial, informational, personal use, without modification or alteration in any way, and only so long as you comply with these Terms." | No |
| Premier League | Terms of Use → Intellectual Property Rights (pre) | "You may download and print material from the Website or App as is reasonable for your own private and personal use. You may also forward such material from the Website or App to other people for their private and personal use provided you credit us as its source and add the Website address." | No |
| La Liga | Legal Notice and Conditions of Use → 3. Use of the Website(lal) | "... The User undertakes to refrain from (a) using the Contents in a manner... (b) reproduce or copy, distribute, allow public access through any form of public communication, adapt, transform or modify the Contents, unless authorised by the owner of the corresponding rights or it is legally permitted..." | Yes |
| Serie A | General Terms and Conditions of the License Agreement → 2. Right limitations → 2.2 (ii) Official Data(leg) | "Except in the case of a separate written agreement between Lega Serie A and the Licensee establishing otherwise, the Licensee may only exploit the data related to the Competitions, the Matches, the Clubs and the players in the context ..." | Yes |
| Bundesliga | Terms of Use Services → 8. Audiovisual Content (bun) | "The audiovisual content available within the Products is made available to the User for personal and non-commercial purposes only. The User is authorized to use this audiovisual content only for the purposes of information and entertainment in the private sphere for themselves and persons personally associated with them (e.g. family members, friends and acquaintances). Limited to these purposes, the DFL grants the User a non-exclusive, non-transferable, non-sub-licensable right of use to access and view the audiovisual content within the Products. With the exception of the aforementioned limited right of use, the User is not granted any rights to the audiovisual content." | No |
| Ligue 1 | Terms and Conditions of Use → 6. Intellectual Property (lig) | "... Any total or partial reproduction of the Site or its elements without prior written authorization from the publisher (LFP) may lead to legal proceedings against the infringers." | Yes |
| PGA Tour | Terms of Use → 7. Conduct(D) (pga) | "You may use real time scoring, statistics and other data (whether current or archival) collected from PGATOUR.COM solely for legitimate news reporting and for personal, non-commercial purposes. You shall not use real time scoring, statistics or other data (whether current or archival) collected from PGATOUR.COM for sale, license or other commercial purposes (including, without limitation, commercial gambling purposes), unless expressly licensed by the PGA TOUR Parties." | No |
| ATP Tour | Terms & Conditions → 7. PROHIBITED USES → A. Ownership (atp) | "ATP owns or has the right to use all of the data, information, text, images, streaming media, video, sounds, icons, scores, rankings, statistics, and other content contained on this Website (the "Content"), and the copyrights and other intellectual property rights therein, unless otherwise noted. You may print one copy of the Content of this Website for your own personal, non-commercial use." | No |

personality scores, we aggregate all those three sets of 5-dimensional personality scores generated by 3 LLMs via voting, and we label those aggregated features as "Final". Regarding the facial attributes, we manually selected 10 attributes (e.g., *Big Nose, High Cheekbones*) from the original 40 attributes in CelebA dataset, those selected features are most likely to present one's inherent property in appearance, and less likely to change over short time than the others (e.g., *Heavy Makeup, Wearing*

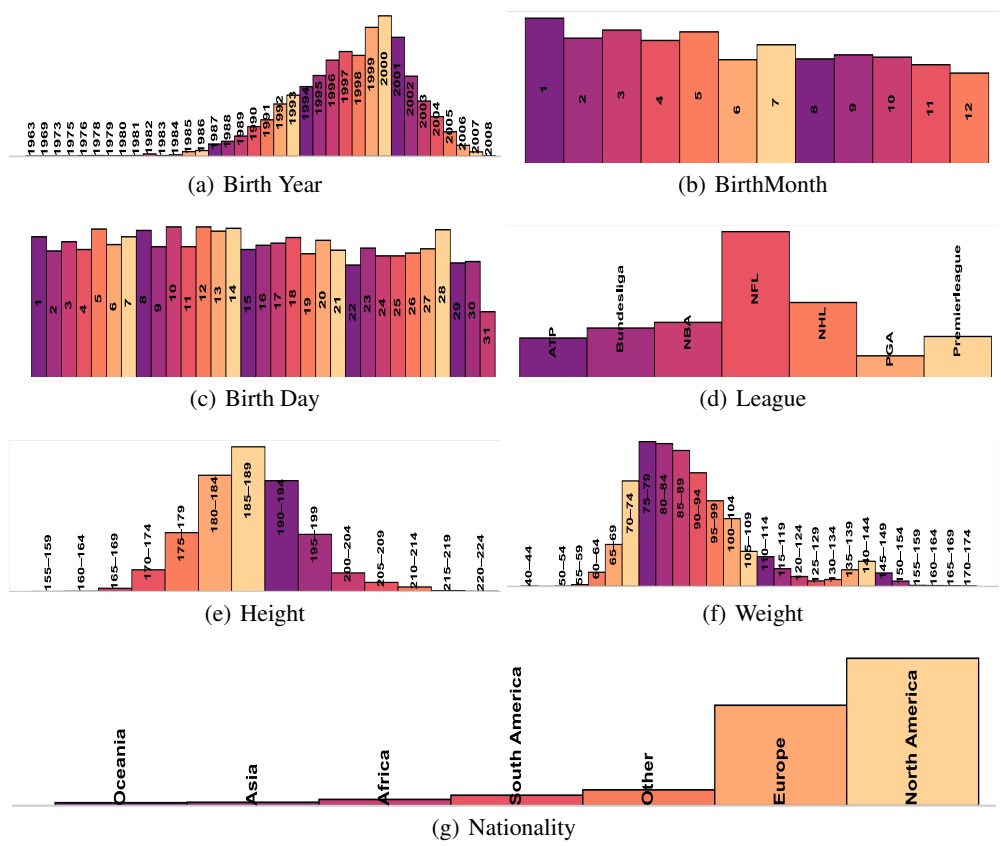

Figure A1: The distributions of the features in AthlePersona Dataset.

*Hat*). These binary facial attributes provide interpretable visual markers. Note that each image has a corresponding attribute value and there are multiple images per celebrity, we therefore aggregate all these attributes from different images by majority voting, to obtain the aggregated facial attributes. For each celebrity sample, there are at least two facial images taken from different angles, and up to 35 facial images per sample, referenced via relative file paths.

The AthlePersona dataset focuses on high-profile athletes and contains similar structure to CelebPersona, with emphasis on athletic context. It captures personal traits such as birth year, month, and day, physical measurements like height and weight, and the name of the athlete's league. Personality descriptions and Big Five scores are again generated by ChatGPT, Gemini, and LLaMA, with final aggregated trait scores summarizing the predictions. Unlike CelebPersona, this dataset includes only a single facial image per athlete but maintains key demographic and geographic metadata. It omits facial feature annotations and categorical occupation labels, instead reflecting the athletic domain through the league information.

### A3.2 DISTRIBUTION PLOTS

The Figure A1 shows the distribution of AthlePersona. The AthlePersona dataset is dominated by athletes born between 1985 and 2005, with a uniform spread across birth months and days. Most individuals are associated with NFL, NHL and NBA, showing a strong skew toward U.S. sports. Heights cluster around 180–199 cm, and weights around 90–109 kg, which aligns with physical norms for elite athletes in contact sports. Nationalities are overwhelmingly North American, with minimal representation from other continents, highlighting a clear Western and U.S.-centric dataset bias.

The Figure A2 shows the distribution of CelebPersona. The CelebPersona dataset predominantly features younger individuals, with birth years peaking between 1990–1999, and shows a balanced

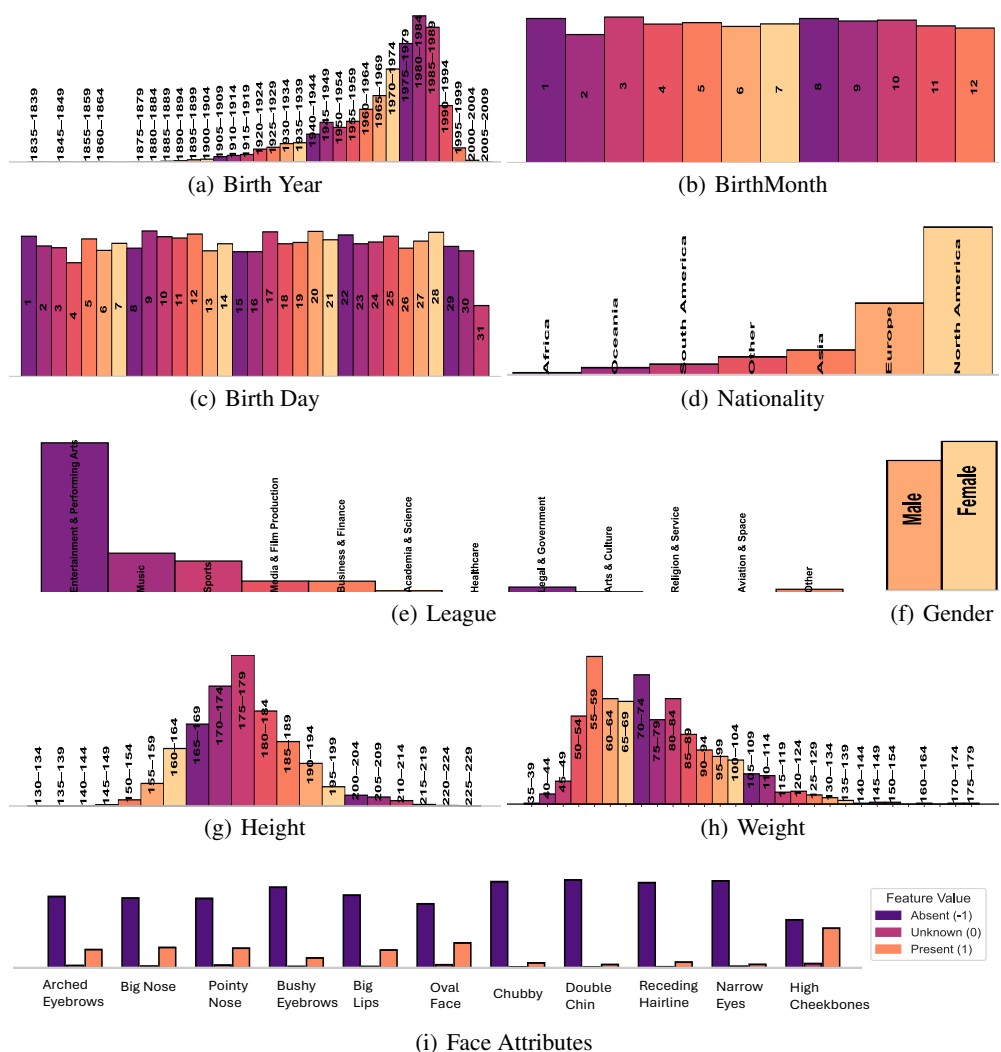

Figure A2: The distributions of the features in CelebPersona Dataset.

distribution across birth months and days. Most individuals are from North America and Europe, with underrepresentation from other continents. There is a notable occupational bias toward Entertainment, Music, and Sports, while fields like Healthcare and Academia are sparsely represented. Females slightly outnumber males. Height and weight distributions center around typical adult ranges, though outliers exist. The weight distribution peaks between 60–69 kg and 50–59 kg, with a sharp drop after 90 kg. The range 135–139 kg and higher has a minimal count. Regarding facial attributes, the majority of features are marked as absent, with a smaller subset present, particularly for traits like Oval Face and High Cheekbones. The unknown values are minimal.

### A3.3 Missing Values

As shown in the dataset features table (Table A3), the Missing Rate column indicates the proportion of unavailable or incomplete values for each feature. Despite efforts to retrieve missing information—particularly from publicly accessible sources like Wikipedia—certain attributes remain incomplete, especially those considered more private or less frequently disclosed. In the CelebPersona dataset, there are a total of 9444 data. Height and Weight have the highest number of missing entries, with 71.5% and 87% missing records respectively. Birthday and Birthmonth are missing in 2% entries each, while Birthyear is missing in 0.6% cases. Geographic coordinates (Latitude and Longitude) are

Table A5: AI Model Arena Scores and API Pricing recorded on April 10 2025.

| Model Name | Company | Arena Score | API Price (I/O) | Used by Us |
|---|---|---|---|---|
| Gemini-2.5-Pro-Exp-03-25 | Google | 1439 | $1.25/$10.00 | yes |
| Llama-4-Maverick-03-26-Experimental | Meta | 1417 | $5.00/$15.00 | yes |
| ChatGPT-4o-latest (2025-03-26) | OpenAI | 1410 | $2.50/$10.00 | yes |
| Grok-3-Preview-02-24 | xAI | 1403 | $3.00/$15.00 | yes |
| GPT-4.5-Preview | OpenAI | 1398 | $75.00/$150.00 | no |
| Gemini-2.0-Flash-Thinking-Exp-01-21 | Google | 1380 | $0.10/$0.40 | yes |
| Gemini-2.0-Pro-Exp-02-05 | Google | 1380 | $0.10/$0.40 | no |
| DeepSeek-V3-0324 | DeepSeek | 1369 | $0.07/$1.10 | yes |
| DeepSeek-R1 | DeepSeek | 1358 | $0.14/$2.19 | yes |
| Gemini-2.0-Flash-001 | Google | 1354 | $0.10/$0.40 | yes |
| Qwen2.5-Max | Alibaba | 1340 | $1.60/$6.40 | yes |
| QwQ-32B | Alibaba | 1315 | $0.29/$0.39 | yes |

absent in 0.2% instances, and categorical attributes such as Occupation_Num and Gender_Num have 0.05% and 0.2% missing values, respectively.

In contrast, the AthlePersona dataset (TableA2) has been fully cleaned by removing all rows that contain any missing values. Prior to finalization, any entry with incomplete demographic, geographic, or profile information was excluded to ensure consistency. As a result, AthlePersona contains no missing values, making it readily usable for downstream analysis without requiring additional preprocessing or imputation.

### A3.4 DETAILS ABOUT CONSENT: TERMS-OF-USE COMPLIANCE

A summary of terms-of-use compliance for the different sports leagues is provided in Tab. A4. In addition, we include below the core consent statements for the CelebA dataset and for WikiData.

- CelebA: (i) The CelebA dataset is available for non-commercial research purposes only. (ii) You agree not to reproduce, duplicate, copy, sell, trade, resell or exploit for any commercial purposes, any portion of the images and any portion of derived data. (iii) You agree not to further copy, publish or distribute any portion of the CelebA dataset. Except, for internal use at a single site within the same organization it is allowed to make copies of the dataset. (iv) The face identities are released upon request for research purposes only. Please contact us for details.

- Wikidata: Wikidata offers a wide range of general data about everything under the sun. All that data is licensed CC0, "No rights reserved", for the public domain.

### A4 DETAILS ABOUT LLM SELECTION AND PROMPT DESIGN

Throughout this paper, we rely on large language models (LLMs) to generate human personality. Fortunately, Benefiting from the recent explosion in the size and availability of LLMs (Achiam et al., 2023; Floridi & Chiriatti, 2020; Jiang et al., 2024; Liu et al., 2024; Bai et al., 2023), some research has shown that personality measurements in the outputs of some LLMs under specific prompting configurations are valid and reliable (Serapio-García et al., 2023; Jiang et al., 2023; Tseng et al., 2024). We built our datasets from a number of professional athletes and celebrities, based on the facts that they are famous and it is likely to have sufficient information about them online.

In Section 2.1, we present some experiments to show how to select LLMs for personality generation, and also how to design the prompts. In the following, we will demonstrate more details.

### A4.1 DETAILS ABOUT HOW TO SELECT LLMS (REGARDING TABLE 1)

**Model Choices in Table 1.** We present the comparative evaluations on 10 of the state-of-the-art LLMs on our two persona datasets. The 10 LLMs include: Gemini-2.5-Pro-Exp-03-25, Llama-4-Maverick-03-26-Experimental, ChatGPT-4o-latest (2025-03-26), Grok-3-Preview-02-24, Gemini-2.0-

Flash-Thinking-Exp-01-21, DeepSeek-V3-0324, DeepSeek-R1, Gemini-2.0-Flash-001, Qwen2.5-Max, and QwQ-32B. Initially, we choose the Top 10 models based on the Arena leaderboard (Chiang et al., 2024). To enhance diversity, we also included Qwen2.5-Max and QwQ-32B (Bai et al., 2023) from Alibaba, both noted for their strong reasoning capabilities. We list all those 12 LLMs and summarize them in Table A5, including the model name, the company name, the arena score, the API input and output price per million tokens, and whether it is used by us for further analysis in Table 1. Specifically, GPT-4.5-Preview (Achiam et al., 2023) was excluded due to prohibitively high API costs where the input and output API prices were $75 and $150 per million tokens, respectively. Gemini-2.0-Pro-Exp-02-05 (Team et al., 2023) was omitted due to inaccessibility, it was merged to the latest Gemini-2.5-Pro-Exp-03-25 model. Therefore, in the end, we only considered 10 LLMs, as shown in Table 1.

**Evaluation Metrics in Table 1.** We list 8 evaluation metrics in the experimental results. For each dataset, we randomly sampled 100 individuals, conducted 100 LLM queries in total, and reported the average results. The query prompt is almost the same as the Prompt 1, except that here we considered 5-level (i.e., strong disagree, disagree, neutral, agree, strongly agree) for scoring scale instead of 3-level. As for each evaluation metric, here are detailed explanations:

*Generation Time (GT)* measures the computational efficiency of each large language model by recording the average inference time required to produce responses. This metric is quantified in seconds and provides insight into the practical usability of different models, with lower values indicating faster processing speeds.

*Missing Rate (MR)* quantifies the frequency at which language models fail to provide the requested scoring output due to limitations in their knowledge base. This metric is calculated as the percentage of instances where the model cannot generate a proper response (with output score 0 - Unknown), highlighting gaps in the model's capability to handle certain types of queries or domains.

*Indecisive Rate (IR)* captures the proportion of responses where models express uncertainty or provide neutral answers rather than definitive judgments (with output score 2 - Neutral). This metric reflects the model's confidence level and willingness to make clear assessments, with higher rates indicating more cautious or uncertain behavior.

*Privacy Preservation (PP)* evaluates the model's ability to protect individual identities by effectively anonymizing personal information in its responses. This metric assesses how well the model handles sensitive data and maintains privacy standards while still providing meaningful analysis. For each response, if there contains any individual name information, return 0, otherwise return 1.

*Output Formatting (OF)* measures adherence to specified response structure and format requirements. This metric evaluates whether the model consistently follows given instructions regarding how responses should be organized and presented, ensuring usability and consistency. For each response, if it absolutely follows the given instructions and the output template format, return 1, otherwise return 0.

*Context Consistency (CC)* assesses the internal coherence between different components of the model's response, specifically examining alignment between the analysis, assigned score, and provided justification.

*Factual Accuracy (FA)* measures the absence of factual errors in the model's output, evaluated through cross-validation using mutual critique between different language models. This metric is crucial for determining the reliability and trustworthiness of the generated content.

Note that both CC and FA metrics were evaluated through a generator-evaluator manner by 4 different evaluator LLMs (Gemini-2.5-Pro-Exp-03-25, Llama-4-Maverick-03-26-Experimental, ChatGPT-4o-latest(2025-03-26), and Grok-3-Preview-02-24), to ensure logical consistency within responses. Basically, we collect the generated trait analysis output by 10 generator LLMs, and feed into other 4 evaluator LLMs. The evaluator LLMs will return 0 (indicating No) or 1 (indicating Yes). The evaluation prompt is presented in Prompt 2. There are mainly two reasons why we do not use human evaluators but instead choosing LLM evaluators: (1) First, human evaluator is expensive and costly; (2) Second, except loyal fans, most people may not have an in-depth understanding about a celebrity or athlete. To that end, LLMs probably have seen more information about certain celebrity or athlete than normal human in general. Therefore, for factual accuracy evaluation, it is reasonable

---

**Prompt 2.** *(Evaluation Prompt for Context Consistency and Factual Accuracy)*

---

*Evaluation Task*
*You are an expert evaluator for behavior trait analysis results generated by LLMs. You will analyze the following output and evaluate it on two specific criteria.*

*Input Information*
- *Name: {name}*
- *Gender: {gender}*
- *Description: {league} player, from {country} (AthlePersona) — {occupation}, from {country} (CelebPersona)*
- *Model Output: {behavior trait analysis generated by LLMs}*

*Evaluation Criteria*
1. ***Context Consistency [0/1]***
   *Check if each of the Big Five trait analyses is consistent with the assigned score (0-5)*
   *Check if the justification for each score aligns with the analysis*
   *Score 1 if all analyses are internally consistent with their scores and justifications*
   *Score 0 if any inconsistencies exist (e.g., describing high extraversion traits but giving a score of 2)*
2. ***Factual Accuracy Assessment [0/1]***
   *Check if the analysis have clear factual errors or highly speculative claims presented as facts*
   *Check if the claims about the celebrity's behaviors, career patterns, or public statements appear generally accurate based on common knowledge*
   *Score 1 if the claims appear generally accurate or the model does not make any claims due to insufficient information*
   *Score 0 if there are clear factual errors or highly speculative claims presented as facts*

*Required Output Format*
   *Context_consistency: [0/1] - [Justification]*
   *Factual_accuracy: [0/1] - [Justification]*
   *Summary: [Score1-Score2]*

---

to use LLM evaluators. Note that in this way, we aim to point out any statement which absolutely violates the factuality or commonsense. As for evaluating context consistency, it turns out to be a text interpretation task, it is also reasonable to apply LLMs.

*Overall Score (OS)* provides a comprehensive performance measure by calculating the average of all evaluation metrics except Generation Time. The score calculation is:

$$OS = \frac{1}{6} \times [PP + OF + CC + FA + (1 - MR) + (1 - IR)]. \tag{4}$$

It offers a holistic view of each model's capabilities across the various assessment dimensions.

**Analysis.** Table 1 presents a comparative evaluation of 10 LLMs. ChatGPT-4o-Latest (Achiam et al., 2023) and Gemini-2.5-Pro (Team et al., 2023) achieved the highest overall scores. Performance is consistently stronger on `CelebPersona` than on `AthlePersona`, indicating that assessing athlete personalities is more challenging. This is particularly reflected in the higher MR on `AthlePersona`, which is possibly due to the limited public information available for younger or less prominent athletes. While GT varies substantially across models, both PP and OF are consistently strong. IR differs notably, e.g., 0.46 for Qwen-Plus (Bai et al., 2023) while 0.11 for DeepSeek-R1 (Liu et al., 2024), suggesting significant variation in models' confidence calibration.

## A4.2 DETAILS ABOUT THE IMPACT OF SCORING SCALE (REGARDING FIGURE 2)

Prior research has demonstrated the importance of prompt engineering strategies in enhancing LLM performance across various tasks (Serapio-García et al., 2023; Jiang et al., 2023; Tseng et al., 2024;

**Prompt 3.** *(Comparison on Scoring Scale for Different Prompts)*

*[Number–L3–Inc]*

- **0 = Insufficient information** – *Not enough reliable public information to assess the trait. The trait's presence or absence is unknown or unclear due to lack of data.*
- **1 = Disagree** – *Clear evidences contradict the trait*
- **2 = Neutral** – *Evidence is mixed or the trait is not prominent. There is enough information, but it does not strongly support or contradict the trait.*
- **3 = Agree** – *Clear evidences support the trait*

*[Number–L3–Dec]*

- **0 = Insufficient information** – *Not enough reliable public information to assess the trait. The trait's presence or absence is unknown or unclear due to lack of data.*
- **3 = Disagree** – *Clear evidences contradict the trait*
- **2 = Neutral** – *Evidence is mixed or the trait is not prominent. There is enough information, but it does not strongly support or contradict the trait.*
- **1 = Agree** – *Clear evidences support the trait*

*[Text–L3–Inc]*

- **Insufficient information** – *Not enough reliable public information to assess the trait. The trait's presence or absence is unknown or unclear due to lack of data.*
- **Disagree** – *Clear evidences contradict the trait*
- **Neutral** – *Evidence is mixed or the trait is not prominent. There is enough information, but it does not strongly support or contradict the trait.*
- **Agree** – *Clear evidences support the trait*

*[Number–L5–Inc]*

- **0 = Insufficient information** – *Not enough reliable public information to assess the trait. The trait's presence or absence is unknown or unclear due to lack of data.*
- **1 = Strongly Disagree** – *Clear evidences contradict the trait*
- **2 = Disagree** – *Some evidences contradict the trait*
- **3 = Neutral** – *Evidence is mixed or the trait is not prominent. There is enough information, but it does not strongly support or contradict the trait.*
- **4 = Agree** – *Some evidences support the trait*
- **5 = Strongly Agree** – *Clear, consistent evidences support the trait*

*[Number–L5–Dec]*

- **0 = Insufficient information** – *Not enough reliable public information to assess the trait. The trait's presence or absence is unknown or unclear due to lack of data.*
- **5 = Strongly Disagree** – *Clear evidences contradict the trait*
- **4 = Disagree** – *Some evidences contradict the trait*
- **3 = Neutral** – *Evidence is mixed or the trait is not prominent. There is enough information, but it does not strongly support or contradict the trait.*
- **2 = Agree** – *Some evidences support the trait*
- **1 = Strongly Agree** – *Clear, consistent evidences support the trait*

*[Text–L5–Inc]*

- **Insufficient information** – *Not enough reliable public information to assess the trait. The trait's presence or absence is unknown or unclear due to lack of data.*
- **Strongly Disagree** – *Clear evidences contradict the trait*
- **Disagree** – *Some evidences contradict the trait*
- **Neutral** – *Evidence is mixed or the trait is not prominent. There is enough information, but it does not strongly support or contradict the trait.*
- **Agree** – *Some evidences support the trait*
- **Strongly Agree** – *Clear, consistent evidences support the trait*

Wang et al., 2023; Li et al., 2024a; Tang et al., 2025). These foundational studies have established that prompt structure and presentation significantly influence model outputs, particularly in psychological assessment applications. Building on this foundation, we systematically investigate how variations in *scoring scale format* affect the consistency of LLM-generated trait assessments across the Big Five traits, with implications for reliable automated psychological evaluation.

**How to Design Prompts?** As shown in Fig. 2, the radar and box plots in the top and middle illustrate the extent of *intra-prompt* variability across the Big Five traits, while the bottom panel reports Manhattan distances between prompts to capture *inter-prompt* differences. Across both `CelebPersona` and `AthlePersona` datasets, Llama-4-Maverick (Touvron et al., 2023) stands out for its highly stable outputs, followed by Gemini-2.5-Pro (Team et al., 2023). In contrast, Qwen2.5-Max (Bai et al., 2023) tends to produce the most variable results. Among these prompts, the "Number-L3-Inc" format consistently yields the lowest variance, suggesting that coarse, numerically formatted 3-point scales help LLMs produce more deterministic responses. Conversely, more complex prompts, especially those using Level-5 textual scales, lead to noticeably higher variability. Taken together, these findings suggest that prompt design, particularly scale granularity and formatting, plays a critical role in shaping the reliability of LLM-based trait assessment.

**Experimental Design and Methodology in Figure 2).** We evaluated five top-performing LLMs using a structured prompt format `[Number/Text]{[L3/L5]{[Inc/Dec]`, where elements specify response type (numerical vs. textual), scale granularity (3-level vs. 5-level), and ordering (increasing vs. decreasing). We list all different scoring scale in different prompts in Prompt 3.

This systematic approach enables comprehensive analysis of how different formatting choices interact to influence model behavior. Each model was tested across 100 trials per prompt format on both CelebPersona and AthlePersona datasets, with temperature set to 0 to reduce stochastic variability (even though temperuture 0 will still have output variation) and isolate prompt-related effects. Consistency was quantified using standard deviation (std) of trait scores across repeated runs, providing direct measures of output stability.

**Comprehensive Analysis Framework.** Our analysis encompasses three complementary perspectives as shown in Figure 2: (1) Top: trait-specific variability patterns through radar plots, (2) Middle: aggregate consistency measures via box plot distributions, and (3) Bottom: inter-prompt relationship quantification using Manhattan distance matrices. This multi-faceted approach provides both granular insights into individual trait reliability and broader patterns in prompt format effectiveness.

**Model Performance Hierarchy and Stability Patterns.** The analysis reveals a clear performance hierarchy among evaluated models. Llama-4-Maverick demonstrates exceptional consistency with standard deviations consistently below 0.2 across all prompt formats and behavior traits, forming tight, regular polygons in radar plots that indicate robust internal mechanisms for maintaining consistent assessments. The model's box plots show minimal variability between prompt formats with few outliers, suggesting sophisticated handling of diverse input structures.

Gemini-2.5-Pro occupies an intermediate position with generally low variability but occasional sensitivity to specific prompt formats, evidenced by longer box plot whiskers and more distributed quartiles. The model shows particular stability with numerical formats while demonstrating increased variance with textual scales, indicating format-dependent reliability patterns. ChatGPT-4o-Latest exhibits moderate consistency overall but with notable prompt-dependent variations, particularly visible through outliers in box plot distributions. While generally reliable, certain prompt-model-trait combinations produce unexpectedly high variability, suggesting sensitivity to specific formatting choices. Grok-3-Beta shows concerning instability, particularly in AthlePersona where some prompt formats yield standard deviations exceeding 0.8. Wide interquartile ranges indicate dramatic consistency variations depending on prompt format, with pronounced radar plot irregularities revealing trait-specific vulnerabilities. Qwen2.5-Max consistently ranks as the least reliable model, exhibiting high median standard deviations and extensive outliers reaching above 1.0. The model's radar plots often show expanded, irregular shapes indicating inconsistent performance across traits, with Manhattan distances exceeding 2.0 for complex formats.

**Trait-Specific Consistency Patterns.** The radar plot analysis reveals compelling trait-specific reliability patterns. Openness emerges as the most stable trait across nearly all models and prompt formats, consistently showing standard deviations below 0.3. This stability suggests that LLMs

demonstrate inherent consistency when evaluating creative and intellectual characteristics, possibly due to clearer linguistic markers for openness-related traits in training data.

Neuroticism presents notable dataset dependency, showing moderate stability in CelebPersona but considerably higher variability in AthlePersona, particularly for less stable models where standard deviations can exceed 1.0. This context-dependent pattern indicates that evaluation domain significantly influences how models interpret emotional stability markers. Extraversion and Agreeableness exhibit intermediate variability levels with distinct model-specific patterns. The geometric shapes formed by different prompt formats in radar plots reveal systematic differences: simpler formats tend to create smaller, more regular polygons, while complex textual formats often produce irregular, expanded shapes indicating inconsistent cross-trait performance.

**Format Optimization and Complexity Trade-offs.** The Number-L3-Inc format consistently yields the lowest variance across models and datasets, demonstrating that simple numerical 3-level scales enhance deterministic responses. Box plot analyses show this format produces the tightest distributions with minimal outliers across all models. Manhattan distance matrices reveal that Number-L3-Inc and Number-L3-Dec formats show consistently low inter-prompt distances (often below 1.0), indicating that scale direction has minimal impact when using simple numerical formats.

Conversely, textual 5-level formats (Text-L5-Inc/Dec) produce significantly higher variability, with standard deviations often exceeding 0.5 and Manhattan distances reaching above 2.0 between prompt pairs. This indicates that textual formats not only increase intra-prompt variability but fundamentally alter response distributions compared to numerical approaches. The increased granularity of 5-level scales appears to introduce additional decision boundaries that models interpret inconsistently. Number-L5 formats show intermediate complexity, exhibiting distances that fall between L3 numerical formats and textual formats. This suggests that 5-level scales represent a transitional complexity level—more challenging than 3-level scales but not as fundamentally different as textual ones.

**Cross-Dataset Insights and Domain Effects.** Systematic comparison between CelebPersona and AthlePersona reveals important domain-dependent patterns. AthlePersona generally produces higher standard deviations and inter-prompt distances across most models, suggesting that athlete trait assessment presents inherent challenges for LLMs. This pattern may reflect training data biases, where celebrity personalities are more extensively documented in text corpora compared to athlete psychological profiles, leading to less robust assessment capabilities in athletic contexts.

**Implications and Final Model Selection.** These findings challenge conventional assumptions about measurement precision in automated assessment contexts. Counter-intuitively, reducing scale granularity and employing numerical rather than textual formats substantially improves reliability, suggesting that cognitive complexity reduction outweighs precision benefits of more detailed scales. Based on our comprehensive analysis across multiple evaluation dimensions, we made the following strategic selections for our trait generation framework: After careful consideration of the consistency patterns, trait-specific reliability, and cross-dataset performance, we chose the **Number-L3-Inc format** as our standardized prompt structure. This format demonstrated the lowest variance across all models and datasets, with standard deviations consistently below 0.3 and minimal inter-prompt distances, ensuring maximum reliability in automated trait assessment.

For model selection, we adopted a multi-model approach incorporating **Llama-4-Maverick**, **ChatGPT-4o-Latest**, and **Gemini-2.5-Pro**. Llama-4-Maverick serves as our primary model due to its exceptional consistency (std ¡ 0.2) across all traits and formats. ChatGPT-4o-Latest provides complementary reliability with moderate consistency and broad accessibility, while Gemini-2.5-Pro offers additional validation particularly for numerical format processing. This ensemble approach leverages the strengths of multiple models while mitigating individual model limitations observed in our analysis. Notably, we excluded Grok-3-Beta and Qwen2.5-Max from our final selection due to their concerning instability patterns, with standard deviations frequently exceeding 0.8 and inconsistent cross-trait performance that could compromise assessment reliability.

The observed trait-specific and dataset-dependent variations underscore the critical importance of careful prompt design in LLM-based psychological evaluation systems. The convergent evidence across radar plots, box plot distributions, and distance matrices demonstrates that prompt engineering represents a fundamental factor in determining assessment reliability, with implications extending beyond trait evaluation to broader automated psychological assessment applications.

Table A6: Descriptions and suitability of different independence test methods used in the paper.

| Test | Full Name | Description | Variable Type |
|------|-----------|-------------|---------------|
| CSQ | Chi-Square Test | A classical test that evaluates whether two categorical variables are statistically independent. | Categorical |
| GSQ | G-Square Test | A likelihood-ratio version of the Chi-Square test, more robust in some small sample cases. | Categorical |
| RCIT | Randomized Conditional Independence Test | A non-parametric method using randomized Fourier features to approximate kernel-based CI testing. | Continuous/Mixed |
| HSIC | Hilbert-Schmidt Independence Criterion | A kernel-based method for measuring dependence in high-dimensional data using reproducing kernel Hilbert spaces. | Continuous/Mixed |
| KCI | Kernel-based Conditional Independence Test | A kernel-based extension of HSIC for testing conditional independence, suitable for complex data. | Continuous/Mixed |

## A5 DETAILS ABOUT INDEPENDENT TEST (IT) RESULTS

To evaluate independence relationships across different variable types in our analysis, we employ five statistical testing methods. For discrete variables, we utilize two classical approaches: the Chi-square test (Tallarida et al., 1987), which evaluates statistical independence between categorical variables, and the G-square test (Tsamardinos et al., 2006), a likelihood-ratio variant that demonstrates improved robustness in small sample scenarios. For continuous and mixed variable types, we implement three kernel-based methods: the Hilbert-Schmidt Independence Criterion (HSIC) (Gretton et al., 2005), which measures dependence in high-dimensional data using reproducing kernel Hilbert spaces; the Randomized Conditional Independence Test (RCIT) (Strobl et al., 2019), a non-parametric approach that employs randomized Fourier features to approximate kernel-based conditional independence testing; and the Kernel-based Conditional Independence Test (KCI) (Zhang et al., 2012), which extends HSIC methodology for testing conditional independence in complex data structures. This comprehensive suite of methods enables robust independence testing across diverse data types encountered in our experimental framework. A dependency is deemed significant if $p < 0.05$, and each cell in Fig. 3(a)/(b) shows the number of methods that detect such significance, and we summarize these 5 methods in Table A3 and Table A4.

### A5.1 DETAILS ON VOTING AND AGGREGATION

As described in the main paper, trait scores for each individual are obtained from three LLMs, which generate text descriptions that are mapped into Big Five trait scores. These outputs are then combined into a single score per trait using a two-step aggregation procedure. First, we discard any score of '0' (denoting *Insufficient Information*) to retain only confident assessments. Second, among the remaining values, we take the median, rounding up when necessary. This median-based rule is more robust to outliers than a simple mean.

**Example.** Suppose three LLMs output scores [2, 3, 0] for Extraversion. After discarding the '0', the remaining scores are [2, 3]. The median is 2.5, which we round up to 3 as the final aggregated trait.

For `CelebPersona`, each individual is associated with multiple images annotated with binary facial attributes (e.g., *Big Nose*, *High Cheekbones*). Since different images may yield different attribute values, we aggregate them by majority voting across all available images. If the votes are unequal, the majority determines the attribute value: $-1$ for "Absent" and $+1$ for "Present." In the case of an exact tie (equal votes), we assign the value 0, denoting an *indeterminate* outcome. This process ensures that each celebrity has a consistent, person-level attribute vector, while explicitly flagging ambiguous cases.

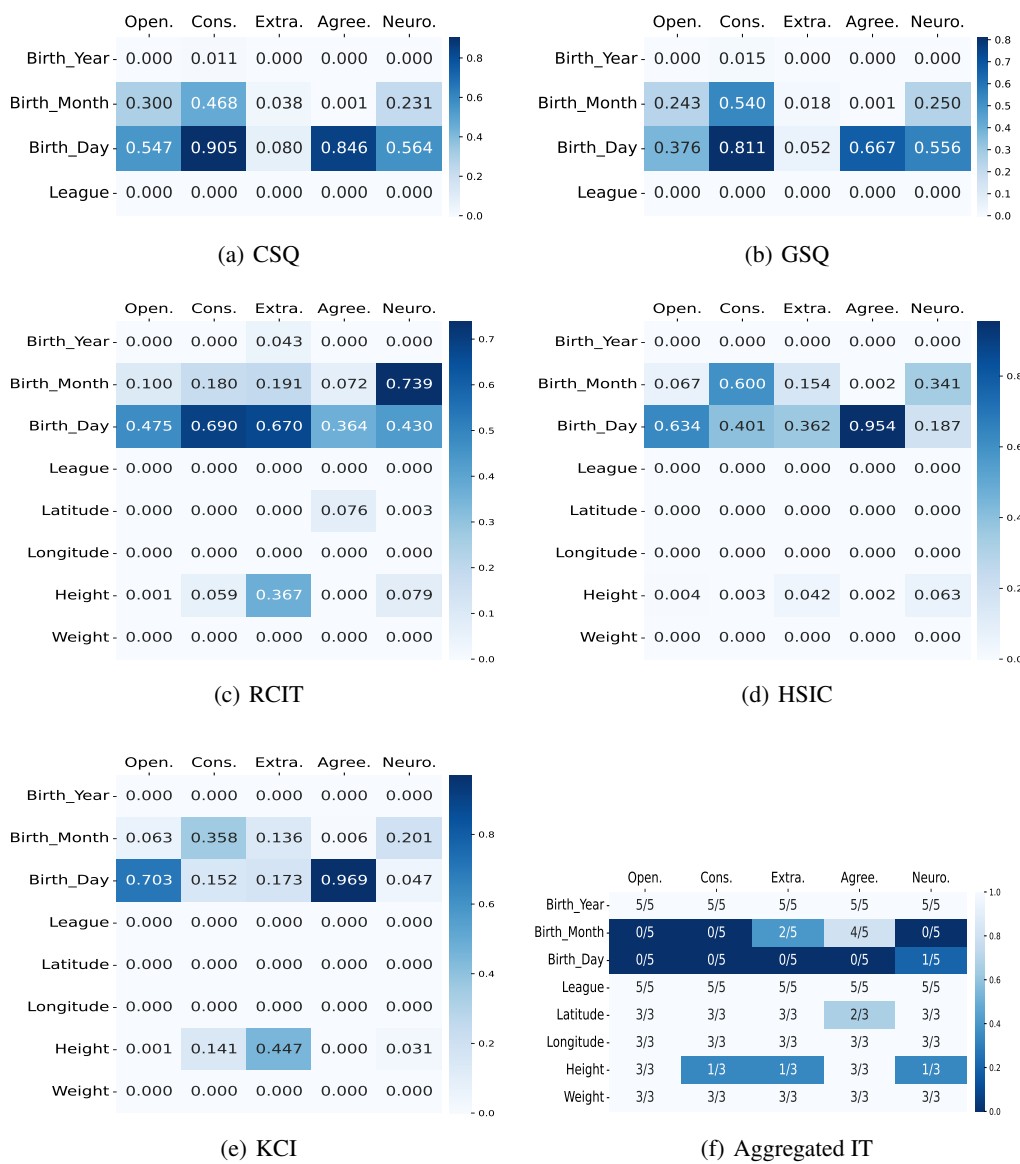

Figure A3: AthlePersona: Heatmap of P-value obtained from different independence test.

Overall, this aggregation strategy increases robustness by filtering uncertain outputs, reducing sensitivity to outliers, and providing interpretable features at the individual level.

## A5.2 DETAILS ABOUT IT RESULTS OF ATHLEPERSONA

Figure A3 presents heatmaps of p-values from different statistical independence. The Chi-Square Test (CSQ) and G-Square Test (GSQ) show remarkably similar patterns, which is expected given their shared theoretical foundation for categorical variables. Overall, the independence test analysis reveals limited but significant demographic-trait dependencies in the AthlePersona dataset. Most relationships show p-values well above the 0.05 significance threshold, indicating statistical independence between demographic features and behavior traits. However, notable exceptions include birth year and league's strong dependence with all big five traits, birth month associations with Agreeableness in the CSQ test (p = 0.001), which represents the strongest dependency detected. Birth day shows somewhat clear independence with trait in most methods. The kernel-based methods (RCIT, HSIC, KCI) generally

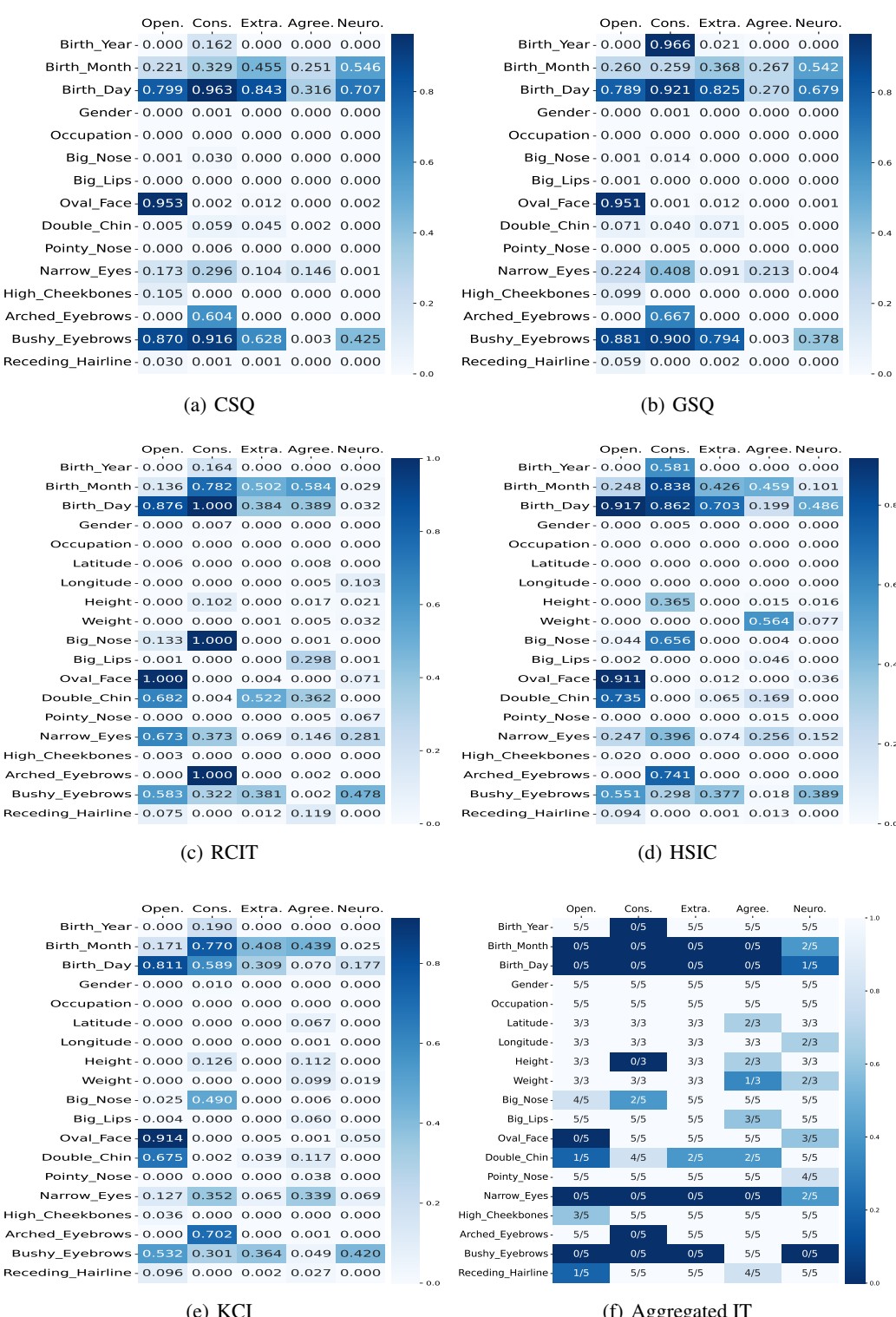

Figure A4: CelebPersona: Heatmap of P-value obtained from different independence test.

produce lower p-values, indicating stronger evidence for dependencies. Most relationships show p-values between 0-0.01, suggesting statistical dependence among variables such as birth year, league,

latitude, longitude, weight, and the Big Five behavior traits. Interestingly, weight is more dependent on openness, agreeableness, and neuroticism, while being more independent of conscientiousness and extraversion.

Trait-specific analysis reveals that most Big Five dimensions operate independently of the measured demographic factors in athletic populations. Agreeableness shows the most consistent evidence of demographic sensitivity, particularly with birth timing variables, though significant relationships ($p < 0.05$) remain infrequent across methods. Openness, Conscientiousness, Extraversion, and Neuroticism demonstrate predominantly dependent relationships with demographic features, with p-values typically samller than 0.05 across most variable-method combinations. Particularly, league affiliation, geographic coordinates (latitude, longitude), and birth year show consistent results, with most methods yielding very low p-values (near 0.000) suggesting dependence, while birth month and birth day produce high p-values indicating independence.

The multi-method validation approach reveals important methodological insights about dependency detection reliability. Classical categorical tests (CSQ, GSQ) occasionally detect marginal associations that kernel-based methods (RCIT, HSIC, KCI) fail to identify, suggesting method-specific sensitivities rather than robust dependencies. The independence test heatmap shows mixed results: some variables like birth month, birth day, and height demonstrate low consensus scores (0-2 out of 5 methods achieving p ¡ 0.05), indicating weak or inconsistent dependencies. However, several variable-trait combinations achieve moderate to high consensus scores, primarily involving league, latitude, longitude, and weight. This pattern suggests a nuanced relationship where certain demographic factors (geographic and league-related variables) show more consistent associations with behavior traits in athletic populations than temporal or physical characteristics.

The dependencies between Big Five behavior traits and league, latitude, longitude, and weight in athletic populations likely reflect a complex interplay of self-selection, environmental influences, and sport-specific demands. League affiliations may attract distinct behavior trait profiles—team sports favoring extraversion and agreeableness for collaboration, while individual sports might select for conscientiousness and controlled neuroticism. Geographic variables (latitude/longitude) capture regional cultural differences in values like individualism versus collectivism, as well as environmental factors such as climate that research has linked to behavior trait development. Weight dependencies may emerge through multiple pathways: conscientiousness influencing self-regulation of diet and exercise, neuroticism affecting stress-related eating behaviors, openness driving willingness to experiment with training regimens, and sport-specific body type requirements that indirectly link physical characteristics to the behavior traits favored in those sports. These relationships represent genuine demographic-trait associations rather than statistical noise because they align with theoretically plausible mechanisms involving cultural adaptation, environmental pressures, and the mutual influence between behavior traits and lifestyle choices in elite athletic contexts.

### A5.3  Details about IT Results of CelebPersona

Figure A4 shows heatmaps of p-values from different statistical independence tests evaluating the relationship between facial/demographic features and Big Five personality traits in the CelebPersona dataset. Features like birth year, gender, occupation, latitude, longitude, pointy nose and big lips frequently show strong associations with Big Five traits. In contrast, attributes like birth day, narrow eyes and bushy eyebrows generally appear independent of traits.

The CelebPersona dataset reveals several robust dependency patterns with p-values consistently below 0.05 across multiple methods. Birth timing variables demonstrate the strongest dependencies: birth day shows significant associations with openness, conscientiousness, and extraversion across kernel-based methods, suggesting developmental timing effects on trait formation. Birth month exhibits dependencies with conscientiousness and moderate associations across other traits. Among facial features, big nose demonstrates consistent dependencies with conscientiousness across kernel methods, while bushy eyebrows shows significant associations with openness and extraversion. Weight exhibits notable dependencies with agreeableness and neuroticism, indicating body composition-trait linkages. Narrow eyes shows dependencies with conscientiousness and agreeableness, while oval face demonstrates associations with neuroticism and other traits.

The aggregated IT results confirm these dependencies with higher consensus scores for birth day (3-4/5 methods), bushy eyebrows (3-4/5 methods), and weight (2-3/5 methods), indicating genuine

associations rather than statistical noise. Classical methods (CSQ, GSQ) detect fewer significant relationships, suggesting that non-linear dependency structures dominate celebrity trait-morphology associations. These findings support evolutionary psychology theories linking facial morphology to behavior traits, particularly the relationship between eyebrow prominence and openness/extraversion, and nose characteristics with conscientiousness. The effects of the timing of the birth may reflect seasonal developmental influences or cohort effects specific to the career trajectories of the entertainment industry, where certain combinations of behavior trait and timing of the birth provide advantages in celebrity achievement.

The dependency patterns in celebrity populations reveal intriguing domain-specific insights that distinguish them from general populations. The pronounced birth timing effects, particularly the strong associations between birth day and multiple behavior traits, suggest that developmental timing may interact with entertainment industry selection pressures in unique ways. Celebrities born on certain days may possess behavior configurations that enhance their ability to navigate public scrutiny, media attention, and performance demands. The facial feature dependencies present a complex picture of appearance-behavior relationships: the consistent association between bushy eyebrows and openness/extraversion aligns with research on facial masculinity and dominance signaling, while the nose-conscientiousness relationship may reflect underlying genetic correlations between facial development and self-regulatory capacity. Weight dependencies with agreeableness and neuroticism indicate that body image management, a critical aspect of celebrity careers, may both influence and be influenced by behavior traits related to social harmony and emotional stability. The higher dependency rates detected by kernel methods compared to classical approaches suggest that celebrity behavior-morphology relationships involve complex, non-linear interactions that traditional statistical methods fail to capture, possibly reflecting the multifaceted nature of public persona where appearance, behavior trait, and career success form intricate feedback loops.

## A6 Theorems and Proofs

In this section, we will present more details about the theorems and their proofs. In Theorem 1, We begin by showing how the modality-specific latent subspaces $[\mathbf{z}_m, \mathbf{s}]$, where $\mathbf{z}_m$ is modality-specific latent variables and $\mathbf{s}$ is modality-shared latent variables, can be recovered in a nonparametric manner using multiple measurements. Building on this result, then in Theorem 2, we demonstrate the identifiability of the shared latent variable $\mathbf{s}$ by leveraging the information across multiple modalities. Finally in Theorem 3,, conditioned on the recovered $\mathbf{s}$, we establish the identifiability of each modality-specific latent variable $\mathbf{z}_m$ up to minor indeterminacies, i.e., component-wise identifiability with an inner-modality permutation. The logical dependencies among the theorems are summarized in the flowchart as shown in Figure A5.

### A6.1 Proof of Theorem 1

> **Theorem 1.** *(Identifiability of Subspace) Under the causal model described above, if the estimated observations matches the true joint distribution of any $\{\mathbf{x}_{m,A}, \mathbf{x}_{m,B}, \mathbf{x}_{m,C}\}$ (they are exchangable) which are three measurements draw from one modality, and:*
>
> > *i (Well-Posed Probability): The joint, marginal, and conditional distributions of $(\mathbf{x}_{m,B}, \mathbf{z}_m)$ are all bounded and continuous.*
> >
> > *ii (Modality Variability): The operators $L_{\mathbf{x}_{m,C}|\mathbf{z}_m}$ and $L_{\mathbf{x}_{m,A}|\mathbf{x}_{m,C}}$ are injective.*
> >
> > *iii (Measurement Changes): For any $\mathbf{z}_t^{(1)}, \mathbf{z}_t^{(2)} \in \mathcal{Z}_t$ where $\mathbf{z}_t^{(1)} \neq \mathbf{z}_t^{(2)}$, we have $p(\mathbf{x}_{m,B}|\mathbf{z}_t^{(1)}) \neq p(\mathbf{x}_{m,B}|\mathbf{z}_t^{(2)}, \mathbf{s})$.*
> >
> > *iv (Differentiability): There exists a functional $M$ such that $M\left[p_{\mathbf{x}_{m,B}|\mathbf{z}_m, \mathbf{s}}(\cdot \mid \mathbf{z}_m, \mathbf{s})\right] = h(\mathbf{z}_m, \mathbf{s})$ for all $\mathbf{z}_m \in \mathcal{Z}_m$ and $\mathbf{s} \in \mathcal{S}$, where $h$ is differentiable.*
>
> *Then we have $[\hat{\mathbf{z}}_m, \hat{\mathbf{s}}] = h(\mathbf{z}_m, \mathbf{s})$, where $h$ is an invertible and differentiable function.*

**Discussion on Insufficient Measurements.** Importantly, Theorem 1 is not limited to the use of multiple measurements within a single modality for recovering latent variables. It also reveals that,

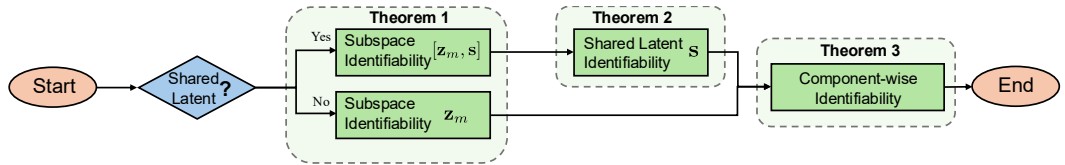

Figure A5: The high-level flowchart of the our theorems.

when the number of measurements in one modality is insufficient (i.e., fewer than 3), additional modalities can provide complementary information, provided that the required assumptions are met.

We first introduce an additional operator to represent pointwise distributional transformations, a concept widely used in the nonparametric identification literature (Hu & Schennach, 2008; Fu et al., 2025; Li et al., 2025b). To preserve generality, we denote any two variables by $a$ and $b$, with corresponding support sets $\mathcal{A}$ and $\mathcal{B}$, respectively.

**Definition 1.** *(Linear Operator) (Dunford & Schwartz, 1971) Consider two random variables $a$ and $b$ with support $\mathcal{A}$ and $\mathcal{B}$, the linear operator $L_{b|a}$ is defined as a mapping from a probability function $p_a$ in some function space $\mathcal{F}(\mathcal{A})$ onto the probability function $p_b = L_{b|a} \circ p_a$ in some function space $\mathcal{F}(\mathcal{B})$,*

$$\mathcal{F}(\mathcal{A}) \rightarrow \mathcal{F}(\mathcal{B}) : p_b = L_{b|a} \circ p_a = \int p_{b|a}(\cdot|a)p_a(a)da. \tag{5}$$

**Definition 2.** *(Diagonal Operator) Consider two random variable $a$ and $b$, density functions $p_a$ and $p_b$ are defined on some support $\mathcal{A}$ and $\mathcal{B}$, respectively. The diagonal operator $D_{b|a}$ maps the density function $p_a$ to another density function $D_{b|a} \circ p_a$ defined by the pointwise multiplication of the function $p_{b|a}$ at a fixed point $b$:*

$$p_{b|a}(b \mid \cdot)p_a = D_{b|a} \circ p_a, where \ D_{b|a} = p_{b|a}(b \mid \cdot). \tag{6}$$

For notational convenience, we define $\mathbf{w}_m := [\mathbf{z}_m, \mathbf{s}]$, with support set $\mathcal{W}_m$.

*Proof.* The vectors $\mathbf{x}_{m,A}$, $\mathbf{x}_{m,B}$, and $\mathbf{x}_{m,C}$ are conditionally independent given $\mathbf{w}_m$, which implies the following two equations:

$$p(\mathbf{x}_{m,A} \mid \mathbf{x}_{m,B}, \mathbf{w}_m) = p(\mathbf{x}_{m,A} \mid \mathbf{w}_m), \quad p(\mathbf{x}_{m,C} \mid \mathbf{x}_{m,B}, \mathbf{x}_{m,A}, \mathbf{w}_m) = p(\mathbf{x}_{m,C} \mid \mathbf{w}_m). \tag{7}$$

We can directly obtain $p(\mathbf{x}_{m,C}, \mathbf{x}_{m,B} \mid \mathbf{x}_{m,A})$ from the observed quantities $p(\mathbf{x}_{m,A})$ and $p(\mathbf{x}_{m,C}, \mathbf{x}_{m,B}, \mathbf{x}_{m,A})$. The corresponding density transformation is then given by

$$p(\mathbf{x}_{m,C}, \mathbf{x}_{m,B} \mid \mathbf{x}_{m,A}) = \underbrace{\int_{\mathcal{W}_m} p(\mathbf{x}_{m,C}, \mathbf{x}_{m,B}, \mathbf{w}_m \mid \mathbf{x}_{m,A})d\mathbf{w}_m}_{\text{integration over } \mathcal{W}_m} \tag{8}$$

$$= \underbrace{\int_{\mathcal{W}_m} p(\mathbf{x}_{m,C} \mid \mathbf{x}_{m,B}, \mathbf{w}_m, \mathbf{x}_{m,A})p(\mathbf{x}_{m,B}, \mathbf{w}_m \mid \mathbf{x}_{m,A})d\mathbf{w}_m}_{\text{factorization of joint conditional probability}} \tag{9}$$

$$= \underbrace{\int_{\mathcal{W}_m} p(\mathbf{x}_{m,C} \mid \mathbf{w}_m)p(\mathbf{x}_{m,B}, \mathbf{w}_m \mid \mathbf{x}_{m,A})d\mathbf{w}_m}_{\text{by } p(\mathbf{x}_{m,C}|\mathbf{x}_{m,B},\mathbf{x}_{m,A},\mathbf{w}_m)=p(\mathbf{x}_{m,C}|\mathbf{w}_m)} \tag{10}$$

$$= \underbrace{\int_{\mathcal{W}_m} p(\mathbf{x}_{m,C} \mid \mathbf{w}_m)p(\mathbf{x}_{m,B} \mid \mathbf{w}_m)p(\mathbf{w}_m \mid \mathbf{x}_{m,A})d\mathbf{w}_m}_{\text{by } p(\mathbf{x}_{m,A}|\mathbf{x}_{m,B},\mathbf{w}_m)=p(\mathbf{x}_{m,A}|\mathbf{w}_m)} \tag{11}$$

We begin by marginalizing over the variable $\mathbf{x}_{m,A}$ using the transformation structure defined in Equation (8):

$$\int_{\mathcal{X}_{m,A}} p(\mathbf{x}_{m,C}, \mathbf{x}_{m,B} \mid \mathbf{x}_{m,A})p(\mathbf{x}_{m,A})d\mathbf{x}_{m,A} =$$

$$\int_{\mathcal{X}_{m,A}} \int_{\mathcal{W}_m} p(\mathbf{x}_{m,C} \mid \mathbf{w}_m)p(\mathbf{x}_{m,B} \mid \mathbf{w}_m)p(\mathbf{w}_m \mid \mathbf{x}_{m,A})p(\mathbf{x}_{m,A})d\mathbf{w}_m d\mathbf{x}_{m,A}. \tag{12}$$

This joint density can be equivalently expressed in terms of the linear operators defined in Definition 1 and Definition 2:

$$[L_{\mathbf{x}_{m,B};\mathbf{x}_{m,C}|\mathbf{x}_{m,A}}p](\mathbf{x}_{m,C}) = [L_{\mathbf{x}_{m,C}|\mathbf{w}_m}D_{\mathbf{x}_{m,B}|\mathbf{w}_m}L_{\mathbf{w}_m|\mathbf{x}_{m,A}}p](\mathbf{x}_{m,C}). \tag{13}$$

Thus, the composed operators satisfy the following identity:

$$L_{\mathbf{x}_{m,B};\mathbf{x}_{m,C}|\mathbf{x}_{m,A}} = L_{\mathbf{x}_{m,C}|\mathbf{w}_m}D_{\mathbf{x}_{m,B}|\mathbf{w}_m}L_{\mathbf{w}_m|\mathbf{x}_{m,A}}. \tag{14}$$

Then, we integrate both sides over the $\mathbf{x}_{m,B} \in \mathcal{X}_{m,B}$:

$$\int_{\mathbf{x}_{m,B}\in\mathcal{X}_{m,B}} L\mathbf{x}_{m,B};\mathbf{x}_{m,C} \mid \mathbf{x}_{m,A}, d\mathbf{x}_{m,B} =$$

$$\int_{\mathbf{x}_{m,B}\in\mathcal{X}_{m,B}} L\mathbf{x}_{m,C} \mid \mathbf{w}_m D_{\mathbf{x}_{m,B}|\mathbf{w}_m}L_{\mathbf{w}_m|\mathbf{x}_{m,A}}, d\mathbf{x}_{m,B}. \tag{15}$$

Since integrating out $\mathbf{x}_{m,B}$ corresponds to marginalizing over the joint representation, we obtain

$$L_{\mathbf{x}_{m,C}|\mathbf{x}_{m,A}} = L_{\mathbf{x}_{m,C}|\mathbf{w}_m}L_{\mathbf{w}_m|\mathbf{x}_{m,A}}. \tag{16}$$

Assuming that $L_{\mathbf{x}_{m,C}|\mathbf{w}_m}$ is injective (see Assumption ii), we may invert this operator to obtain

$$L_{\mathbf{x}_{m,C}|\mathbf{w}_m}^{-1}L_{\mathbf{x}_{m,C}|\mathbf{x}_{m,A}} = L_{\mathbf{w}_m|\mathbf{x}_{m,A}}. \tag{17}$$

Substituting Equation (17) into the operator composition in Equation (14) yields

$$L_{\mathbf{x}_{m,B};\mathbf{x}_{m,C}|\mathbf{x}_{m,A}} = L_{\mathbf{x}_{m,C}|\mathbf{w}_m} D_{\mathbf{x}_{m,B}|\mathbf{w}_m} L_{\mathbf{x}_{m,C}|\mathbf{w}_m}^{-1} L_{\mathbf{x}_{m,C}|\mathbf{x}_{m,A}}. \tag{18}$$

Multiplying both sides of Equation (18) by $L_{\mathbf{x}_{m,C}|\mathbf{x}_{m,A}}^{-1}$ gives

$$L_{\mathbf{x}_{m,B};\mathbf{x}_{m,C}|\mathbf{x}_{m,A}} L_{\mathbf{x}_{m,C}|\mathbf{x}_{m,A}}^{-1} = L_{\mathbf{x}_{m,C}|\mathbf{w}_m} D_{\mathbf{x}_{m,B}|\mathbf{w}_m} L_{\mathbf{x}_{m,C}|\mathbf{w}_m}^{-1}. \tag{19}$$

The right-hand side of Equation (19) takes a canonical conjugation form. Under Assumption i and by the uniqueness of spectral decomposition (see (Conway, 1994), Chapter VII, and (Dunford & Schwartz, 1971), Theorem XV.4.5), we have

$$L_{\mathbf{x}_{m,C}|\mathbf{w}_m}D_{\mathbf{x}_{m,B}|\mathbf{w}_m}L_{\mathbf{x}_{m,C}|\mathbf{w}_m}^{-1} = (CL_{\mathbf{x}_{m,C}|\mathbf{w}_m}P)(P^{-1}D_{\mathbf{x}_{m,B}|\mathbf{w}_m}P)(P^{-1}L_{\mathbf{x}_{m,C}|\mathbf{w}_m}^{-1}C^{-1}), \tag{20}$$

where $C$ is a nonzero scalar and $P$ is an invertible operator representing a permutation of the eigenbasis.

This yields identification up to permutation and scaling:

$$L_{\mathbf{x}_{m,C}|\mathbf{w}_m} = CL_{\mathbf{x}_{m,C}|\hat{\mathbf{w}}_m}P, \qquad D_{\mathbf{x}_{m,B}|\mathbf{w}_m} = P^{-1}D_{\mathbf{x}_{m,B}|\hat{\mathbf{w}}_m}P. \tag{21}$$

Equation Equation (21) provides a unique spectral decomposition up to permutation and scaling indeterminacies. We next show how these indeterminacies can be resolved, and when they cannot, what informative conclusions may still be drawn.

First, the normalization condition

$$\int_{\mathcal{X}_{m,C}} p_{\mathbf{x}_{m,C}|\hat{\mathbf{w}}_m} \, d\mathbf{x}_{m,C} = 1 \tag{22}$$

must hold for every $\hat{\mathbf{w}}_m$. Hence, the only solution is $C = 1$.

Next, consider

$$D_{\mathbf{x}_{m,B}|\mathbf{w}_m} = P^{-1}D_{\mathbf{x}_{m,B}|\hat{\mathbf{w}}_m}P.$$

For fixed $\mathbf{x}_{m,B}$, the operator $D_{\mathbf{x}_{m,B}|\mathbf{w}_m}$ corresponds to the collection $\{p_{\mathbf{x}_{m,B}|\mathbf{w}_m}(\mathbf{x}_{m,B} \mid \mathbf{w}_m)\}$ over all $\mathbf{w}_m$. Since $P$ only permutes entries, this collection admits a unique solution:

$$\{p_{\mathbf{x}_{m,B}|\mathbf{w}_m}(\mathbf{x}_{m,B} \mid \mathbf{w}_m)\} = \{p_{\mathbf{x}_{m,B}|\hat{\mathbf{w}}_m}(\mathbf{x}_{m,B} \mid \hat{\mathbf{w}}_m)\}, \quad \forall \mathbf{w}_m, \hat{\mathbf{w}}_m. \tag{23}$$

Because these sets are unordered, consistent matching requires a reindexing of the conditioning variables. Specifically,

$$\{p_{\mathbf{x}_{m,B}|\mathbf{w}_m}(\mathbf{x}_{m,B} \mid \mathbf{w}_m^{(1)}), \ldots\} = \{p_{\mathbf{x}_{m,B}|\hat{\mathbf{w}}_m}(\mathbf{x}_{m,B} \mid \hat{\mathbf{w}}_m^{(1)}), \ldots\}, \tag{24}$$

$$\implies [p_{\mathbf{x}_{m,B}|\mathbf{w}_m}(\mathbf{x}_{m,B} \mid \mathbf{w}_m^{(\pi(1))}), \ldots] = [p_{\mathbf{x}_{m,B}|\hat{\mathbf{w}}_m}(\mathbf{x}_{m,B} \mid \hat{\mathbf{w}}_m^{(\pi(1))}), \ldots], \tag{25}$$

where $\pi$ denotes a permutation of indices.

Let $h$ denote the corresponding relabeling map. Then

$$p_{\mathbf{x}_{m,B}|\mathbf{w}_m}(\mathbf{x}_{m,B} \mid h(\mathbf{w}_m)) = p_{\mathbf{x}_{m,B}|\hat{\mathbf{w}}_m}(\mathbf{x}_{m,B} \mid \hat{\mathbf{w}}_m), \quad \forall \, \mathbf{w}_m, \hat{\mathbf{w}}_m. \tag{26}$$

By Assumption iii, distinct $\mathbf{w}_m$ correspond to distinct conditional densities, implying that $h$ is one-to-one and invertible. Moreover, Assumption iii ensures that each conditional density uniquely determines its conditioning variable, so that

$$p_{\mathbf{x}_{m,B}|\mathbf{w}_m}(\mathbf{x}_{m,B} \mid h(\mathbf{w}_m)) = p_{\mathbf{x}_{m,B}|\hat{\mathbf{w}}_m}(\mathbf{x}_{m,B} \mid \hat{\mathbf{w}}_m) \implies \hat{\mathbf{w}}_m = h(\mathbf{w}_m). \tag{27}$$

Finally, Assumption iv implies that $h$ must be differentiable. Since the VAE architecture is differentiable, it can learn such a function $h$. Writing $\hat{\mathbf{w}}_m = h(\mathbf{w}_m)$, we have

$$M\big[p_{\mathbf{x}_{m,B}|\hat{\mathbf{w}}_m}(\cdot \mid \hat{\mathbf{w}}_m)\big] = M\big[p_{\mathbf{x}_{m,B}|\mathbf{w}_m}(\cdot \mid h(\mathbf{w}_m))\big] = h(\mathbf{w}_m), \tag{28}$$

which equals $\hat{\mathbf{w}}_m$ precisely when $h$ is differentiable. $\qquad\square$

### A6.2  PROOF OF THEOREM 2

Theorem 1 establishes that the modality-specific latent variables $\mathbf{w}_m$ are block-wise identifiable. Given multiple instances of block-wise identifiability for $[\mathbf{z}_m, \mathbf{s}]$ across different modalities $m$, the shared component $\mathbf{s}$ is expected to be identifiable as well. To support this insight, we first present a related lemma from multi-view causal representation learning.

**Lemma 1** (Identifiability from a Set of Views (Yao et al., 2023)). *Consider a set of modality observations $\mathbf{x}_m$ that satisfy Assumption 2.1 in (Yao et al., 2023). Suppose there exists a set of modality-specific encoders, each mapping to a common latent space. Let $\hat{g}_{\mathbf{x}_k}^{-1}$ denote a family of encoders aimed at recovering the shared latent variables by minimizing the total entropy: $\sum_{k\in[M]} H\big(\hat{g}_{\mathbf{x}_k}^{-1}(\mathbf{x}_k)\big)$. Then, under the stated assumptions, the shared latent variables $\mathbf{s}$ are block-identifiable.*

> **Theorem 2.** *(Identifiability of Shared Subspace) Suppose assumptions are hold true for all the modality and the whole latent space, and we further assume*
>
> *i (Entropy Regularization): $\hat{g}_{\mathbf{x}_m}^{-1}$ represent a set of shared latent variable encoders that minimizes $\sum_{k\in[M]} H\big(\hat{g}_{\mathbf{x}_k}^{-1}(\mathbf{x}_k)\big)$.*
>
> *Then we have the $\hat{\mathbf{s}} = h_s(\mathbf{s})$, where $h_s$ is an invertible function.*

*Proof.* We now relate our results to Lemma 1. In (Yao et al., 2023), identifiability is established under the assumption that multiple measurement views are available for a shared latent space, and that each measurement process is invertible. This setting guarantees block identifiability of the latent space by aligning the outputs of modality-specific encoders. Specifically, for each modality m, we have:

$$[\hat{\mathbf{z}}_m, \hat{\mathbf{s}}] = h(\mathbf{z}_m, \mathbf{s}), \tag{29}$$

where the key insight is that $\hat{\mathbf{s}}$ corresponds to the shared component across all modality-specific representations $\mathbf{w}_m$, extracted via their respective encoders.

Furthermore, Lemma 1 establishes that any set of encoders minimizing the total entropy

$$\sum_{k\in[M]} H\big(\hat{g}_{\mathbf{x}_k}^{-1}(\mathbf{x}_k)\big) \tag{30}$$

can recover the ground-truth shared latent variables $\mathbf{s}$ from each modality $\mathbf{x}_m \in \mathcal{X}m$, up to a bijective transformation $h_s$:

$$\hat{\mathbf{s}} = h_s(\mathbf{s}). \tag{31}$$

That is, the shared latent content $\mathbf{s}$ is block-identified from the multi-view observations $\{\mathbf{x}_m\}_{m\in[M]}$.

Finally, since each modality-specific latent variable $\mathbf{z}_m$ is causally influenced by the shared component $\mathbf{s}$, we may apply the identifiability conditions in (Von Kügelgen et al., 2021) as a base case. This allows us to further identify $\mathbf{z}_m$ up to a modality-specific bijection $h_z$:

$$\hat{\mathbf{z}}_m = h_z(\mathbf{z}_m). \tag{32}$$

Hence, both the shared latent component $\mathbf{s}$ and the modality-specific components $\mathbf{z}_m$ are block-identifiable. $\qquad\square$

**Discussion.** In the final step of our proof, we build on the identifiability result from (Yao et al., 2023), which assumes that multiple invertible measurement processes are available to recover the shared latent variables. In contrast, our framework relaxes this assumption by not requiring each measurement process to be invertible. Instead, Theorem 1 ensures block identifiability of each modality-specific latent variable $\mathbf{w}_m$ by exploiting the information-sharing structure inherent in multi-modal and multi-measurement settings.

We further leverage a structural prior where the shared component $\mathbf{s}$ is a common cause of the modality-specific variables, rather than an effect. This causal asymmetry eliminates the need for stronger conditions such as global optimization or invariance constraints. Consequently, the conditions in (Von Kügelgen et al., 2021) apply, providing identifiability guarantees for the modality-specific latent variables $\mathbf{z}_m$.

## A6.3 PROOF OF THEOREM 3

We begin by presenting a useful lemma from (Zhang et al., 2024), which connects group-wise transformations to component-wise transformations in a Markov network. This lemma is instrumental for the subsequent proof, in particular, it enables us to first recover the latent variables within groups of adjacent nodes in the Markov network.

**Lemma 2** (Identifiability of Hidden Causal Variables). *If $\mathbf{z}_i$ is a function of at most one of $\hat{\mathbf{z}}_k$ and $\hat{\mathbf{z}}_l$, and given that $\mathbf{z}_i$ and $\mathbf{z}_j$ are adjacent in Markov network $\mathcal{M}_{\mathbf{z}}$, at most one of them is a function of $\hat{\mathbf{z}}_k$ or $\hat{\mathbf{z}}_l$. Then, there exists a permutation $\pi$ of the estimated hidden variables, denoted as $\hat{\mathbf{z}}_\pi$, such that each $\hat{\mathbf{z}}_{\pi(i)}$ is a function of (a subset of) the variables in $\{\mathbf{z}_i\} \cup \Psi_{\mathbf{z}_i}$.*

**Theorem 3.** *(Component-wise Identifiability)* Suppose the assumptions (a lot abuse) in Theorem 1, Theorem 2 is satisfied, suppose we have

 i *(Sufficient Variability):* Denote $|\mathcal{M}_{\mathbf{z}_m}|$ as the number of edges in Markov network $\mathcal{M}_{\mathbf{z}_m}$. Let

$$
\begin{aligned}
w(m) = &\Big(\frac{\partial^3 \log p(\mathbf{z}_m|\mathbf{s})}{\partial z_{m,1}^2 \partial s_{d_s}}, \cdots, \frac{\partial^3 \log p(\mathbf{z}_m|\mathbf{s})}{\partial z_{m,d_m}^2 \partial s_{d_s}}\Big) \oplus \\
&\Big(\frac{\partial^2 \log p(\mathbf{z}_m|\mathbf{s})}{\partial z_{m,1} \partial s_{d_s}}, \cdots, \frac{\partial^2 \log p(\mathbf{z}_m|\mathbf{s})}{\partial z_{m,d_m} \partial s_{d_s}}\Big) \oplus \Big(\frac{\partial^3 \log p(\mathbf{z}_m|\mathbf{s})}{\partial c_{t,i} \partial c_{t,j} \partial s_{d_s}}\Big)_{(i,j)\in\mathcal{E}(\mathcal{M}_{\mathbf{z}_m})},
\end{aligned}
\tag{33}
$$

 where $\oplus$ denotes concatenation operation and $(i,j) \in \mathcal{E}(\mathcal{M}_{\mathbf{z}_m})$ denotes all pairwise indice such that $z_{m,i}, z_{m,j}$ are adjacent in $\mathcal{M}_{\mathbf{z}_m}$. For $m \in [1, \cdots, n]$, there exist $4n + |\mathcal{M}_{\mathbf{z}_m}|$ different values of $\mathbf{s}_{d_s}$, such that the $4n + |\mathcal{M}_{\mathbf{z}_m}|$ values of vector functions $w(m)$ are linearly independent.

 ii *(Sparsity Regularization):* Let $\mathbf{G} \in \{0,1\}^{d_z \times d_z}$ denote the true adjacency matrix of the latent causal graph, and $\hat{\mathbf{G}} \in \{0,1\}^{d_z \times d_z}$ be the estimated adjacency matrix. We assume that the estimated graph is at most as dense as the true graph:

$$\|\hat{\mathbf{G}}\|_0 \leq \|\mathbf{G}\|_0,$$

 where $\|\cdot\|_0$ denotes the element-wise $\ell_0$ norm, i.e., the number of nonzero entries.

Then we have $\hat{\mathbf{z}}_{m,i} = h_i(\mathbf{z}_{m,\pi(j)})$, where $h_i$ is an invertible and differentiable function.

*Proof.* By Theorem 2, we have

$$h(\hat{\mathbf{z}}) = \mathbf{z} \implies p_{h(\hat{\mathbf{z}})} = p_{\mathbf{z}},$$

Let $J_h$ be the Jacobian matrix of $h$. The change-of-variable formula implies

$$p(\hat{\mathbf{z}}|\hat{\mathbf{s}})|\det J_{h^{-1}}| = p(\mathbf{z}|\mathbf{s})$$
$$\log p(\hat{\mathbf{z}}|\hat{\mathbf{s}}) = \log p(\mathbf{z}|\mathbf{s}) + \log|\det J_h|. \tag{34}$$

Suppose $\hat{\mathbf{z}}_k$ and $\hat{\mathbf{z}}_l$ are conditionally independent given $\hat{\mathbf{z}}_{[n]\setminus\{k,l\}}$ i.e., they are not adjacent in the Markov network over $\hat{\mathbf{z}}$. For each $\hat{\mathbf{s}}$, by (Lin, 1997), we have

$$\frac{\partial^2 \log p(\hat{\mathbf{z}}|\hat{\mathbf{s}})}{\partial\hat{\mathbf{z}}_k \partial\hat{\mathbf{z}}_l} = 0. \tag{35}$$

To see what it implies, we find the first-order derivative of Eq. equation 34:

$$\frac{\partial \log p(\hat{\mathbf{z}}|\hat{\mathbf{s}})}{\partial\hat{\mathbf{z}}_k} = \sum_{i=1}^{n} \frac{\partial \log p(\mathbf{z}|\mathbf{s})}{\partial\mathbf{z}_i} \frac{\partial\mathbf{z}_i}{\partial\hat{\mathbf{z}}_k} + \frac{\partial \log|\det J_v|}{\partial\hat{\mathbf{z}}_k}.$$

Let

$$\eta(\mathbf{s}) := \log p(\mathbf{z}|\mathbf{s}), \quad \eta_i'(\mathbf{s}) := \frac{\partial \log p(\mathbf{z}|\mathbf{s})}{\partial\mathbf{z}_i},$$

$$\eta_{ij}''(\mathbf{s}) := \frac{\partial^2 \log p(\mathbf{z}|\mathbf{s})}{\partial\mathbf{z}_i \partial\mathbf{z}_j}, \quad h_{i,l}' := \frac{\partial\mathbf{z}_i}{\partial\hat{\mathbf{z}}_l}, \quad h_{i,kl}'' := \frac{\partial^2\mathbf{z}_i}{\partial\hat{\mathbf{z}}_k \partial\hat{\mathbf{z}}_l}.$$

We then derive the second-order derivative w.r.t. $\hat{\mathbf{z}}_k$ and $\hat{\mathbf{z}}_l$ and apply Eq. equation 35:

$$0 = \sum_{j=1}^{n}\sum_{i=1}^{n} \frac{\partial^2 \log p(\mathbf{z}|\mathbf{s})}{\partial\mathbf{z}_i \partial\mathbf{z}_j} \frac{\partial\mathbf{z}_j}{\partial\hat{\mathbf{z}}_l} \frac{\partial\mathbf{z}_i}{\partial\hat{\mathbf{z}}_k} + \sum_{i=1}^{n} \frac{\partial \log p(\mathbf{z}|\mathbf{s})}{\partial\mathbf{z}_i} \frac{\partial^2\mathbf{z}_i}{\partial\hat{\mathbf{z}}_k \partial\hat{\mathbf{z}}_l} + \frac{\partial^2 \log|\det J_v|}{\partial\hat{\mathbf{z}}_k \partial\hat{\mathbf{z}}_l}$$

$$= \sum_{i=1}^{n} \frac{\partial^2 \log p(\mathbf{z}|\mathbf{s})}{\partial\mathbf{z}_i^2} \frac{\partial\mathbf{z}_i}{\partial\hat{\mathbf{z}}_l} \frac{\partial\mathbf{z}_i}{\partial\hat{\mathbf{z}}_k} + \sum_{j=1}^{n} \sum_{i:\{\mathbf{z}_j,\mathbf{z}_i\}\in\mathcal{E}(\mathcal{M}_{\mathbf{z}})} \frac{\partial^2 \log p(\mathbf{z}|\mathbf{s})}{\partial\mathbf{z}_i \partial\mathbf{z}_j} \frac{\partial\mathbf{z}_j}{\partial\hat{\mathbf{z}}_l} \frac{\partial\mathbf{z}_i}{\partial\hat{\mathbf{z}}_k}$$

$$+ \sum_{i=1}^{n} \frac{\partial \log p(\mathbf{z}|\mathbf{s})}{\partial\mathbf{z}_i} \frac{\partial^2\mathbf{z}_i}{\partial\hat{\mathbf{z}}_k \partial\hat{\mathbf{z}}_l} + \frac{\partial^2 \log|\det J_v|}{\partial\hat{\mathbf{z}}_k \partial\hat{\mathbf{z}}_l} \tag{36}$$

$$= \sum_{i=1}^{n} \eta_{ii}''(\mathbf{s})h_{i,l}'h_{i,k}' + \sum_{j=1}^{n} \sum_{i:\{\mathbf{z}_j,\mathbf{z}_i\}\in\mathcal{E}(\mathcal{M}_{\mathbf{z}})} \eta_{ij}''(\mathbf{s})h_{j,l}'h_{i,k}' + \sum_{i=1}^{n} \eta_i'(\mathbf{s})h_{i,kl}'' + \frac{\partial^2 \log|\det J_v|}{\partial\hat{\mathbf{z}}_k \partial\hat{\mathbf{z}}_l}. \tag{37}$$

Recall that $\mathcal{E}(\mathcal{M}_{\mathbf{z}})$ denotes the set of edges in the Markov network over $Z$. In the equation above, we made use of the fact that if $\mathbf{z}_i$ and $\mathbf{z}_j$ are not adjacent in the Markov network, then $\frac{\partial^2 \log p(\mathbf{z}|\mathbf{s})}{\partial\mathbf{z}_i \partial\mathbf{z}_j} = 0$ by (Lin, 1997).

By Assumption i, consider the $2d_z + |\mathcal{M}_{\mathbf{z}}| + 1$ values of $\mathbf{s}$, i.e., $\mathbf{s}^{(u)}$ with $u = 0, \ldots, 2d_z + |\mathcal{M}_{\mathbf{z}}|$, such that Eq. (37) hold. Then, we have $2d_z + |\mathcal{M}_{\mathbf{z}}| + 1$ such equations. Subtracting each equation corresponding to $\mathbf{s}^{(u)}, u = 1, \ldots, 2d_z + |\mathcal{M}_{\mathbf{z}}|$ with the equation corresponding to $\mathbf{s}^{(0)}$ results in $2d_z + |\mathcal{M}_{\mathbf{z}}|$ equations:

$$0 = \sum_{i=1}^{n}(\eta_{ii}''(\mathbf{s}^{(u)}) - \eta_{ii}''(\mathbf{s}^{(0)}))h_{i,l}'h_{i,k}' + \sum_{j=1}^{n} \sum_{i:\{\mathbf{z}_j,\mathbf{z}_i\}\in\mathcal{E}(\mathcal{M}_{\mathbf{z}})} (\eta_{ij}''(\mathbf{s}^{(u)}) - \eta_{ij}''(\mathbf{s}^{(0)}))h_{j,l}'h_{i,k}'$$

$$+ \sum_{i=1}^{n}(\eta_i'(\mathbf{s}^{(u)}) - \eta_i'(\mathbf{s}^{(0)}))h_{i,kl}'',$$

where $u = 1, \ldots, 2d_z + |\mathcal{M}_{\mathbf{z}}|$. Since $p_{\mathbf{z}}$ is twice continuously differentiable, we have

$$\eta_{ij}''(\mathbf{s}^{(u)}) - \eta_{ij}''(\mathbf{s}^{(0)}) = \eta_{ji}''(\mathbf{s}^{(u)}) - \eta_{ji}''(\mathbf{s}^{(0)}),$$

and therefore Eq. equation 38 can be written as

$$0 = \sum_{i=1}^{n}(\eta_{ii}''(\mathbf{s}^{(u)}) - \eta_{ii}''(\mathbf{s}^{(0)}))h_{i,l}'h_{i,k}' + \sum_{\substack{i,j:\\ i<j,\\ \{\mathbf{z}_i,\mathbf{z}_j\}\in\mathcal{E}(\mathcal{M}_{\mathbf{z}})}}(\eta_{ij}''(\mathbf{s}^{(u)}) - \eta_{ij}''(\mathbf{s}^{(0)}))(h_{j,l}'h_{i,k}' + h_{i,l}'h_{j,k}')$$

$$+ \sum_{i=1}^{n}(\eta_i'(\mathbf{s}^{(u)}) - \eta_i'(\mathbf{s}^{(0)}))h_{i,kl}''.$$

Consider the vectors formed by collecting the corresponding coefficients in the equation above where $u = 1, \ldots, 2d_z + |\mathcal{M}_{\mathbf{z}}|$. By Assumption A2, these $2d_z + |\mathcal{M}_{\mathbf{z}}|$ vectors are linearly independent. Thus, for any $i$ and $j$ such that $\{\mathbf{z}_i, \mathbf{z}_j\} \in \mathcal{E}(\mathcal{M}_{\mathbf{z}})$, we have the following equations:

$$h_{i,k}'h_{i,l}' = 0, \tag{38}$$

$$h_{i,k}'h_{j,l}' + h_{j,k}'h_{i,l}' = 0, \tag{39}$$

$$h_{i,kl}'' = 0.$$

It remains to show $h_{i,k}'h_{j,l}' = 0$. Suppose by contradiction that

$$h_{i,k}'h_{j,l}' \neq 0, \tag{40}$$

which implies $h_{i,k}' \neq 0$. By Eq. equation 38, we have $h_{i,l}' = 0$, which, by plugging into Eq. equation 39, indicates $h_{i,k}'h_{j,l}' = 0$. This is a contradiction with Eq. equation 40. Thus, we must have $h_{i,k}'h_{j,l}' = 0$, which indicates that $\mathbf{z}_i$ is a function of at most one of $\hat{\mathbf{z}}_k$ and $\hat{\mathbf{z}}_l$, and given that $\mathbf{z}_i$ and $\mathbf{z}_j$ are adjacent in Markov network $\mathcal{M}_{\mathbf{z}}$, at most one of them is a function of $\hat{\mathbf{z}}_k$ or $\hat{\mathbf{z}}_l$.

Then, using Lemma 2, we can obtain that there exists a permutation $\pi$ of the estimated hidden variables, denoted as $\hat{\mathbf{z}}_\pi$, such that each $\hat{\mathbf{z}}_{\pi(i)}$ is a function of (a subset of) the variables in $\{\mathbf{z}_i\} \cup \Psi_{\mathbf{z}_i}$. It is worth noting that in many cases, the above result already enables us to recover some of the hidden variables up to a component-wise transformation, that is, $\hat{\mathbf{z}}_{\cdot,i} = h_i(\mathbf{z}_{\cdot,\pi(j)})$, where $h_i$ is an invertible function. □

We next present a proposition that shows how an arbitrary permutation over all components can be resolved into a permutation within each modality block.

**Proposition 1.** *(Resolving Block-Wise Permutation) if $\hat{\mathbf{z}}_{\cdot,i} = h(\mathbf{z}_{\cdot,\pi(j)})$ and $\hat{\mathbf{z}}_m = h_m(\mathbf{z}_m)$ for any $m \in [M]$, we have $\hat{\mathbf{z}}_{m,i} = h_i(\mathbf{z}_{m,\pi(j)})$, where $h_i$ is an invertible function.*

*Proof.* Since the global mapping is given by $\hat{\mathbf{z}} = h(\mathbf{z})$, where $h = [h_1, h_2, \ldots, h_M]$ acts block-wise on each modality $\mathbf{z}_m$, the Jacobian $J_h(\mathbf{z}) = \frac{\partial \hat{\mathbf{z}}}{\partial \mathbf{z}}$ is block-diagonal:

$$J_h(\mathbf{z}) = \begin{bmatrix} J_{h_1}(\mathbf{z}_1) & 0 & 0 \\ 0 & \ddots & 0 \\ 0 & 0 & J_{h_M}(\mathbf{z}_M) \end{bmatrix}.$$

This implies that each $\hat{\mathbf{z}}_m$ depends only on $\mathbf{z}_m$.

Given the global identifiability condition $\hat{\mathbf{z}}_{\cdot,i} = h_i(\mathbf{z}_{\cdot,\pi(j)})$, and the fact that both $\hat{\mathbf{z}}_{\cdot,i}$ and $\mathbf{z}_{\cdot,\pi(j)}$ must lie in the same modality $m$ due to the block-diagonal structure, we conclude:

$$\hat{\mathbf{z}}_{m,i} = h_i(\mathbf{z}_{m,\pi(j)}).$$

□

**Discussion.** We demonstrate that multi-modality information enables the use of the shared confounder $\mathbf{s}$ as a continuous conditional prior over the modality-specific latent variables $\mathbf{z}_m$. This represents the key distinction from conventional multi-modality or multi-view frameworks (Sun et al., 2025; Yao et al., 2023; Von Kügelgen et al., 2021). By conditioning on $\mathbf{s}$—for example, a gene-level representation—we can achieve component-wise identifiability of latent variables and recover their causal graph under milder assumptions. Furthermore, Proposition 1 shows that the modality-specific latent structure $\mathbf{z}_m$, obtained via Theorem 2, facilitates the resolution of permutation indeterminacies across the latent spaces associated with different modalities.

## A7 DETAILS ABOUT NETWORK TRAINING FOR CAUSAL REPRESENTATION LEARNING

In this section, inspired by identifiability results as shown in the Theorems, we will introduce our estimation framework which enforces the proposed assumptions as constraints to identify the latent variables in each modality, in total we use several loss functions as constraints. The details are given as follows.

**Network Architecture.** For the high-dimensional data, we use a large foundation model to extract a high-dimensional feature first, and then use the 3-layer multi-layer perception (MLP) for the encoders and decoders. Specifically, for image data, we utilize ImageBind (Girdhar et al., 2023) to extract 1024-dimensional embedding vectors, as this model excels at multi-modal embedding extraction. For text descriptions, we employ the gte-Qwen2-7B-instruct model from Alibaba (Bai et al., 2023), which is specifically designed for long-sentence embedding tasks and demonstrates superior performance in capturing semantic representations from extended textual content. After this gte model, we will get a 3584-dimensional embedding vector for each input text description.

**Encoder and decoder.** Each modality $\mathbf{x}_m$ is given as an input to the corresponding encoder and outputs the estimated modality-specific latent $\hat{\mathbf{z}}_m$, exogenous variables $\hat{\eta}_m$, and shared latent variables $\mathbf{s}$ across different modalities. In one modality, to ensure the conditional independence among different $\hat{\mathbf{x}}_{m,k}$ given $\hat{\mathbf{z}}_m$, $\hat{\mathbf{x}}_{m,k}$ are passed to their corresponding $k$-th decoders, respectively, to reconstruct the observations $\hat{\mathbf{x}}_{m,k}$ in each measurement. The reconstruction loss is calculated using the mean squared error (MSE) as

$$\mathcal{L}_{\text{Recon}} = \sum_{m=1}^{M} \sum_{k=1}^{d_m} ||\mathbf{x}_{m,k} - \hat{\mathbf{x}}_{m,k}||_2^2.$$

**Conditional independence constraints.** We enforce the conditional independence condition $\mathbf{x}_{m,j} \perp\!\!\!\perp \mathbf{x}_{m,k} \mid \mathbf{z}_m$ (where $\mathbf{x}_{m,j}$ and $\mathbf{x}_{m,k}$ refer to the $j$-th and $k$-th measurements in $m$-th modality) and the independence condition on $\eta_m \perp\!\!\!\perp \mathbf{z}_m$ by enforcing the independence among components in $\gamma = [\{\hat{\mathbf{z}}_m\}_{m=1}^{M}, \{\hat{\eta}_m\}_{m=1}^{M}, \{\hat{\epsilon}_i\}_{i=1}^{d_z}]$. To implement it, we assume that $\gamma$ follows an independent prior distribution $p(\gamma)$, such as a standard isotropic Gaussian, and enforce the independence by matching the distribution of $\hat{\gamma}$ to the prior distribution. Specifically, we minimize the KL divergence between the posterior and a Gaussian prior distribution as follows:

$$\mathcal{L}_{\text{Ind}} = \text{KL}(p(\gamma)||\mathcal{N}(\mathbf{0}, \boldsymbol{I})).$$

**Proposition 2** (Conditional Independence Condition). *Denote $\mathbf{x}_{m,j}$ and $\mathbf{x}_{m,k}$ are two different measurements in one modality for the $m$-th modality with modality-specific latent variable $\mathbf{z}_m$. $\mathbf{z}_m \subset \mathbf{z}$ is the set of block-identified latent variables, and $\eta_m \subset \eta$ are exogenous variables in modality $m$. We have $\mathbf{x}_{m,j} \perp\!\!\!\perp \mathbf{x}_{m,k} \mid \mathbf{z}_m \iff \epsilon_{m,j} \perp\!\!\!\perp \epsilon_{m,k}$.*

**Proposition 3** (Independent Noise Condition). *Denote $\mathbf{z}$ and $\eta$ as the block-identified latent variables and exogenous variables across all modalities. $\epsilon$'s are the causally-related noise terms. We have $\eta \perp\!\!\!\perp \mathbf{z} \iff \eta \perp\!\!\!\perp \epsilon$.*

**Sparsity regularization.** We use normalization flow (Huang et al., 2018) to estimate the exogenous variables $\epsilon$ and implement the causal relations through a learnable adjacency matrix $\hat{\mathbf{A}}$. The binary values in $\hat{\mathbf{A}}$ represent the causal generation process between latent variables, e.g. $\hat{A}_{i,j} = 1$ indicates $\hat{z}_j$ is the parent of $\hat{z}_i$, while $\hat{A}_{i,j} = 0$ means $\hat{z}_j$ dose not contribute to the generation of $\hat{z}_i$. For each component $\hat{z}_i$, we select its parents $\text{Pa}(\hat{z}_i)$ based on the estimated causal adjacency matrix, and apply the flow transformation from $\text{Pa}(\hat{z}_i)$ to $\hat{\epsilon}_i$.

To encourage sparsity among the latent variables $\hat{\mathbf{z}}$, we introduce a regularization term on the learned adjacency matrix. The sparsity assumption indicates that the optimal causal graph should be the minimal one which still allows the model to successfully match the ground truth observational distribution. In particular, we reduce the dependencies between different components of $\hat{\mathbf{z}}$ by adding a $\mathcal{L}_1$ penalty on the adjacency matrix, s.t.,

Table A7: Key hyperparameters used in experiments.

| Hyperparameter | MNIST | PersonaX |
|---|---|---|
| Learning Rate | 2e-6 | 3e-4 |
| Training Epochs | 3000 | 3000 |
| Reconstruction Loss Coefficient | 2 | 1 |
| Conditional Independence Loss Coefficient | 1e-2 | 1e-2 |
| Sparsity Loss Coefficient | 1e-3 | 1e-3 |

$$\mathcal{L}_{\text{Sp}} = ||\hat{\mathbf{A}}||_1.$$

**Network Training.** In summary, the model parameters are optimized using the combination objective:

$$\mathcal{L} = \alpha_{\text{Recon}}\mathcal{L}_{\text{Recon}} + \alpha_{\text{Ind}}\mathcal{L}_{\text{Ind}} + \alpha_{\text{Sp}}\mathcal{L}_{\text{Sp}}. \tag{41}$$

## A8 DETAILS ABOUT SYNTHETIC EXPERIMENTS ON VARIANT MNIST

In this section, we will introduce the synthetic experiments designed to validate our proposed causal representation learning framework. We conduct comprehensive evaluations using carefully constructed datasets with known causal relationships, allowing us to systematically assess the performance of our method against established baselines.

### A8.1 DETAILS ABOUT EXPERIMENTAL SETUP

To systematically evaluate our proposed causal representation learning framework, we construct a synthetic dataset with known ground-truth causal relationships using variants of the MNIST dataset. Our synthetic dataset consists of two modalities: colored MNIST and fashion MNIST, each containing causally related latent variables. For colored MNIST, we define horizontal position as a latent cause that influences image transparency, where digits are positioned at different horizontal locations and their transparency varies accordingly. For fashion MNIST, we establish vertical position as a latent cause that affects grayscale intensity of the clothing items. The causal structure connects these modalities through a cross-modal relationship: the horizontal position in colored MNIST serves as a causal factor for the vertical position in fashion MNIST, creating a meaningful inter-modal dependency. Notably, our dataset design reflects different measurement characteristics across modalities: for fashion MNIST, each sample contains a single image representing one measurement, while for colored MNIST, we generate three images with different background colors (red, green, blue) for each sample, providing three distinct measurements that capture different aspects of the same underlying latent variables. The generated image examples are shown in Figure 5(a). The key hyper-parameters are listed in Table A7.

**Ground Truth Causal Graph and Training Configuration.** The underlying causal relationships in our synthetic dataset are illustrated in Figure 5(b). The causal graph demonstrates how latent variables within and across modalities interact: horizontal position in colored MNIST causally influences both the image transparency within the same modality and the vertical position in fashion MNIST across modalities. Subsequently, the vertical position in fashion MNIST determines the grayscale intensity of the fashion items. This carefully designed causal structure enables us to evaluate whether our method can correctly identify and disentangle these known causal relationships from the observed multi-modal data.

### A8.2 DETAILS ABOUT RESULTS AND ANALYSIS

We compare our approach against several baseline methods including MCL, BetaVAE, and MMCRL using two key metrics: $R^2$ (coefficient of determination) and MCC (Matthews Correlation Coefficient). As shown in Figure Figure 5(c), our method consistently outperforms all baseline approaches across both evaluation metrics. Specifically, our approach achieves $R^2$ scores of 0.96 and MCC scores of 0.92, demonstrating superior performance in both regression and classification tasks for causal

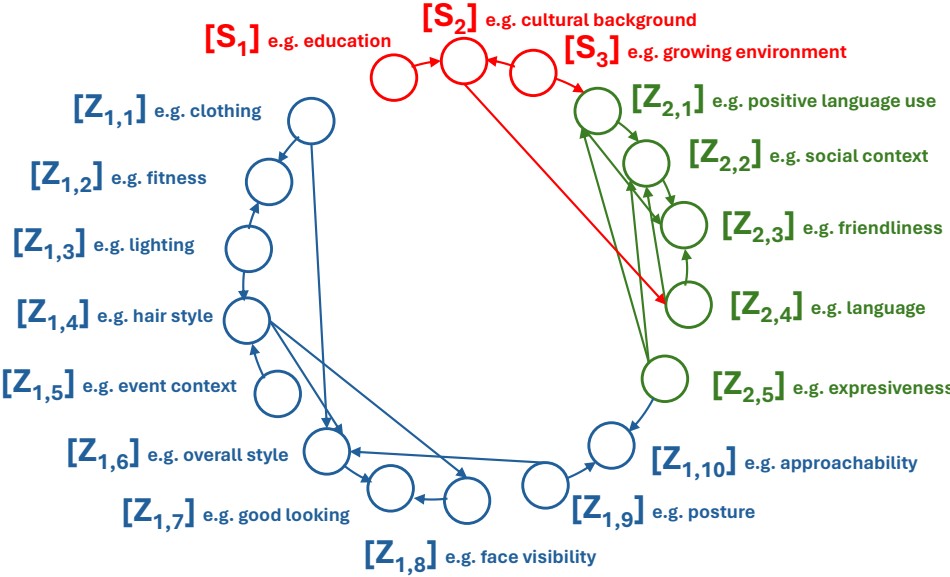

Figure A6: The causal graph with latent variables learned from `CelebPersona` dataset. Red, blue, and green nodes correspond to shared latents, facial image latents, and trait text latents.

variable identification. The substantial improvement over strong baselines like MMCRL ($R^2 = 0.90$, MCC = 0.85) validates the effectiveness of our proposed framework in learning causally meaningful representations from multi-modal observations. These results confirm that our method successfully captures the underlying causal structure while maintaining high fidelity in representation learning, even when dealing with asymmetric measurement structures across different modalities.

## A9 DETAILS ABOUT REAL-WORLD BEHAVIOR TRAIT ANALYSIS ON PERSONA$\mathbb{X}$

### A9.1 DETAILS ABOUT EXPERIMENTAL SETUP

We conduct real-world behavior trait analysis by training our network to extract latent representations from both the image and text modalities of the `CelebPersona` dataset, followed by the application of causal discovery to reveal underlying structures. The key hyper-parameters are listed in Table A7. The resulting causal graph for AthlePersona is at Fig. 6. For CelebPersona the causal graph is shown in Fig. A6, we identify three shared latent variables ($S_1$, $S_2$, $S_3$), ten latent variables derived from facial images ($Z_{1,1}$ to $Z_{1,10}$), and five latent variables extracted from behavior trait descriptions ($Z_{2,1}$ to $Z_{2,5}$). Each variable is grounded in real-world interpretable features, enabling meaningful analysis of the causal pathways.

### A9.2 DETAILS ABOUT RESULTS AND ANALYSIS

We interpret the shared latent variables $S_1$, $S_2$, and $S_3$ as representing education, cultural background, and growing environment, respectively. Notably, $S_2$ influences $Z_{2,4}$, which we interpret as cultural background shaping one's language use, while $S_3$ influences $Z_{2,1}$, suggesting that the growing environment affects the use of positive language. Furthermore, expressiveness ($Z_{2,5}$) is found to causally influence approachability ($Z_{1,10}$), reinforcing the idea that one's ability to convey emotions plays a key role in how approachable they appear. On the visual side, we observe that variations in event context ($Z_{1,5}$) and lighting conditions ($Z_{1,3}$) lead to changes in hairstyle ($Z_{1,4}$), which in turn influence face visibility ($Z_{1,8}$), overall style ($Z_{1,6}$), and how good-looking ($Z_{1,7}$) the person appears.

To validate our example, we conducted an RCIT test between the Big Five traits (Final_O to Final_N) and two sets of latent variables: five derived from trait descriptions across both datasets. We also

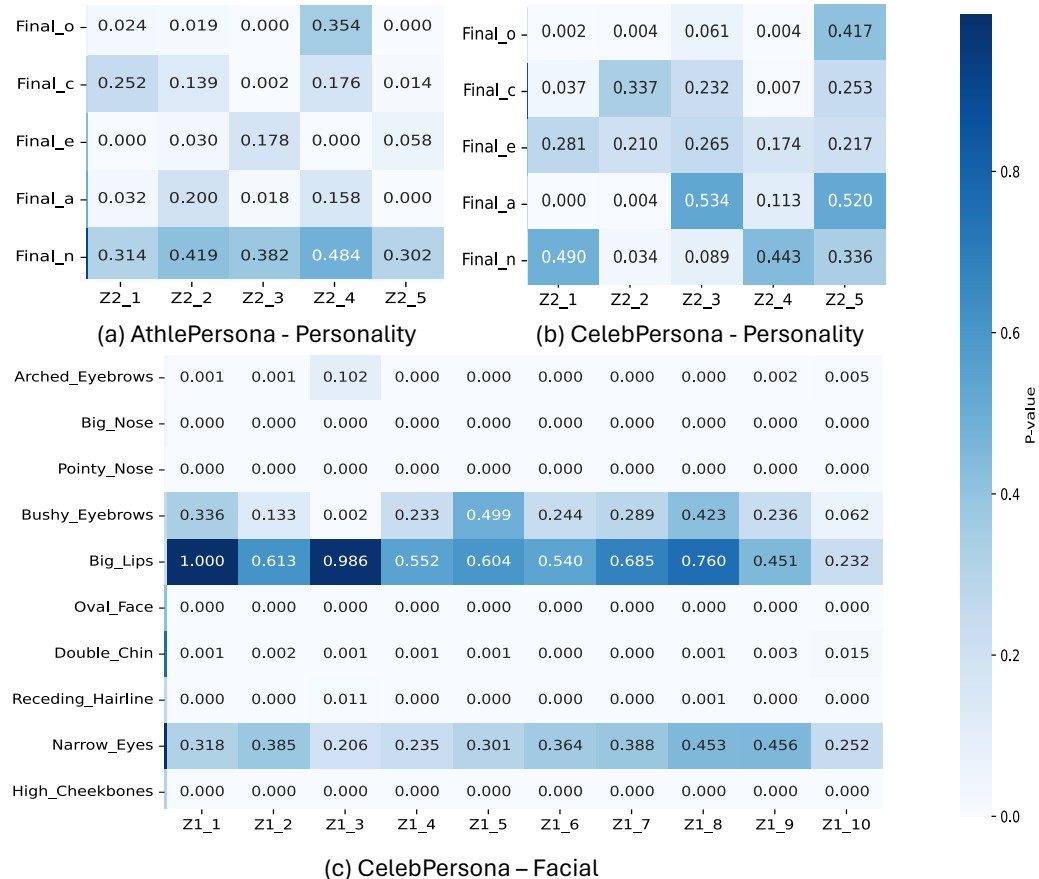

Figure A7: The RCIT test between the Big Five traits (Final_O to Final_N) and two sets of latent variables: five derived from behavior trait descriptions across both datasets, (a) AthlePersona and (b) CelebPersona. (c) refer to the same test on ten facial attributes from CelebPersona and ten latent variables derived from facial images.

carry out the same tests on ten facial attributes from CelebPersona and ten latent variables derived facial images. As shown in Figure A7 (a), confidence ($Z_{2,1}$) exhibits strong statistical dependence with Openness, Extraversion, and Agreeableness. In contrast, Self-awareness ($Z_{2,4}$) is significantly associated only with Extraversion, suggesting that more extraverted individuals tend to be more self-aware, likely due to their expressiveness, social engagement, and sensitivity.

For the test result of CelebPersona in Figure A7 (b), positive language use ($Z_{2,1}$) has significant dependence with Agreeableness indicates that more agreeable individuals are likely to use warmer and more positive language, aligning with their prosocial and empathetic tendencies. On the other hand, the high p-values across all Big Five traits suggest that expressiveness ($Z_{2,5}$) operates independently of stable behavior trait dimensions in this dataset, possibly reflecting more situational or behavior factors not captured by self-reported traits. In Figure A7 (c), the p-value heatmap confirms that many facial attributes are significantly influenced by latent appearance factors like clothing style ($Z_{1,1}$), lighting ($Z_{1,3}$), and event context ($Z_{1,5}$), as shown in the causal graph. Traits like Big_Nose, Pointy_Nose, and Oval_Face are tightly linked to hairstyle and good looking ($Z_{1,7}$).

### A9.3 RESULTS AND ANALYSIS OF BASELINE MMCRL

Similarly to above, We conduct real-world behavior trait analysis by the baseline method MMCRL (Sun et al., 2025) to extract latent representations from both the image and text modalities of the `CelebPersona` dataset, followed by the application of causal discovery to reveal underlying struc-

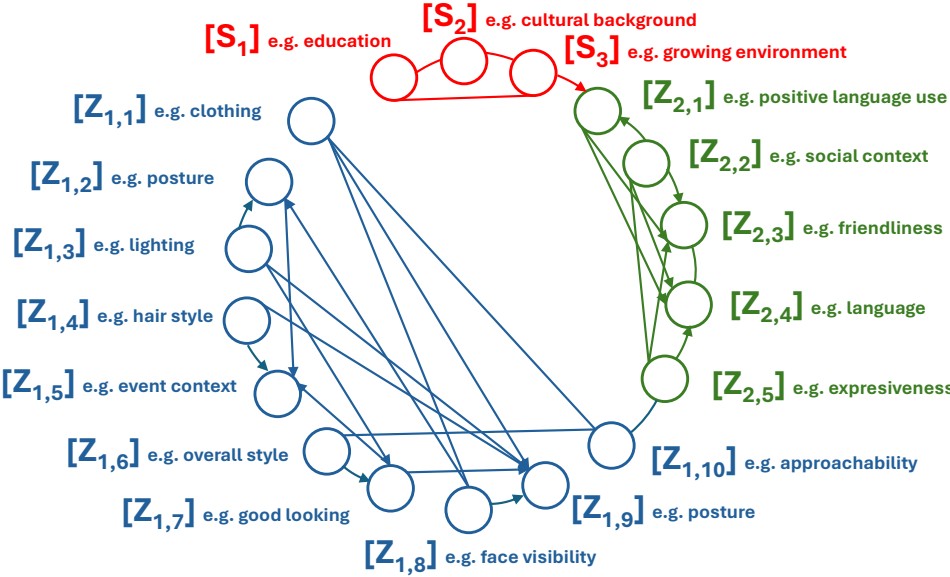

Figure A8: The causal graph by baseline method MMCRL learned from `CelebPersona` dataset. Red, blue, and green nodes correspond to shared latents, facial image latents, and trait text latents.

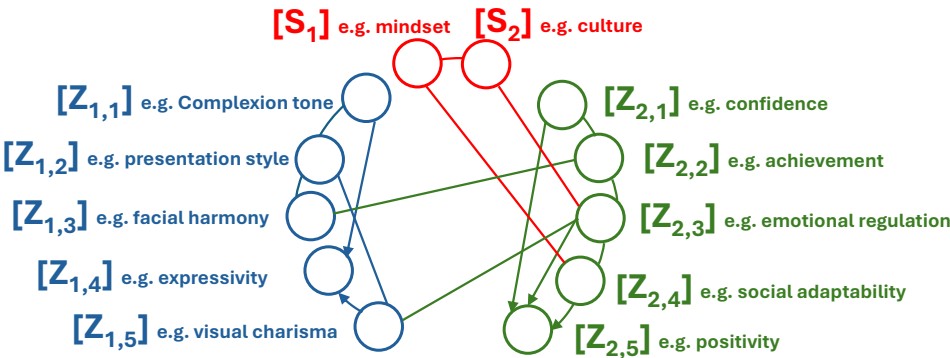

Figure A9: The causal graph by baseline method MMCRL learned from `AthlePersona` dataset. Red, blue, and green nodes correspond to shared latents, facial image latents, and trait text latents.

tures. The key hyper-parameters are listed in Table A7. The resulting causal graph for AthlePersona is at Fig. A9. For CelebPersona the causal graph is shown in Fig. A8, we identify three shared latent variables ($S_1$, $S_2$, $S_3$), ten latent variables derived from facial images ($Z_{1,1}$ to $Z_{1,10}$), and five latent variables extracted from behavior trait descriptions ($Z_{2,1}$ to $Z_{2,5}$). Each variable is grounded in real-world interpretable features, enabling meaningful analysis of the causal pathways.

We use similar way to analyze each latent variables. The results show that MMCRL, which assumes single-measurement data, produces denser and less interpretable causal graphs. In contrast, our multi-measurement CRL yields sparser, more stable, and semantically coherent cross-modal structures. Quantitatively, our method achieves higher causal alignment metrics (for example, MCC and $R^2$ in synthetic results as shown in Section 4.4) and here qualitatively shows clearer separation between visual and behavioral latent factors (in this real-world experiments). These findings confirm the advantages of our design for real-world multimodal causal analysis.

