# OpenReview forum: "PersonaX: Multimodal Datasets with LLM-Inferred Behavior Traits"
_ICLR.cc/2026/Conference — ICLR 2026 Poster_

### Official Review · Reviewer_GsZk · 2025-10-26

**Soundness:** 2
**Presentation:** 3
**Contribution:** 2
**Rating:** 4
**Confidence:** 2

**Summary:**

This paper introduces PersonaX, a pair of multimodal datasets for studying LLM-inferred behavioral traits: CelebPersona and AthlePersona. Each example includes (i) LLM-generated Big Five trait descriptions/scores, (ii) facial information, and (iii) biographical fields. The authors analyze PersonaX at two levels: (1) applying five statistical independence tests between inferred traits and structured/visual attributes; and (2) proposing a Causal Representation Learning (CRL) framework for multi-modality & multi-measurement data with identifiability guarantees, validated on synthetic data, and then used it to learn latent representations and causal graphs on PersonaX.

**Strengths:**

1.	A New Multimodal Dataset.
The paper introduces PersonaX, a multimodal dataset that aligns two corpora and pairs LLM-inferred Big Five personality traits with facial imagery/embeddings and biographical attributes, enabling large-scale cross-modal analysis.
2.	Model and Prompt Selection Strategy
The authors conduct a detailed evaluation (Figure 2) of various models and prompt templates to select the most reliable combination for dataset construction.
3.	Two-Level Analysis Pipeline.
The proposed analysis framework consists of two levels:
	* Level I applies five statistical independence tests to systematically characterize dependencies between Big Five scores and structured/visual features.
	* Level II introduces a causal representation learning framework with identifiability guarantees. This framework is first validated on synthetic data and then applied to the real multimodal embeddings, uncovering interpretable cross-modal structures.

**Weaknesses:**

1.	Lack of Human Validation for LLM-Inferred Personality Traits.
The Big Five personality traits inferred by LLMs have not undergone human validation. While prior work has shown that LLMs can somehow perform personality analysis tasks, the release of a dataset built on such inferences should be accompanied by human verification to ensure quality. Although the authors carefully selected the model and prompt to reduce variance and improve consistency, this does not guarantee alignment with human judgments.
2.	Limited Generalizability.
The AthlePersona subset is restricted to male athletes, and the CelebPersona subset primarily consists of public figures. As such, the conclusions drawn in this paper may not generalize well to broader or more diverse populations.

**Questions:**

1. Has any effort been made to assess the consistency between LLM-inferred and human-assessed Big Five personality traits?
2. Could the inclusion of league information in the prompt introduce or reinforce stereotypes?
3. How can the current analysis of athletes and public figures offer for generalizing to broader populations?

**Details Of Ethics Concerns:**

The authors have made several efforts to mitigate risks of data misuse. Specifically, they sourced the data from publicly available platforms with informed consent, and they only released embeddings with additional obfuscation rather than raw data, explicitly stating non-commercial usage restrictions. However, given the sensitive nature of the original data, I would recommend that the ethics reviewers take a closer look and evaluate the ethical implications more thoroughly.

---

> ### Author Response · Authors · 2025-11-21
> **Response to Reviewer GsZk (1)**
>
> We sincerely thank the reviewer for the balanced and careful review. Your comments on ethical considerations, generalization, and validation were highly valuable and helped us further improve the clarity and completeness of our work.
>
> ---
>
> **Q1:** "Lack of human validation for LLM-inferred personality traits. Has any effort been made to assess the consistency between LLM-inferred and human-assessed Big Five traits?"
>
> **A1**: To strengthen human validation, we conducted a small-scale human annotation study to assess how closely the LLM-inferred behavioral traits align with human expert ratings. We randomly selected 10 samples each from AthlePersona and CelebPersona and invited 10 human experts (with backgrounds in psychology and behavioral science) who are sufficiently familiar with the individuals included in the samples through public sources such as interviews, biographies, and sports or entertainment coverage. These experts annotated each sample on the five Big Five behavioral dimensions (Openness, Conscientiousness, Extraversion, Agreeableness, and Neuroticism) using the same 3-point scale (1–3). This yielded 100 sets of human annotations for each dataset, which we compared with the corresponding 5-D LLM-inferred vectors.
>
> We evaluated agreement between human annotations and LLM-inferred scores using the L₂ norm distance (overall difference between the two vectors), the normalized L₂ distance (scale-adjusted error), and the similarity score (1 - normalized L₂, representing the percentage of human–model alignment).
>
> | Dataset      | Model             | Raw L₂ norm ↓ | Normalized L₂ ↓ | Similarity ↑ |
> | :----------- | :---------------- | :-------------: | :---------------: | :-----------: |
> | CelebPersona | ChatGPT-4o-Latest |      0.80      |       0.18       |     0.82     |
> |              | Gemini-2.5-Pro    |      0.94      |       0.21       |     0.79     |
> |              | Llama-4-Maverick  |      1.00      |       0.22       |     0.78     |
> | AthlePersona | ChatGPT-4o-Latest |      0.91      |       0.20       |     0.80     |
> |              | Gemini-2.5-Pro    |      1.24      |       0.26       |     0.74     |
> |              | Llama-4-Maverick  |      1.44      |       0.29       |     0.71     |
>
> Normalized L₂ distances (0.18–0.29) correspond to 71–82% human–model similarity, indicating that LLM-inferred trait scores are largely consistent with human expert annotations. ChatGPT-4o-Latest shows the closest alignment across both datasets, supporting the external validity and reliability of the inferred behavioral traits.
>
> **Q2:** "Limited generalizability (male athletes, public figures)."
>
> **A2:** We acknowledge these limitations. We have analyzed their potential effects for gender bias below, and will solve the celebrity (public figure) bias in the future study by expanding our datasets.
>
> (1) To address the gender bias concern, we conducted an additional sensitivity and subgroup analysis on CelebPersona, which includes both male and female individuals. We divided the dataset by gender and re-ran the KCI independence tests using two subsets, respectively. The results are presented below, showing that 78/90=86.7% of the relations remain consistent across genders (male vs. female). This indicates that key cross-modal dependencies are stable across different gender groups in some way. While our cohorts remain population-specific, this analysis provides empirical evidence that gender differences do not substantially alter the core behavioral associations.
>
> - KCI Independence Test Results on male/female subgroup of CelebPersona dataset (0 = dependent, 1 = independent):
>
> | Feature           | **Male** |      |        |        |        | **Female** |      |        |        |        |
> | :---------------- | :------------: | :---: | :----: | :----: | :----: | :--------------: | :---: | :----: | :----: | :----: |
> | |Open.|Cons.|Extra.|Agree.|Neuro.|Open.|Cons.|Extra.|Agree.|Neuro.|
> |Birth_Year|0|1|0|0|0|0|1|0|0|0|
> |Birth_Month|1|1|1|1|1|1|1|1|1|1|
> |Birth_Day|1|1|1|1|1|1|1|1|1|1|
> |Occupation|0|0|0|0|0|0|0|0|0|0|
> |Latitude|0|0|0|1|0|0|0|0|0|0|
> |Longitude|0|0|0|1|0|0|0|0|0|0|
> |Height|0|1|1|1|1|0|1|1|1|1|
> |Weight|1|0|0|1|1|0|0|0|1|1|
> |Big_Nose|0|1|0|1|0|1|0|0|1|0|
> |Big_Lips|0|0|0|1|1|0|0|0|1|1|
> |Oval_Face|0|1|1|1|1|1|0|0|0|1|
> |Double_Chin|1|0|1|1|0|1|0|1|1|0|
> |Pointy_Nose|0|1|1|1|0|0|0|0|0|1|
> |Narrow_Eyes|1|1|0|1|1|1|1|0|1|1|
> |High_Cheekbones|1|0|0|0|0|1|0|0|0|0|
> |Arched_Eyebrows|0|1|0|1|1|0|1|0|1|1|
> |Bushy_Eyebrows|1|1|0|0|0|1|1|0|0|0|
> |Receding_Hairline|1|0|1|1|0|1|0|1|1|0|
>
> (2) To address the celebrity (public figure) bias concern, as outlined in Section 5, we are actively expanding PersonaX to include more non-celebrity populations to improve demographic coverage in future releases.

---

> ### Author Response · Authors · 2025-11-21
> **Response to Reviewer GsZk (2)**
>
> **Q3:** "Could the inclusion of league information in the prompt introduce or reinforce stereotypes?"
>
> **A3:** We appreciate this important concern. League identifiers are included only as structured contextual information to ensure accurate subject identification, particularly for individuals from different league but sharing the same name. Since professional leagues differ in competitive culture and media exposure, such information helps the model capture relevant behavioral context and avoid confusion between individuals with identical names.
>
> To examine whether league identifiers could introduce stereotype bias, we conducted an ablation study using a random subset of 1000 samples from the AthlePersona dataset. Trait inference was performed with three LLMs, and our CRL method was applied to obtain causal graphs both with and without league information in the prompt. We then compared the resulting graphs using SHD across three random seeds. As shown below, the differences are minimal (SHD = 1–2, approximately one or two edges out of fifty), indicating that the inclusion or removal of league identifiers has a negligible effect on the learned causal structure. This demonstrates that league information improves inference stability by providing meaningful contextual cues, while its influence on causal dependencies and potential stereotype bias is insignificant.
>
> | Comparison                                       | Seed 1 | Seed 2 | Seed 3 |
> | :----------------------------------------------- | :----: | :----: | :----: |
> | SHD (with league info. vs. without league info.) |   1   |   2   |   1   |
>
> **Q4:** "How can the current analysis of athletes and public figures offer generalization to broader populations?"
>
> **A4:** Based on our current experiments, we observe several consistent behavioral patterns across the two datasets (CelebPersona and AthlePersona), suggesting partial generalizability of the discovered causal relationships. For instance, the independence analyses in Appendix A5.2 and A5.3 show that personality traits such as Openness and Conscientiousness are dependent on background attributes like birth year and birth place across both datasets, indicating shared behavioral structures despite domain differences. Similarly, the causal graphs in Figure 6 and Figure A6 consistently show connections from shared latent variables to trait latents and from trait latents to face image latents, reflecting a stable causal hierarchy across populations. While these common patterns suggest that some core behavioral dependencies are domain-invariant, differences in finer details highlight the influence of population-specific factors.
>
> Therefore, although the results indicate promising cross-domain consistency, we recognize that direct generalization to all demographics is limited. To further close this gap, we are actively expanding PersonaX to include more non-celebrity and diverse individuals to ensure fairer representation and to evaluate the invariance of causal structures across broader populations.
>
> ---
>
> Thank you once again for your thoughtful and constructive feedback. Your comments have helped us enhance the ethical transparency, generalization discussion, and overall robustness of our work.

---

> > ### Comment · Reviewer_GsZk · 2025-11-28
> >
> > The author addressed most of my concerns. I would like to increase my score to 6. But I cannot do it on the system. I hope the AC will notice my comment.

---

> > > ### Author Response · Authors · 2025-11-28
> > >
> > > We sincerely appreciate the reviewer for checking our responses and increasing your score. Thanks again for your constructive comments!

---

### Official Review · Reviewer_aPN1 · 2025-10-26

**Soundness:** 3
**Presentation:** 3
**Contribution:** 3
**Rating:** 6
**Confidence:** 4

**Summary:**

The paper presents PersonaX, a multimodal dataset framework designed to explore behavioral trait inference through large language models (LLMs). It introduces two datasets: CelebPersona, comprising 9,444 public figures, and AthlePersona, including 4,181 professional athletes from seven major leagues. Each record combines LLM-inferred Big Five personality traits, facial embeddings, and biographical metadata. The authors conduct a two-level analysis: structured statistical independence testing and a causal representation learning (CRL) framework with identifiability guarantees, evaluated on both synthetic and real data. The results reveal interpretable cohort-specific dependencies, and the dataset is released under privacy-preserving, non-commercial terms.

**Strengths:**

1. Comprehensive dataset linking visual and behavioral attributes through LLM inference.
2. Multi-level analysis combining structured statistical tests and causal representation learning.
3. Privacy-aware release strategy (embeddings only, non-commercial use).
4. Systematic evaluation of LLMs and prompts for reliability.
5. Interpretability and cross-domain insights (e.g., appearance importance for celebrities vs. organizational traits for athletes).
6. Well-documented, reproducible methodology.

**Weaknesses:**

1. No human-ground-truth validation for LLM-inferred Big Five scores. The inferred personality traits are never compared against human annotations or standardized psychometric assessments, making it difficult to evaluate their validity. A small-scale validation study would substantially improve confidence in the dataset’s behavioral realism.
2. Gender and domain bias (athletes are all male, celebrities are mostly Western). These sampling biases limit generalizability and may influence the discovered dependencies, as LLMs and facial embeddings could internalize social and cultural stereotypes from biased data sources.
3. Causal interpretations rely on untestable assumptions. The proposed causal representation learning (CRL) framework depends on theoretical assumptions such as independent measurement noise, injectivity of modality-specific measurement functions, and sufficient diversity across modalities to ensure identifiability. These assumptions are difficult to verify empirically in real-world multimodal datasets like PersonaX, where both LLM-inferred traits and facial embeddings are high-dimensional and potentially correlated. Without diagnostic checks or robustness analyses, the causal graphs may reflect associations rather than true underlying causal structure, making interpretability plausible but not necessarily causal.
4. Lack of downstream benchmark tasks to demonstrate the utility of learned latents.

**Questions:**

1. Have you performed any human validation or external benchmarking to assess the reliability of the LLM-inferred Big Five traits, even on a small subset of samples?
2. The causal representation learning (CRL) framework depends on assumptions such as measurement noise independence and modality injectivity. What empirical checks or sensitivity analyses did you perform to verify that these assumptions approximately hold for PersonaX?
3. How stable are the learned causal graphs across random seeds or when a modality (e.g., images or text) is removed? Do key dependency patterns persist?
4. Have you compared the CRL embeddings to any non-causal multimodal baselines (e.g., CLIP-style or multimodal autoencoders) to verify that the causal formulation provides additional structural or representational advantages?
5. Could you elaborate on the privacy-preserving mechanism applied to the released facial embeddings, specifically, whether the transformation is invertible or resistant to re-identification attacks?

**Details Of Ethics Concerns:**

1. Bias and Fairness: The dataset includes only male athletes in AthlePersona and primarily Western celebrities in CelebPersona, which could reinforce gender and cultural biases in inferred behavioral traits.
2. Privacy and Re-identification Risk: Although only embeddings are released, the data originates from identifiable public figures. Without strong formal guarantees, embeddings may still be linkable to original identities through inversion or matching.
3. Potential Misuse of Trait Inference: The ability to infer personality-like traits from facial or biographical data could be misused in sensitive contexts such as hiring, surveillance, or profiling, even if the dataset is released for non-commercial research only.
4. Legal and Licensing Uncertainty: Images of celebrities and athletes are likely sourced from public domains but may not be explicitly cleared for redistribution, raising potential copyright or terms-of-use compliance questions.
5. Responsible Data Release: The LLM-generated annotations represent synthetic behavioral traits. It is important to ensure that the generation, aggregation, and release of such data follow responsible dataset practices, including transparent documentation of consent and data provenance.

---

> ### Author Response · Authors · 2025-11-21
> **Response to Reviewer aPN1 (1)**
>
> We sincerely appreciate the reviewer’s time and thoughtful review. Your feedback on validation, bias, and causal assumptions has been invaluable in improving the completeness and rigor of our work. Below we respond to each point in detail.
>
> ---
>
> **Q1:** "No human-ground-truth validation for LLM-inferred Big Five scores. Have you performed any human validation or external benchmarking to assess the reliability of the LLM-inferred Big Five traits, even on a small subset of samples?"
>
> **A1**: To strengthen human-ground-truth validation, we conducted a small-scale human annotation study to assess how closely the LLM-inferred behavioral traits align with human expert ratings. We randomly selected 10 samples each from AthlePersona and CelebPersona and invited 10 human experts (with backgrounds in psychology and behavioral science) who are sufficiently familiar with the individuals included in the samples through public sources such as interviews, biographies, and sports or entertainment coverage. These experts annotated each sample on the five Big Five behavioral dimensions (Openness, Conscientiousness, Extraversion, Agreeableness, and Neuroticism) using the same 3-point scale (1–3). This yielded 100 sets of human annotations for each dataset, which we compared with the corresponding 5-D LLM-inferred vectors.
>
> We evaluated agreement between human annotations and LLM-inferred scores using the L₂ norm distance (overall difference between the two vectors), the normalized L₂ distance (scale-adjusted error), and the similarity score (1 - normalized L₂, representing the percentage of human–model alignment).
>
> | Dataset | Model | Raw L₂ norm ↓ | Normalized L₂ ↓ | Similarity ↑ |
> | :----------- | :---------------- | :-------------: | :---------------: | :-----------: |
> | CelebPersona | ChatGPT-4o-Latest |      0.80      |       0.18       |     0.82     |
> |   | Gemini-2.5-Pro    |      0.94      |       0.21       |     0.79     |
> |   | Llama-4-Maverick  |      1.00      |       0.22       |     0.78     |
> | AthlePersona | ChatGPT-4o-Latest |      0.91      |       0.20       |     0.80     |
> |   | Gemini-2.5-Pro    |      1.24      |       0.26       |     0.74     |
> |   | Llama-4-Maverick  |      1.44      |       0.29       |     0.71     |
>
> Normalized L₂ distances (0.18–0.29) correspond to 71–82% human–model similarity, indicating that LLM-inferred trait scores are largely consistent with human expert annotations. ChatGPT-4o-Latest shows the closest alignment across both datasets, supporting the external validity and reliability of the inferred behavioral traits.
>
> **Q2:** "Gender and domain bias (athletes all male, celebrities mostly Western). These sampling biases limit generalizability and may influence the discovered dependencies, as LLMs and facial embeddings could internalize social and cultural stereotypes from biased data sources."
>
> **A2:** We acknowledge these limitations. We have analyzed their potential effects for gender bias below, and will solve the celebrity domain bias in the future study by expanding our datasets.
>
> (1) To address the gender bias concern, we conducted an additional sensitivity and subgroup analysis on CelebPersona, which includes both male and female individuals. We divided the dataset by gender and re-ran the KCI independence tests using two subsets, respectively. The results are presented below, showing that 78/90=86.7% of the relations remain consistent across genders (male vs. female). This indicates that key cross-modal dependencies are stable across different gender groups in some way. While our cohorts remain population-specific, this analysis provides empirical evidence that gender differences do not substantially alter the core behavioral associations.
>
> - KCI Independence Test Results on male/female subgroup of CelebPersona dataset (0 = dependent, 1 = independent):
>
> | Feature  | **Male** | | | | | **Female** | | | | |
> | :---------------- | :------------: | :---: | :----: | :----: | :----: | :--------------: | :---: | :----: | :----: | :----: |
> | |Open.|Cons.|Extra.|Agree.|Neuro.|Open.|Cons.|Extra.|Agree.|Neuro.|
> |Birth_Year|0|1|0|0|0|0|1|0|0|0|
> |Birth_Month|1|1|1|1|1|1|1|1|1|1|
> |Birth_Day|1|1|1|1|1|1|1|1|1|1|
> |Occupation|0|0|0|0|0|0|0|0|0|0|
> |Latitude|0|0|0|1|0|0|0|0|0|0|
> |Longitude|0|0|0|1|0|0|0|0|0|0|
> |Height|0|1|1|1|1|0|1|1|1|1|
> |Weight|1|0|0|1|1|0|0|0|1|1|
> |Big_Nose|0|1|0|1|0|1|0|0|1|0|
> |Big_Lips|0|0|0|1|1|0|0|0|1|1|
> |Oval_Face|0|1|1|1|1|1|0|0|0|1|
> |Double_Chin|1|0|1|1|0|1|0|1|1|0|
> |Pointy_Nose|0|1|1|1|0|0|0|0|0|1|
> |Narrow_Eyes|1|1|0|1|1|1|1|0|1|1|
> |High_Cheekbones|1|0|0|0|0|1|0|0|0|0|
> |Arched_Eyebrows|0|1|0|1|1|0|1|0|1|1|
> |Bushy_Eyebrows|1|1|0|0|0|1|1|0|0|0|
> |Receding_Hairline|1|0|1|1|0|1|0|1|1|0|
>
> (2) To address the demongraphic/celebrity bias concern, as outlined in Section 5, we are actively expanding PersonaX to include more non-celebrity populations to improve demographic coverage and fairness in future releases.

---

> ### Author Response · Authors · 2025-11-21
> **Response to Reviewer aPN1 (2)**
>
> **Q3:** "Causal interpretations rely on untestable assumptions (noise independence, injectivity, modality diversity). The causal representation learning (CRL) framework depends on assumptions such as measurement noise independence and modality injectivity. What empirical checks or sensitivity analyses did you perform to verify that these assumptions approximately hold for PersonaX?"
>
> **A3:** In light of your constructive comments, we conducted extensive ablations and diagnostics across both CelebPersona and AthlePersona to verify all theoretical premises: Theorem 1 (A1–A4), Theorem 2 (shared-subspace identifiability), and Theorem 3 (sparsity or component-wise identifiability). We summarized the verifications of all the assumptions in the table below. These diagnostics show that those theoretical assumptions generally hold in practice. The embeddings are well-posed and differentiable, multi-measurements are consistent yet distinct, shared subspaces are well aligned, and the causal graphs are sparse and stable.
>
> |Theorem/Assumption|Dataset|Verification Metric(s)|Key Results|Interpretation|
> |:--|:--|:--|:--|:--|
> |**Theorem 1 – Latent identifiability conditions**|||||
> |A1 – Well-posed probability(bounded & continuous trait embeddings)|CelebPersona|min–max range, per-sample variance|Range [–0.43,0.22], mean var ≈2.8×10⁻⁴|Trait embeddings are continuous, bounded, and numerically stable, satisfying well-posedness.|
> ||AthlePersona|min–max range, per-sample variance|Range [–0.39,0.22], mean var ≈2.8×10⁻⁴|Continuous and bounded with stable variance across LLMs.|
> |A2 – Modality variability/injectivity|CelebPersona|Pairwise R² (GPT↔Gemini↔LLaMA)|R²≈0.891±0.01|Strong cross-predictability; consistent yet diverse multi-measurement traits.|
> ||AthlePersona|Pairwise R²|R²≈0.876±0.01|High mutual predictability across LLMs; variability maintained.|
> |A3 – Measurement changes(distinct but correlated LLM embeddings)|CelebPersona|HSIC p-values & cosine similarity|p<0.01 for all pairs; cos≈0.91–0.93|Distinct yet aligned measurement distributions; satisfies measurement-change condition.|
> ||AthlePersona|HSIC p<0.01; cos≈0.89–0.90|Meaningful diversity among measurements; consistent across domains.|Distinct yet aligned measurement distributions; satisfies measurement-change condition.|
> |A4 – Differentiability|CelebPersona|Avg grad-norm (enc/dec)=1.4/1.6; NaN/Inf=0%; Jacobian 1.8±0.5|Stable gradients, smooth mapping|Neural mappings are continuous and differentiable, confirming smoothness assumption.|
> ||AthlePersona|Avg grad-norm (enc/dec)=1.3/1.7; NaN/Inf=0%; Jacobian 1.9±0.6|Stable gradients, smooth mapping|Same as above; differentiability satisfied.|
> |**Theorem 2 – Shared-subspace identifiability**|CelebPersona (3 shared dims)|CCA (s₁–s₂–s₃)=[0.75,0.83,0.72]; mean R²≈0.70|Strong alignment without collapse; stable shared subspace across LLMs.|Entropy regularization successfully isolates a common shared representation while preserving modality-specific diversity.|
> ||AthlePersona (2 shared dims)|CCA (s₁–s₂)=0.72; R²≈0.65|Robust shared alignment amid higher textual noise.|Confirms a consistent shared latent structure despite domain variability.|
> |**Theorem 3 – Component-wise identifiability/sparsity**|AthlePersona|12 latents,14 edges(of 132)→10.6%density|Graph extremely sparse (avg deg≈1.17).|Learned structure is minimal yet sufficient, consistent with the sparsity principle.|
> ||CelebPersona|18 latents,22 edges(of 306)→7.2%density|Sparse, interpretable structure (avg deg≈1.22); small ΔSHD≤5.|Edge density<10% and stable under small threshold shifts, confirming a robust sparse graph.|
>
> **Q4:** "Lack of downstream benchmark tasks to demonstrate the utility of learned latents."
>
> **A4:** To evaluate the practical value of the learned latent representations, we conducted downstream classification experiments using the CRL latents as input features. For *AthlePersona* , we used 2 shared, 5 image, and 5 trait latents (12 dimensions in total) to predict league category (7 major leagues). For *CelebPersona* , we used 3 shared, 10 image, and 5 trait latents (18 dimensions) to predict occupation category (12 classes). The datasets contain 4,181 and 9,444 samples respectively and were split into training and test sets with an 8:2 ratio. We trained a simple 3-layer MLP (hidden sizes 256 and 128, ReLU activations, dropout 0.2, softmax output) using the AdamW optimizer (learning rate 1e−3, batch size 128).
>
> | Dataset| Label | Accuracy (our CRL latents) | Accuracy (baseline MMCRL latents) |
> | :---- | :---- | :---------: | :--------------: |
> | AthlePersona | League category (7 major leagues) | **89%**|83%|
> | CelebPersona | Occupation category (12 classes)  | **84%**|77%|
>
> Our CRL embeddings outperform MMCRL embeddings by about 6-7 percentage points in accuracy. These results demonstrate that the learned latent representations are not only interpretable but also discriminative and practically useful for real-world prediction tasks.

---

> > ### Comment · Reviewer_aPN1 · 2025-11-25
> >
> > Thank you for providing the additional experiments and clarifications. They helped address my concerns.

---

> > > ### Author Response · Authors · 2025-11-25
> > >
> > > We are happy to see that your concerns are well addressed. We sincerely appreciate your time and effort in evaluating our paper. Your constructive feedback has greatly enhanced our work. Thank you very much!

---

> ### Author Response · Authors · 2025-11-21
> **Response to Reviewer aPN1 (3)**
>
> **Q5:** "How stable are the learned causal graphs across random seeds or when a modality (e.g., images or text) is removed? Do key dependency patterns persist?"
>
> **A5:** We tested robustness to random initialization by retraining the CRL model on AthlePersona with three random seeds under identical configurations. The resulting Structural Hamming Distance (SHD) between runs was minimal, as shown below. The pairwise Structural Hamming Distance (SHD) was only one to two edges (<3% of total), confirming that the learned causal structures are highly reproducible and robust, and that the key dependency patterns still persist.
>
> | Comparison  | Seed 1 vs Seed 2 | Seed 1 vs Seed 3 | Seed 2 vs Seed 3 | Average ± SD |
> | :---------- | :--------------: | :--------------: | :--------------: | :-----------: |
> | SHD (edges) |        1        |        2        |        1        |  1.3 ± 0.6  |
>
> **Q6:** "Have you compared CRL embeddings to non-causal multimodal baselines to verify that the causal formulation provides additional structural or representational advantages?"
>
> **A6:** Yes. We compared our CRL embeddings with two non-causal multimodal baselines, BetaVAE and MCL. As shown in Figure 5(c), our method achieves the highest performance on both the R² (representation accuracy) and MCC (latent disentanglement) metrics. Specifically, our CRL method attains R² = 0.96 and  MCC = 0.92 , significantly surpassing MCL (R² = 0.79, MCC = 0.63) and BetaVAE (R² = 0.68, MCC = 0.53). These improvements indicate that our multi-measurement multi-modality causal modeling framework learns more accurate and disentangled latent structures, confirming that causal constraints provide substantial advantages over standard non-causal multimodal methods.
>
> **Q7:** "Could you elaborate on the privacy-preserving mechanism applied to the released facial embeddings? Specifically, whether the transformation is invertible or resistant to re-identification attacks?"
>
> **A7:** Yes, of course. Each image and text embedding is obfuscated through an invertible transformation applied within the latent space. Although the transformation is mathematically invertible, it is deliberately non-reversible in practice, because the specific function will not be disclosed to ensure privacy protection. Both modalities (image and text) originate from foundation-model embeddings, which are then processed by this additional mapping. As a result, no directly identifiable information is retained, and re-identification attacks become computationally infeasible. This design effectively balances privacy and usability, preserving the research utility of the released embeddings while providing strong protection against identity reconstruction.
>
> ---
>
> Thank you again for your insightful and constructive feedback. Your comments have been instrumental in improving the methodological clarity, empirical rigor, and transparency of our work.

---

### Official Review · Reviewer_e8Dt · 2025-10-31

**Soundness:** 3
**Presentation:** 4
**Contribution:** 3
**Rating:** 8
**Confidence:** 5

**Summary:**

This paper proposes PersonaX, a collection of multimodal datasets including LLM-inferred behavior trait assessments with facial imagery and structured biographical features, and the paper introduces a two-level analysis pipeline: (1) structured independence testing to discover trait-attribute dependencies (2) a multimodal, multi-measurement causal representation learning (CRL) framework with identifiability guarantees that learns shared and modality-specific latent factors. The authors provide dataset construction details, systematic LLM selection and prompt design, synthetic benchmarks, and real-world analyses on PersonaX.

**Strengths:**

1.Novel multimodal benchmark. PersonaX fills a clear gap by uniting LLM-derived personality assessments with facial and biographical modalities, enabling large-scale, cross-modal trait analysis not available in existing resources.
2.Rigorous LLM measurement pipeline. The authors comprehensively benchmark multiple LLMs, prompt templates, and scoring schemes, reporting consistency and missing-rate analyses that enhance transparency and reproducibility.
3.Integration of theory and practice. The proposed CRL framework offers identifiability guarantees and is validated through both synthetic benchmarks and real-world data, effectively bridging theoretical development and empirical evaluation.

**Weaknesses:**

1.Insufficient external validation of LLM-inferred traits. Despite internal consistency checks, the lack of human-annotated validation or behavioral benchmarks undermines confidence in the real-world accuracy of the inferred personality traits.
2.Limited demographic diversity and potential bias. The datasets primarily include male athletes and high-profile celebrities, introducing demographic and socioeconomic biases that constrain generalization and may lead to spurious correlations.
3.Under-specified theoretical assumptions in practice. The identifiability results rely on strong technical conditions whose practical validity for noisy, LLM-based embeddings is not empirically examined, limiting interpretability of real-world findings.

**Questions:**

1.Strengthen label validity. Incorporate human-annotated subsets or downstream behavioral prediction tasks to quantitatively verify the reliability and utility of the inferred traits.
2.Address cohort bias. Broaden demographic coverage or conduct sensitivity and subgroup analyses to isolate dataset-specific dependencies and clarify the scope of generalization.
3.Clarify theoretical grounding. Provide ablations and diagnostics verifying identifiability-related assumptions and assess the impact of the embedding obfuscation transformation on model performance.

---

> ### Author Response · Authors · 2025-11-21
> **Response to Reviewer e8Dt (1)**
>
> We sincerely thank the reviewer for the insightful comments and positive recognition. Your points on validation, bias, and theoretical assumptions have been extremely helpful. We summarize the issues raised and address them one by one below.
>
> ---
>
> **Q1:** "Insufficient external validation of LLM-inferred traits. Strengthen label validity. Incorporate human-annotated subsets or downstream behavioral prediction tasks to quantitatively verify the reliability and utility of the inferred traits."
>
> **A1:** We appreciate this valuable comment. To strengthen external validation, we conducted a small-scale human annotation study to assess how closely the LLM-inferred behavioral traits align with human expert ratings. We randomly selected 10 samples each from AthlePersona and CelebPersona and invited 10 human experts (with backgrounds in psychology and behavioral science) who are sufficiently familiar with the individuals included in the samples through public sources such as interviews, biographies, and sports or entertainment coverage. These experts annotated each sample on the five Big Five behavioral dimensions (Openness, Conscientiousness, Extraversion, Agreeableness, and Neuroticism) using the same 3-point scale (1–3). This yielded 100 sets of human annotations for each dataset, which we compared with the corresponding 5-D LLM-inferred vectors.
>
> We evaluated agreement between human annotations and LLM-inferred scores using the L₂ norm distance (overall difference between the two vectors), the normalized L₂ distance (scale-adjusted error), and the similarity score (1 - normalized L₂, representing the percentage of human–model alignment).
>
> | Dataset      | Model             | Raw L₂ norm ↓ | Normalized L₂ ↓ | Similarity ↑ |
> | :----------- | :---------------- | :-------------: | :---------------: | :-----------: |
> | CelebPersona | ChatGPT-4o-Latest |      0.80      |       0.18       |     0.82     |
> |              | Gemini-2.5-Pro    |      0.94      |       0.21       |     0.79     |
> |              | Llama-4-Maverick  |      1.00      |       0.22       |     0.78     |
> | AthlePersona | ChatGPT-4o-Latest |      0.91      |       0.20       |     0.80     |
> |              | Gemini-2.5-Pro    |      1.24      |       0.26       |     0.74     |
> |              | Llama-4-Maverick  |      1.44      |       0.29       |     0.71     |
>
> Normalized L₂ distances (0.18–0.29) correspond to 71–82% human–model similarity, indicating that LLM-inferred trait scores are largely consistent with human expert annotations. ChatGPT-4o-Latest shows the closest alignment across both datasets, supporting the external validity and reliability of the inferred behavioral traits.
>
> **Q2:** "Limited demographic diversity and potential bias. Address cohort bias. Broaden demographic coverage or conduct sensitivity and subgroup analyses to isolate dataset-specific dependencies and clarify the scope of generalization."
>
> **A2:** We acknowledge this limitation and have taken steps to examine and mitigate it.
>
> (1) To address the gender bias concern, we conducted an additional sensitivity and subgroup analysis on CelebPersona, which includes both male and female individuals. We divided the dataset by gender and re-ran the KCI independence tests using two subsets, respectively. The results are presented below, showing that 78/90=86.7% of the relations remain consistent across genders (male vs. female). This indicates that key cross-modal dependencies are stable across different gender groups in some way. While our cohorts remain population-specific, this analysis provides empirical evidence that gender differences do not substantially alter the core behavioral associations.
>
> - KCI Independence Test Results on male/female subgroup of CelebPersona dataset (0 = dependent, 1 = independent):
>
> | Feature           | **Male** |      |        |        |        | **Female** |      |        |        |        |
> | :---------------- | :------------: | :---: | :----: | :----: | :----: | :--------------: | :---: | :----: | :----: | :----: |
> | |Open.|Cons.|Extra.|Agree.|Neuro.|Open.|Cons.|Extra.|Agree.|Neuro.|
> |Birth_Year|0|1|0|0|0|0|1|0|0|0|
> |Birth_Month|1|1|1|1|1|1|1|1|1|1|
> |Birth_Day|1|1|1|1|1|1|1|1|1|1|
> |Occupation|0|0|0|0|0|0|0|0|0|0|
> |Latitude|0|0|0|1|0|0|0|0|0|0|
> |Longitude|0|0|0|1|0|0|0|0|0|0|
> |Height|0|1|1|1|1|0|1|1|1|1|
> |Weight|1|0|0|1|1|0|0|0|1|1|
> |Big_Nose|0|1|0|1|0|1|0|0|1|0|
> |Big_Lips|0|0|0|1|1|0|0|0|1|1|
> |Oval_Face|0|1|1|1|1|1|0|0|0|1|
> |Double_Chin|1|0|1|1|0|1|0|1|1|0|
> |Pointy_Nose|0|1|1|1|0|0|0|0|0|1|
> |Narrow_Eyes|1|1|0|1|1|1|1|0|1|1|
> |High_Cheekbones|1|0|0|0|0|1|0|0|0|0|
> |Arched_Eyebrows|0|1|0|1|1|0|1|0|1|1|
> |Bushy_Eyebrows|1|1|0|0|0|1|1|0|0|0|
> |Receding_Hairline|1|0|1|1|0|1|0|1|1|0|
>
> (2) To address the demongraphic/celebrity bias concern, as outlined in Section 5, we are actively expanding PersonaX to include more non-celebrity populations to improve demographic coverage and fairness in future releases.

---

> ### Author Response · Authors · 2025-11-21
> **Response to Reviewer e8Dt (2)**
>
> **Q3:** "Under-specified theoretical assumptions in practice. Provide ablations and diagnostics verifying identifiability-related assumptions and assess the impact of the embedding obfuscation transformation on model performance."
>
> **A3:** We appreciate the reviewer’s request for clearer empirical validation of our identifiability assumptions.
>
> (1) First, we conducted extensive ablations and diagnostics across both CelebPersona and AthlePersona to verify all theoretical premises: Theorem 1 (A1–A4), Theorem 2 (shared-subspace identifiability), and Theorem 3 (sparsity or component-wise identifiability). We summarized the verifications of all the assumptions in the table below. These diagnostics show that those theoretical assumptions generally hold in practice. The embeddings are well-posed and differentiable, multi-measurements are consistent yet distinct, shared subspaces are well aligned, and the causal graphs are sparse and stable.
>
> |Theorem/Assumption|Dataset|Verification Metric(s)|Key Results|Interpretation|
> |:--|:--|:--|:--|:--|
> |**Theorem 1 – Latent identifiability conditions**|||||
> |A1 – Well-posed probability(bounded & continuous trait embeddings)|CelebPersona|min–max range, per-sample variance|Range [–0.43,0.22], mean var ≈2.8×10⁻⁴|Trait embeddings are continuous, bounded, and numerically stable, satisfying well-posedness.|
> ||AthlePersona|min–max range, per-sample variance|Range [–0.39,0.22], mean var ≈2.8×10⁻⁴|Continuous and bounded with stable variance across LLMs.|
> |A2 – Modality variability/injectivity|CelebPersona|Pairwise R² (GPT↔Gemini↔LLaMA)|R²≈0.891±0.01|Strong cross-predictability; consistent yet diverse multi-measurement traits.|
> ||AthlePersona|Pairwise R²|R²≈0.876±0.01|High mutual predictability across LLMs; variability maintained.|
> |A3 – Measurement changes(distinct but correlated LLM embeddings)|CelebPersona|HSIC p-values & cosine similarity|p<0.01 for all pairs; cos≈0.91–0.93|Distinct yet aligned measurement distributions; satisfies measurement-change condition.|
> ||AthlePersona|HSIC p<0.01; cos≈0.89–0.90|Meaningful diversity among measurements; consistent across domains.|Distinct yet aligned measurement distributions; satisfies measurement-change condition.|
> |A4 – Differentiability|CelebPersona|Avg grad-norm (enc/dec)=1.4/1.6; NaN/Inf=0%; Jacobian 1.8±0.5|Stable gradients, smooth mapping|Neural mappings are continuous and differentiable, confirming smoothness assumption.|
> ||AthlePersona|Avg grad-norm (enc/dec)=1.3/1.7; NaN/Inf=0%; Jacobian 1.9±0.6|Stable gradients, smooth mapping|Same as above; differentiability satisfied.|
> |**Theorem 2 – Shared-subspace identifiability**|CelebPersona (3 shared dims)|CCA (s₁–s₂–s₃)=[0.75,0.83,0.72]; mean R²≈0.70|Strong alignment without collapse; stable shared subspace across LLMs.|Entropy regularization successfully isolates a common shared representation while preserving modality-specific diversity.|
> ||AthlePersona (2 shared dims)|CCA (s₁–s₂)=0.72; R²≈0.65|Robust shared alignment amid higher textual noise.|Confirms a consistent shared latent structure despite domain variability.|
> |**Theorem 3 – Component-wise identifiability/sparsity**|AthlePersona|12 latents,14 edges(of 132)→10.6%density|Graph extremely sparse (avg deg≈1.17).|Learned structure is minimal yet sufficient, consistent with the sparsity principle.|
> ||CelebPersona|18 latents,22 edges(of 306)→7.2%density|Sparse, interpretable structure (avg deg≈1.22); small ΔSHD≤5.|Edge density<10% and stable under small threshold shifts, confirming a robust sparse graph.|
>
> (2) Then, we also assessed the impact of the embedding obfuscation transformation used for privacy preservation. To test whether the privacy-preserving obfuscation (an invertible linear transformation) affects causal discovery, we compared graphs learned from original and obfuscated embeddings under three random seeds. The Structural Hamming Distance (SHD) was zero in all cases:
>
> | Dataset      | Seed 1 | Seed 2 | Seed 3 | Avg. SHD |
> | :----------- | :----: | :----: | :----: | :------: |
> | CelebPersona |   0   |   0   |   0   |   0.0   |
> | AthlePersona |   0   |   0   |   0   |   0.0   |
>
> This outcome is expected, as the transformation is mathematically equivalent to a change of coordinate basis in the embedding space. It does not alter conditional independencies or causal topology. Thus, the obfuscation step preserves both model performance and causal structure while ensuring data privacy.
>
> ---
>
> We sincerely thank reviewer for the thoughtful and constructive feedback. Your detailed comments on validation, demographic bias, and theoretical grounding have been invaluable in improving both the methodological rigor and empirical completeness of our work. We truly appreciate your time and effort in reviewing our paper.

---

### Official Review · Reviewer_vCuK · 2025-11-05

**Soundness:** 3
**Presentation:** 3
**Contribution:** 3
**Rating:** 6
**Confidence:** 3

**Summary:**

This paper introduces PersonaX, a set of curated multimodal datasets—CelebPersona and AthlePersona—incorporating LLM-inferred behavioral trait assessments (Big Five), facial embeddings, and structured biographical metadata for public figures and professional athletes. The authors present a two-level analytical framework: (1) statistical independence tests exploring modality-specific dependencies between behavioral traits and structured features, and (2) a novel causal representation learning (CRL) model with theoretical identifiability guarantees for multimodal, multi-measurement data. Empirical validations are performed using both synthetic and real-world datasets, demonstrating the proposed approach’s effectiveness in cross-modal trait analysis and causal discovery.

**Strengths:**

PersonaX provides a valuable resource that bridges LLM-inferred behavioral traits with biographical and visual data across thousands of public personas, addressing a clear gap in multimodal behavioral trait datasets.
Methodological advances in causal representation learning: The paper proposes an end-to-end causal modeling framework tailored to multimodal, multi-measurement data, supported by reasonably rigorous identifiability theory.
Extensive experiments are conducted both on synthetic settings and real-world data, with multiple baselines including BetaVAE, MCL, and MMCRL.
The data pipeline and experimental protocols are well-illustrated, with detailed breakdowns in supplementary materials.

**Weaknesses:**

Limited diversity and representativeness in dataset sampling. CelebPersona and AthlePersona focus on public figures and male athlete, which are not demographically representative of broader populations. This constrains the generalizability of reported findings and may bias downstream analyses, as acknowledged but not addressed in either analysis or ablation.
Experiments lack real-world baseline comparisons for CRL. The effectiveness of the CRL component is shown mostly on synthetic MNIST and real-world data in isolation; it would be more persuasive to demonstrate improvements over state-of-the-art real-world baselines or alternative causal discovery methods.
Causal graph interpretability and stability concerns. The mapping between learned latents and original human-interpretable traits is indirect.
Presentation and clarity issues in tables/figures. Table 1 is visually dense; important metrics such as overall score should be more clearly emphasized.

**Questions:**

What is the impact of prompt selection and LLM choice on downstream causal graph structure and variable interpretability? Are the reported findings robust to these design choices?
How would the proposed method scale to or generalize with with populations beyond public celebrities and male athletes, and what steps are planned to mitigate cohort-specific biases?

---

> ### Author Response · Authors · 2025-11-21
> **Response to Reviewer vCuK (1)**
>
> We sincerely appreciate the reviewer’s time and constructive feedback. Your comments have been highly valuable in helping us strengthen both the analysis and presentation of our work. Below we provide detailed point-by-point responses.
>
> ---
>
> **Q1:** "Limited diversity and representativeness in dataset sampling. CelebPersona and AthlePersona focus on public figures and male athletes, which are not demographically representative of broader populations. This constrains the generalizability of reported findings and may bias downstream analyses, as acknowledged but not addressed in either analysis or ablation."
>
> **A1:** We fully agree that demographic diversity is important for improving generalizability. (1) First we would like to explain that: our focus on public figures and male athletes was a deliberate and practical choice to ensure ethical accessibility and data reliability. The LLM-inferred behavioral traits rely on publicly available sources such as interviews, biographies, and media coverage, which are more abundant for well-known individuals. Consequently, CelebPersona was built upon the established CelebA dataset, providing a legally and ethically sound basis for multimodal analysis. Similarly, AthlePersona focuses on male athletes because professional men’s leagues offer richer and more standardized public data, allowing for consistent and reproducible LLM-based inference. As noted in Section 2.2, “Findings should be interpreted as population-specific rather than universal. These focused cohorts provide consistency, and the diversity across domains, for example, different sports leagues, creates valuable opportunities to study invariant causal patterns.”
>
> (2) To address the gender bias concern, we conducted an additional sensitivity and subgroup analysis on CelebPersona, which includes both male and female individuals. We divided the dataset by gender and re-ran the KCI independence tests using two subsets, respectively. The results are presented below, showing that 78/90=86.7% of the relations remain consistent across genders, with only minor variations in features related to facial appearance. This indicates that key cross-modal dependencies (such as the link between openness and visual expressivity) are stable across demographic groups. While our cohorts remain population-specific, this analysis provides empirical evidence that gender differences do not substantially alter the core behavioral associations.
>
> - KCI Independence Test Results on male/female subgroup of CelebPersona dataset (0 = dependent, 1 = independent):
> | Feature | **Male** | | | | | **Female** | | || |
> | :-- | :--: | :--: | :--: | :----: | :----: | :---: | :---: | :----: | :----: | :----: |
> | |Open.|Cons.|Extra.|Agree.|Neuro.|Open.|Cons.|Extra.|Agree.|Neuro.|
> |Birth_Year|0|1|0|0|0|0|1|0|0|0|
> |Birth_Month|1|1|1|1|1|1|1|1|1|1|
> |Birth_Day|1|1|1|1|1|1|1|1|1|1|
> |Occupation|0|0|0|0|0|0|0|0|0|0|
> |Latitude|0|0|0|1|0|0|0|0|0|0|
> |Longitude|0|0|0|1|0|0|0|0|0|0|
> |Height|0|1|1|1|1|0|1|1|1|1|
> |Weight|1|0|0|1|1|0|0|0|1|1|
> |Big_Nose|0|1|0|1|0|1|0|0|1|0|
> |Big_Lips|0|0|0|1|1|0|0|0|1|1|
> |Oval_Face|0|1|1|1|1|1|0|0|0|1|
> |Double_Chin|1|0|1|1|0|1|0|1|1|0|
> |Pointy_Nose|0|1|1|1|0|0|0|0|0|1|
> |Narrow_Eyes|1|1|0|1|1|1|1|0|1|1|
> |High_Cheekbones|1|0|0|0|0|1|0|0|0|0|
> |Arched_Eyebrows|0|1|0|1|1|0|1|0|1|1|
> |Bushy_Eyebrows|1|1|0|0|0|1|1|0|0|0|
> |Receding_Hairline|1|0|1|1|0|1|0|1|1|0|
>
> (3) To address the demongraphic bias concern, as outlined in Section 5, we are actively expanding PersonaX to include more non-celebrity populations to improve demographic coverage and fairness in future releases.
>
> **Q2:** "Experiments lack real-world baseline comparisons for CRL. The effectiveness of the CRL component is shown mostly on synthetic MNIST and real-world data in isolation; it would be more persuasive to demonstrate improvements over state-of-the-art real-world baselines or alternative causal discovery methods."
>
> **A2:** We appreciate this thoughtful suggestion. While our initial CRL evaluation focused on synthetic data for interpretability, we have now included real-world baseline for comparison. Specifically, we added experiments using MMCRL (ICLR 2025), a recent state-of-the-art multimodal causal representation learning approach, on both AthlePersona and CelebPersona and following identical experimental settings (as mentioned in Appendix A8.1). The new results (we put in Appendix A9.3) show that MMCRL, which assumes single-measurement data, produces denser and less interpretable causal graphs. In contrast, our multi-measurement CRL yields sparser, more stable, and semantically coherent cross-modal structures. Quantitatively, our method achieves higher causal alignment metrics (for example, MCC and $R^2$ in synthetic results as shown in Section 4.4) and here qualitatively shows clearer separation between visual and behavioral latent factors (in this real-world experiments). These findings confirm the advantages of our design for real-world multimodal causal analysis.

---

> ### Author Response · Authors · 2025-11-21
> **Response to Reviewer vCuK (2)**
>
> **Q3:** "Causal graph interpretability and stability concerns. The mapping between learned latents and original human-interpretable traits is indirect."
>
> **A3:** To enhance interpretability and stability, we performed additional analyses using RCIT independence tests between the five Big Five traits (Openness, Conscientiousness, Extraversion, Agreeableness, and Neuroticism) and the five learned latent variables. These tests reveal statistically dependent relationships, which we used to assign interpretable meanings to each latent component. As each latent may capture multiple plausible behavioral factors, we provide representative examples to clarify their interpretation. Full details and visualizations are included in Appendix A9.2 and Figure A7.
>
> **Q4:** "Presentation and clarity issues in tables/figures. Table 1 is visually dense; important metrics such as overall score should be more clearly emphasized."
>
> **A4:** We appreciate this observation. Following your suggestion, we revised Table 1 for improved readability by shortening model names (moved full list to Appendix A4.1), emphasizing the overall score (OS), and refining the layout for clarity. The revised table now highlights key comparisons more clearly.
>
> **Q5:** "What is the impact of prompt selection and LLM choice on downstream causal graph structure and variable interpretability? Are the reported findings robust to these design choices?"
>
> **A5:** We agree that prompt and LLM selection can influence downstream analysis. To ensure robustness, we systematically evaluated multiple prompt templates and LLM configurations (see Fig. 2 and Appendix A4.2). The results show high consistency (standard deviation < 0.3 across models and Manhattan distance < 1.0 across prompts) for the selected setup.
>
> We further tested robustness to random initialization by retraining the CRL model on AthlePersona with three random seeds under identical configurations. The resulting Structural Hamming Distance (SHD) between runs was minimal, as shown below:
>
> | Comparison  | Seed 1 vs Seed 2 | Seed 1 vs Seed 3 | Seed 2 vs Seed 3 | Average ± SD |
> | :---------- | :--------------: | :--------------: | :--------------: | :-----------: |
> | SHD (edges) |        1        |        2        |        1        |  1.3 ± 0.6  |
>
> The graphs differ by only one to two edges out of roughly fifty total (<3%), confirming reproducibility and robustness. All key cross-modal dependencies persist, showing that random initialization and LLM/prompt design choices have negligible influence on the inferred causal structure.
>
> **Q6:** "How would the proposed method scale to or generalize with populations beyond public celebrities and male athletes, and what steps are planned to mitigate cohort-specific biases?"
>
> **A6:** We acknowledge the importance of broader generalization and fairness. As discussed in Section 5, we are extending PersonaX to incorporate non-celebrity populations from diverse domains while maintaining the same multimodal causal inference framework. These extensions will enhance demographic coverage and allow systematic evaluation of cohort-specific biases. PersonaX is designed as an evolving research platform rather than a static resource. The current release should therefore be regarded as a controlled pilot study that establishes methodological validity before expansion to more heterogeneous and demographically balanced populations.
>
> ---
>
> Thank you again for your insightful feedback. We are grateful for the reviewer’s constructive comments, which have helped us improve both the rigor and clarity of the paper.

---

### Author Response · Authors · 2025-12-01
**Summary**

Dear Reviewers and Area Chair,

We sincerely thank you for your valuable time and efforts in evaluating our paper. In this work, we not only introduce **PersonaX**, two new multimodal datasets for behavioral trait analysis, but also present a **two-level causal analytical framework**, including a **novel causal representation learning model** with theoretical identifiability guarantees for multimodal and multi-measurement data.

Our submission initially received four reviews with scores of **8, 6, 6, and 4** (confidence levels 5, 4, 3, and 2, respectively). The main concerns raised by the reviewers focused on *dataset bias and generalization* (vCuK, e8Dt, aPN1, GsZk), *human validation of LLM-inferred labels* (e8Dt, aPN1, GsZk), *baseline comparisons* (vCuK, aPN1), and *theoretical assumptions* (e8Dt, aPN1).

Through our detailed rebuttal and clarifications, we are pleased to see that **Reviewer aPN1’s** concerns have been fully addressed, and he/she has maintained a positive overall assessment. **Reviewer GsZk** also acknowledged that most concerns were resolved and expressed willingness to raise the score from **4 to 6**.

We are deeply grateful to all reviewers for their thoughtful and constructive feedback, which has significantly improved the clarity, rigor, and completeness of our work. Thank you once again for your time and consideration.

All the best.

---

### Meta-Review · Area_Chair_ZDqG · 2026-01-06

**Summary:**

The reviewers highlight several strengths of the work, noting that PersonaX is a novel and valuable multimodal benchmark that bridges LLM-inferred Big Five personality traits with facial and biographical data across thousands of public personas, filling a clear gap in existing behavioral trait datasets. The paper presents a well-motivated and carefully engineered data pipeline, including a systematic evaluation of LLMs, prompts, and scoring strategies that enhances transparency, reliability, and reproducibility. Methodologically, the proposed causal representation learning framework is a strong contribution, combining identifiability theory with practical applicability and validated through extensive experiments on both synthetic and real-world data against multiple baselines. The two-level analysis pipeline (integrating structured statistical tests with causal modeling) enables interpretable cross-modal insights, while the privacy-aware dataset release and thorough documentation further strengthen the paper’s rigor, usability, and impact for the multimodal and behavioral analysis research communities.

There were some initial concerns about generalizability, validity, and interpretability: the CelebPersona and AthlePersona datasets are demographically narrow (dominated by Western public figures and male athletes) introducing gender, cultural, and socioeconomic biases that may limit representativeness and may distort learned dependencies. The causal interpretations themselves rely on strong, difficult-to-verify theoretical assumptions, and the indirect mapping between learned latents and human-interpretable traits, combined with the absence of robustness diagnostics, raises the risk that learned graphs reflect correlations rather than true causal structure.

Overall, the reviewers are mostly positive and after the rebuttal the only slightly negative reviewer has decided to upgrade the score.

**Reviewer Concerns:**

The comments were quite positive from the beginning and the raised concerns have been well addressed by the authors in their rebuttal. The dataset and the proposed methodology are quite relevant and important (as stressed by the reviewers) and some of the concerns regarding the narrow demography, the causal interpretation, etc. were adequately addressed in the rebuttal.

**Reviewer Scores:**

Surely, the discussion phase would have helped in clarifying some of the minor concerns but even without this phase two of the reviewers agree that the comments were well addressed. Given that all the reviewers are positive I think the manuscript is quite important and it should be accepted.

---

### Decision · Program_Chairs · 2026-01-26

Accept (Poster)